# Solving Constrained Variational Inequalities via a First-order Interior Point-based Method

**Tong Yang**[*]
University of California, Berkeley
pptmiao@berkeley.edu

**Michael I. Jordan**[*]
University of California, Berkeley
jordan@cs.berkeley.edu

**Tatjana Chavdarova**[*]
University of California, Berkeley
tatjana.chavdarova@berkeley.edu

## Abstract

We develop an interior-point approach to solve constrained variational inequality (cVI) problems. Inspired by the efficacy of the *alternating direction method of multipliers* (ADMM) method in the single-objective context, we generalize ADMM to derive a first-order method for cVIs, that we refer to as **A**DMM-based interior-point method for **c**onstrained **VI**s (ACVI). We provide convergence guarantees for ACVI in two general classes of problems: *(i)* when the operator is $\xi$-monotone, and *(ii)* when it is monotone, some constraints are active and the game is not purely rotational. When the operator is, in addition, L-Lipschitz for the latter case, we match known lower bounds on rates for the gap function of $\mathcal{O}(1/\sqrt{K})$ and $\mathcal{O}(1/K)$ for the last and average iterate, respectively. To the best of our knowledge, this is the first presentation of a *first*-order interior-point method for the general cVI problem that has a global convergence guarantee. Moreover, unlike previous work in this setting, ACVI provides a means to solve cVIs when the constraints are nontrivial. Empirical analyses demonstrate clear advantages of ACVI over common first-order methods. In particular, *(i)* cyclical behavior is notably reduced as our methods approach the solution from the analytic center, and *(ii)* unlike projection-based methods that zigzag when near a constraint, ACVI efficiently handles the constraints.

## 1 Introduction

We are interested in the *constrained variational inequality* problem (Stampacchia, 1964):

$$\text{find } \boldsymbol{x}^\star \in \mathcal{X} \quad \text{s.t.} \quad \langle \boldsymbol{x} - \boldsymbol{x}^\star, F(\boldsymbol{x}^\star) \rangle \geq 0, \quad \forall \boldsymbol{x} \in \mathcal{X}, \tag{cVI}$$

where $\mathcal{X}$ is a subset of the Euclidean $n$-dimensional space $\mathbb{R}^n$, and where $F : \mathcal{X} \mapsto \mathbb{R}^n$ is a continuous map. Finding (an element of) the solution set $\mathcal{S}^\star_{\mathcal{X}, F}$ of cVI is a key problem in multiple fields such as economics and game theory. More pertinent to machine learning, CVIs generalize standard single-objective optimization, complementarity problems (Cottle & Dantzig, 1968), zero-sum games (von Neumann & Morgenstern, 1947; Rockafellar, 1970) and multi-player games. For example, solving cVI is the optimization problem underlying *reinforcement learning* (e.g., Omidshafiei et al., 2017)—and *generative adversarial networks* (Goodfellow et al., 2014). Moreover, even when training one set of parameters with one loss $f$, that is $F(\boldsymbol{x}) \equiv \nabla_{\boldsymbol{x}} f(\boldsymbol{x})$, a natural way to improve the model's robustness in some regard is to introduce an adversary to perturb the objective or the input, or to consider the worst sample distribution of the empirical objective. As has been noted in many problem domains, including robust classification (Mazuelas et al., 2020), adversarial training (Szegedy et al., 2014), causal inference (Christiansen et al., 2020), and robust objectives (e.g., Rothenhäusler et al., 2018), this leads to a min-max structure, which is an instance of the cVI problem. To see this, consider two sets of parameters (agents), $\boldsymbol{x}_1 \in \mathcal{X}_1$ and $\boldsymbol{x}_2 \in \mathcal{X}_2$, that share a loss/utility function, $f : \mathcal{X}_1 \times \mathcal{X}_2 \to \mathbb{R}$, which the first agent aims to minimize

---

[*]All authors contributed equally. Link to source code: https://github.com/Chavdarova/ACVI.

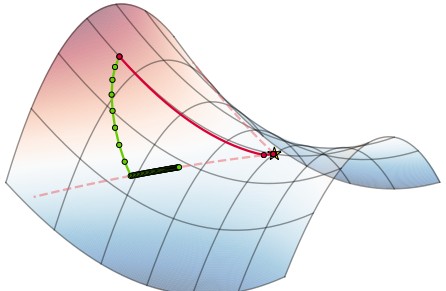

Figure 1: ACVI (Algorithm 1) and EG iterates—depicted in *red* and *green*, resp.—on the game:

$$\min_{x_1 \in \mathbb{R}_+} \max_{x_2 \in \mathbb{R}_+} 0.05 \cdot x_1^2 + x_1 x_2 - 0.05 \cdot x_2^2 \,.$$

The constraints are depicted with dashed lines and the iterates with circles. ACVI gets close to the Nash Equilibrium (⋆) in a single step, whereas EG zigzags when hitting a constraint. The remaining commonly used methods—GDA, OGDA, and LA-GDA—perform similarly to EG, see App. E.

and the second agent aims to maximize. Then the problem is to find a *saddle point* of $f$, i.e., a point $(\boldsymbol{x}_1^\star, \boldsymbol{x}_2^\star)$ such that $f(\boldsymbol{x}_1^\star, \boldsymbol{x}_2) \leq f(\boldsymbol{x}_1^\star, \boldsymbol{x}_2^\star) \leq f(\boldsymbol{x}_1, \boldsymbol{x}_2^\star)$. This corresponds to a cVI with $F(\boldsymbol{x}) \equiv [\nabla_{\boldsymbol{x}_1} f(\boldsymbol{x}_1, \boldsymbol{x}_2) \quad -\nabla_{\boldsymbol{x}_2} f(\boldsymbol{x}_1, \boldsymbol{x}_2)]^\mathsf{T}$.

Solving cVIs is significantly more challenging than single-objective optimization problems, due to the fact that $F$ is a general vector field, leading to "rotational" trajectories in parameter space (App. A). In response, the development of efficient algorithms with provable convergence has recently been the focus of interest in machine learning and optimization, particularly in the unconstrained setting, where $\mathcal{X} \equiv \mathbb{R}^n$ (e.g., Tseng, 1995; Daskalakis et al., 2018; Mokhtari et al., 2019; 2020; Golowich et al., 2020b; Azizian et al., 2020; Chavdarova et al., 2021a; Gorbunov et al., 2022; Bot et al., 2022).

In many applications, however, we have *constraints* on (part of) the decision variable $\boldsymbol{x}$, that is, $\mathcal{X}$ is often a strict subset of $\mathbb{R}^n$. As an example, let us revisit the aforementioned distributionally robust prediction problem: consider a linear setting (cf. Eq. 1 in Rothenhäusler et al., 2018) and class of parametrized distributions $\triangle \equiv \{\boldsymbol{w} \in \mathbb{R}^d | \boldsymbol{w} \geq 0, \boldsymbol{e}^\mathsf{T}\boldsymbol{w} = 1\}$, where $\boldsymbol{e} \in \mathbb{R}^d$ is a vector of all ones. Thus, the robust problem is: $\min_{\boldsymbol{x} \in \mathbb{R}^n} \max_{\boldsymbol{w} \in \mathbb{R}^d} \boldsymbol{w}^\mathsf{T}(\boldsymbol{y} - \boldsymbol{D}\boldsymbol{x})$, *subject to* $\boldsymbol{w} \geq 0$, $\boldsymbol{e}^\mathsf{T}\boldsymbol{w} = 1$, where $\boldsymbol{D} \in \mathbb{R}^{d \times n}$ contains $d$ samples of an $n$-dimensional covariate vector, and $\boldsymbol{y} \in \mathbb{R}^d$ is the vector of target variables (the constraint $\boldsymbol{w} \leq 1$ is implied). This illustrates that given a standard minimization problem, its robustification immediately leads to an instance of the cVI problem; see further examples in § 5. Additional example applications include (i) machine learning applications in business, finance, and economics where often the sum of the decision variables—representing, for example, *resources*—cannot exceed a specific value, (ii) contract theory (e.g. §2.3.2 in (Bates et al., 2022) where one player is the parameters of a probability distribution as above), and (iii) solving optimal control problems numerically, among others.

Significantly fewer works address the convergence of first-order optimization methods in the constrained setting; see § 2 for an overview. Recently, Cai et al. (2022) established a convergence rate for the *projected extragradient method* (Korpelevich, 1976), when $F$ is monotone and Lipschitz (see § 3 for definitions). However, (i) the proof that the authors presented is computer-assisted, which makes it hard to interpret and of limited usefulness for inspiring novel (e.g., accelerated) methods, and (ii) the considered setting assumes the projection is fast to compute and thus ignores the projection in the rate. The latter assumption only holds in rare cases when the constraints are relatively simple so that operations such as clipping suffice. However, when the inequality and/or equality constraints are of a general form, each EG update requires *two* projections (see App. A.4). Each projection requires solving a new/separate *constrained* optimization problem, which if given general constraints implies the need for a *second-order method* as explained next.

*Interior point* (IP) methods are the de facto family of iterative algorithms for constrained optimization. These methods enjoy well-established guarantees and theoretical understanding in the context of single-objective optimization [see, e.g., Boyd & Vandenberghe (2004), Ch.11, Megiddo (1989), Wright (1997)], and have extensions to a wide range of problem settings (e.g., Tseng, 1993; Nesterov & Nemirovski, 1994; Nesterov & Todd, 1998; Renegar, 2001; Wright, 2001). They build on a natural idea of solving a simplified homotopic problem that makes it possible to "smoothly" transition to the original complex problem; see § 3.1. Several works extend IP methods to cVI, by applying the second-order Newton method to a modified *Karush-Kuhn-Tucker* (KKT) system appropriate for the cVI (Ralph & Wright, 2000; Qi & Sun, 2002; Fan & Yan, 2010; Monteiro & Pang, 1996; Chen et al., 1998). Many of these approaches, however, rely on strong assumptions—see § 2. Moreover, although these methods enjoy fast convergence in terms of the number of iterations, each iteration involves the computation of the Jacobian of $F$ (or Hessian when $F \equiv \nabla f(\boldsymbol{x})$) which quickly becomes prohibitive for large dimension $n$ of $\boldsymbol{x}$. Hence first-order methods are preferred in practice.

We are currently missing a *first*-order optimization method for solving cVI with *general* constraints. Accordingly, in this paper, we focus on the following open question:

> *Can we derive **first-order** algorithms for the cVI problem that (i) can be applied when **general** constraints are given, and that (ii) have **global** convergence guarantees?*

In this paper, we develop precisely such a method. To mitigate the computational burden of second-derivative computation, we replace the Newton step of the traditional IP methods with the *alternating direction method of multipliers* (ADMM) method. ADMM was designed with a different purpose: it is applicable only when the objective is separable into two or more different functions whose arguments are non disjoint—see § 3.1 for full description—and can be seen as equivalent to *Douglas–Rachford* operator splitting (Douglas & Rachford, 1956) applied in the *dual* space (see e.g. Lin et al., 2022). ADMM owes its popularity primarily to its computational efficiency (Boyd et al., 2011) for large-scale machine learning problems and its fast convergence in some machine-learning settings (e.g., Nishihara et al., 2015). The core idea of our approach is to reformulate the original cVI problem with equality and inequality constraints via the KKT conditions, so as to apply ADMM in such a way that the subproblems of the resulting algorithm have desirable properties (see § 4.1). That is, by generalizing the technique underlying ADMM, we derive a novel first-order algorithm for solving monotone VIs with very general constraints. Furthermore, this framework can be used to design novel algorithms for solving cVIs; see App. C.

Our contributions can be summarized as follows:

- Based on the KKT system for the constrained VI problem and the ADMM technique, we derive an algorithm that we refer to as the *ADMM-based Interior Point Method for Constrained VIs* (ACVI)—see § 4.1 and Algorithm 1.
- We prove the global convergence of ACVI given two sets of assumptions: (i) when $F$ is $\xi$-monotone, and (ii) when it is monotone, the constraints are active at the solution, and the game is not purely rotational. By further assuming $F$ is a Lipschitz operator, we upper bound the rate of decrease of the gap function and we match the known lower bound for the gap function of $\mathcal{O}(1/\sqrt{K})$ for the last iterate—see § 4.2.
- Empirically, we document two notable advantages of ACVI over popular projection-based saddle-point methods: (i) the ACVI iterates exhibit significantly reduced rotations, as they approach the solution from the analytic center, and (ii) while projection-based methods show extensive zigzagging when hitting a constraint, ACVI avoids this, resulting in more efficient updates—§ 5.

Our convergence guarantees are parameter-free, meaning these do not require a priori knowledge of the constants of the problem (such as the Lipschitz constant), and, interestingly, the convergence guarantee does not require that $F$ is Lipschitz. This assumption is solely used to express the rate of decrease of the gap function (in contrast to the extragradient method (Korpelevich, 1976) where such an assumption is necessary to show convergence). To the best of our knowledge, the proposed ACVI method is the first first-order IP algorithm for VIs with a global convergence proof.

## 2 RELATED WORK

**Unconstrained VIs: methods and guarantees.** Apart from the standard gradient descent ascent (GDA) method, among the most commonly used methods for VI optimization are the extragradient method (EG, Korpelevich, 1976), optimistic GDA (OGDA, Popov, 1980), and the lookahead method (LA, Zhang et al., 2019; Chavdarova et al., 2021b). (See App. A for a full description). In contrast to gradient fields (as in a single-objective setting), when $F$ is a general vector field, the last iterate can be far from the solution even though the average iterate converges to it (Daskalakis et al., 2018; Chavdarova et al., 2019). This is problematic since it implies that the average convergence guarantee is weaker in the sense that it may not extend to more general setups where we can no longer rely on the convexity of $\mathcal{X}$. Golowich et al. (2020b;a) provided a last-iterate lower bound of $\mathcal{O}(\frac{1}{\tilde{p}\sqrt{K}})$ for the broad class of $\tilde{p}$-*stationary canonical linear iterative* ($\tilde{p}$-SCLI) first-order methods (Arjevani et al., 2016). An extensive line of further work has provided guarantees for the last iterate for other problem classes. For the general monotone VI (MVI) class, the following $\tilde{p}$-SCLI methods come with guarantees that match the lower bound: (i) Golowich et al. (2020b) obtained a rate in terms of

the gap function relying on first- and second-order smoothness of $F$, and Gorbunov et al. (2022) obtained a rate of $\mathcal{O}\left(\frac{1}{K}\right)$ in terms of reducing the squared norm of the operator relying on first-order smoothness of $F$ (Assumption 1), using a computer-assisted proof, and (ii) Golowich et al. (2020b) and Chavdarova et al. (2021a) provided the best iterate rate for OGDA.

**Constrained zero-sum and VI classes of problems.** Gidel et al. (2017b) extended the Frank-Wolfe (Frank & Wolfe, 1956; Jaggi, 2013; Lacoste-Julien & Jaggi, 2015) method—also known as the *conditional gradient* (V.F Demyanov, 1970)—to solve a subclass of cVI, specifically constrained zero-sum problems. This extension was carried out under a strong convex-concavity assumption and also under the assumption that the constraint set is *strongly* convex; that is, it has sublevel sets that are strongly convex functions (Vial, 1983). Daskalakis & Panageas (2019) provided an asymptotic proof for the last iterate for zero-sum convex-concave constrained problems for the *optimistic multiplicative weights update* (OMWU) method. Wei et al. (2021) focused on OGDA and OMWU in the constrained setting and provided convergence rates for bilinear games over the simplex. In her seminal work, Korpelevich (1976) proposed the classical (projected) extragradient method (EG)—see App. A—and proved its convergence for monotone (c)VIs with an $L$-Lipschitz operator, and Cai et al. (2022) established a rate with respect to the gap function using a computer-aided proof. Tseng (1995) built on (Pang, 1987) and provided a linear convergence rate for EG in the setting of *strongly* monotone $F$, whereas Malitsky (2015) focused on the same constrained setting but on the projected reflected gradient method. Diakonikolas (2020) obtained parameter-free guarantee for Halpern iteration (Halpern, 1967) for cocoercive operators. Goffin et al. (1997) described a second-order cutting-plane method for solving pseudomonotone VIs with linear inequalities.

**Interior point (IP) methods in single-objective and VI settings.** Traditionally IP methods primarily express the inequality constraints by augmenting the objective with a log-barrier penalty (see § 3.1), and then use Newton's method to solve the subproblem (Boyd & Vandenberghe, 2004). The latter involves computing either the inverse of a large matrix or a Cholesky decomposition and yet it can be highly efficient in low dimensions as it requires only a few iterations to converge. When the dimensionality of the variable is large, however, the computation becomes infeasible. Among other IP variants that address this issue, Lin et al. (2018) replaced the Newton step with the ADMM method, which is known to be highly scalable in terms of the dimension (Boyd et al., 2011). In the context of cVIs, a few works apply IP methods, mostly Newton-based (e.g., Nesterov & Nemirovski, 1994, Chapter 7). Monteiro & Pang (1996) analyze path-following IP methods for complementarity problems, which are a subclass of cVI, using local homeomorphic maps. Chen et al. (1998) provided a superlinear global convergence rate of the smoothing Newton method when $F$ is semi-smooth for *box constrained* VIs. Similarly, Qi & Sun (2002); Qi et al. (2000) focused on the smoothing Newton method and provided the rate for the outer loop. Ralph & Wright (2000) showed superlinear convergence for MVI problems under inequality constraints, under the following set of assumptions: (i) existence of a *strictly* complementary solution, (ii) full rank of the Jacobian of the active constraints at the solution, and (iii) twice differentiable constraints. They provided a *local* convergence rate. Fan & Yan (2010) considered inequality constraints and proposed a second-order Newton-based method that has global convergence guarantees under certain conditions. A rate was not provided.

## 3 PRELIMINARIES

**Notation.** Bold small and capital letters denote vectors and matrices, respectively. Sets are denoted with curly capital letters, e.g., $\mathcal{S}$. The Euclidean norm of $\boldsymbol{v}$ is denoted by $\|\boldsymbol{v}\|$, and the inner product in Euclidean space with $\langle \cdot, \cdot \rangle$. With $\odot$ we denote element-wise product. We let $[n]$ denote $\{1, \ldots, n\}$ and let $\boldsymbol{e}$ denote vector of all 1's. We let $\boldsymbol{x} \perp \boldsymbol{y}$ denote $\boldsymbol{x}$ and $\boldsymbol{y}$ are perpendicular.

In the remainder of the paper, we consider a general setting in which the constraint set $\mathcal{C} \subseteq \mathcal{X}$ is defined as an intersection of finitely many inequalities and linear equalities:

$$\mathcal{C} = \{\boldsymbol{x} \in \mathbb{R}^n | \varphi_i(\boldsymbol{x}) \leq 0, i \in [m], \ \boldsymbol{C}\boldsymbol{x} = \boldsymbol{d}\}, \tag{CS}$$

where each $\varphi_i : \mathbb{R}^n \mapsto \mathbb{R}$, $\boldsymbol{C} \in \mathbb{R}^{p \times n}$, $\boldsymbol{d} \in \mathbb{R}^p$, where we assume $rank(\boldsymbol{C}) = p$. For brevity, with $\varphi$ we denote the concatenated $\varphi_i(\cdot), i \in [m]$, and in the remainder of the paper, each $\varphi_i \in C^1(\mathbb{R}^n), i \in [m]$ and is convex. For convenience we denote:

$$\mathcal{C}_{\leq} \triangleq \{\boldsymbol{x} \in \mathbb{R}^n \,|\, \varphi(\boldsymbol{x}) \leq \boldsymbol{0}\}, \quad \mathcal{C}_{<} \triangleq \{\boldsymbol{x} \in \mathbb{R}^n \,|\, \varphi(\boldsymbol{x}) < \boldsymbol{0}\}, \quad \text{and} \quad \mathcal{C}_{=} \triangleq \{\boldsymbol{y} \in \mathbb{R}^n | \boldsymbol{C}\boldsymbol{y} = \boldsymbol{d}\};$$

thus the *relative* interior of $\mathcal{C}$ is *int* $\mathcal{C} \triangleq \mathcal{C}_{<} \cap \mathcal{C}_{=}$, and we consider *int* $\mathcal{C} \neq \emptyset$ and $\mathcal{C}$ is compact.

In the following, we list the definitions and assumptions we refer to later on.

**Definition 1** ((strong/$\xi$) monotonicity). *An operator $F : \mathcal{X} \supseteq \mathcal{S} \to \mathbb{R}^n$ is* monotone *on $\mathcal{S}$ if:* $\langle \boldsymbol{x} - \boldsymbol{x}', F(\boldsymbol{x}) - F(\boldsymbol{x}') \rangle \geq 0, \forall \boldsymbol{x}, \boldsymbol{x}' \in \mathcal{S}$. *$F$ is said to be $\xi$–monotone on $\mathcal{S}$ iff there exist $c > 0$ and $\xi > 1$ such that $\langle \boldsymbol{x} - \boldsymbol{x}', F(\boldsymbol{x}) - F(\boldsymbol{x}') \rangle \geq c \|\boldsymbol{x} - \boldsymbol{x}'\|^\xi$, for all $\boldsymbol{x}, \boldsymbol{x}' \in \mathcal{S}$. Finally, $F$ is $\mu$-strongly monotone on $\mathcal{S}$ if there exists $\mu > 0$, such that $\langle \boldsymbol{x} - \boldsymbol{x}', F(\boldsymbol{x}) - F(\boldsymbol{x}') \rangle \geq \mu \|\boldsymbol{x} - \boldsymbol{x}'\|^2$, for all $\boldsymbol{x}, \boldsymbol{x}' \in \mathcal{S}$. Moreover, we say that an operator $F$ is* star–monotone, star–$\xi$-monotone *or* star–strongly-monotone *(on $\mathcal{S}$) if the respective definition holds for $\boldsymbol{x}' \equiv \boldsymbol{x}^\star$, where $\boldsymbol{x}^\star \in \mathcal{S}^\star_{\mathcal{S}, F}$.*

Note that the "star–" definitions are weaker relative to their respective non-star counterparts. The above definition holds similarly for unconstrained VIs, by setting $\mathcal{S} \equiv \mathbb{R}^n$. The analog for cVI of the function values used as a performance measure for convergence rates in convex optimization is the *gap function* (a.k.a., the *optimality* gap or *primal* gap), defined next.

**Definition 2** (gap function). *Given a candidate point $\boldsymbol{x}' \in \mathcal{X}$ and a map $F : \mathcal{X} \supseteq \mathcal{S} \to \mathbb{R}^n$ where $\mathcal{S}$ is compact, the* gap function *$\mathcal{G} : \mathbb{R}^n \to \mathbb{R}$ is defined as $\mathcal{G}(\boldsymbol{x}', \mathcal{S}) \triangleq \max_{\boldsymbol{x} \in \mathcal{S}} \langle F(\boldsymbol{x}'), \boldsymbol{x}' - \boldsymbol{x} \rangle$.*

Note that the gap function requires $\mathcal{S}$ to be compact in order to be defined (as otherwise, it can be infinite). We will rely on the following assumption to express our rates in terms of the gap function.

**Assumption 1** (first-order smoothness). *Let $F : \mathcal{X} \supseteq \mathcal{S} \to \mathbb{R}^n$ be an operator, we say that $F$ satisfies $L$-first-order smoothness on $\mathcal{S}$, or $L$-smoothness, if $F$ is an $L$-Lipschitz map; that is, there exists $L > 0$ such that $\|F(\boldsymbol{x}) - F(\boldsymbol{x}')\| \leq L \|\boldsymbol{x} - \boldsymbol{x}'\|$, for all $\boldsymbol{x}, \boldsymbol{x}' \in \mathcal{S}$.*

As an informal summary, a solution existence guarantee follows when $\mathcal{X}$ is compact; see Chapter 2.2 of (Facchinei & Pang, 2003), and App. A.2.

## 3.1 RELEVANT PATH-FOLLOWING INTERIOR-POINT METHODS AND ADMM

In this section, we overview the interior-point approach to single-objective optimization, focusing on aspects that are most relevant to our proposed method. Consider the following problem:

$$\min_{\boldsymbol{x}} f(\boldsymbol{x}) \qquad s.t. \quad \varphi(\boldsymbol{x}) \leq \boldsymbol{0} \quad and \quad \boldsymbol{C}\boldsymbol{x} = \boldsymbol{d}, \tag{cCVX}$$

where $f, \varphi_i : \mathbb{R}^n \to \mathbb{R}$ are convex and continuously differentiable, $\boldsymbol{x} \in \mathbb{R}^n$, $\boldsymbol{C} \in \mathbb{R}^{p \times n}$, and $\boldsymbol{d} \in \mathbb{R}^p$. IP methods solve problem (cCVX) by reducing it to a sequence of linear equality-constrained problems via a logarithmic barrier (see, e.g., Boyd & Vandenberghe, 2004, Chapter 11):

$$\min_{\boldsymbol{x}} f(\boldsymbol{x}) - \mu \sum_{i=1}^{m} \log(-\varphi_i(\boldsymbol{x})) \quad s.t. \quad \boldsymbol{C}\boldsymbol{x} = \boldsymbol{d}, \qquad with \quad \mu > 0. \tag{l-cCVX}$$

Assume that (l-cCVX) has a solution for each $\mu > 0$, and let $\boldsymbol{x}^\mu$ denote the solution of (l-cCVX) for a given $\mu$. The *central path* of (l-cCVX) is defined as the set of points $\boldsymbol{x}^\mu, \mu > 0$. Note that $\boldsymbol{x}^\mu \in \mathbb{R}^n$ is a strictly feasible point of (cCVX) as it satisfies $\varphi(\boldsymbol{x}^\mu) < \boldsymbol{0}$ and $\boldsymbol{C}\boldsymbol{x}^\mu = \boldsymbol{d}$.

**Alternating direction method of multipliers (ADMM) method.** ADMM (Glowinski & Marroco, 1975; Gabay & Mercier, 1976; Lions & Mercier, 1979; Glowinski & Le Tallec, 1989) is a gradient-based algorithm for convex optimization problems that splits the objective into subproblems each of which is easier to solve. Its popularity is due to its computational scalability (Boyd et al., 2011). Consider a problem of the following form:

$$\min_{\boldsymbol{x}, \boldsymbol{y}} f(\boldsymbol{x}) + g(\boldsymbol{y}) \quad s.t. \quad \boldsymbol{A}\boldsymbol{x} + \boldsymbol{B}\boldsymbol{y} = \boldsymbol{b}, \tag{ADMM-Pr}$$

where $f, g : \mathbb{R}^n \to \mathbb{R}$ are convex, $\boldsymbol{x}, \boldsymbol{y} \in \mathbb{R}^n$, $\boldsymbol{A}, \boldsymbol{B} \in \mathbb{R}^{n' \times n}$, and $\boldsymbol{b} \in \mathbb{R}^{n'}$. The augmented Lagrangian function, $\mathcal{L}_\beta(\cdot)$, of the (ADMM-Pr) problem is:

$$\mathcal{L}_\beta(\boldsymbol{x}, \boldsymbol{y}, \boldsymbol{\lambda}) = f(\boldsymbol{x}) + g(\boldsymbol{y}) + \langle \boldsymbol{A}\boldsymbol{x} + \boldsymbol{B}\boldsymbol{y} - \boldsymbol{b}, \boldsymbol{\lambda} \rangle + \frac{\beta}{2} \|\boldsymbol{A}\boldsymbol{x} + \boldsymbol{B}\boldsymbol{y} - \boldsymbol{b}\|^2, \tag{AL-CVX}$$

where $\beta > 0$ is referred to as the *penalty parameter*. If the augmented Lagrangian method is used to solve (AL-CVX), at each step $k$ we have: $\boldsymbol{x}_{k+1}, \boldsymbol{y}_{k+1} = \operatorname{argmin}_{\boldsymbol{x}, \boldsymbol{y}} \mathcal{L}_\beta(\boldsymbol{x}, \boldsymbol{y}, \boldsymbol{\lambda}_k)$ and $\boldsymbol{\lambda}_{k+1} = \boldsymbol{\lambda}_k + \beta(\boldsymbol{A}\boldsymbol{x}_{k+1} + \boldsymbol{B}\boldsymbol{y}_{k+1} - \boldsymbol{b})$, where the latter step is gradient ascent on the dual. In contrast, ADMM updates $\boldsymbol{x}$ and $\boldsymbol{y}$ in an alternating way as follows:

$$\boldsymbol{x}_{k+1} = \operatorname{argmin}_{\boldsymbol{x}} \mathcal{L}_\beta(\boldsymbol{x}, \boldsymbol{y}_k, \boldsymbol{\lambda}_k),$$

$$\boldsymbol{y}_{k+1} = \operatorname{argmin}_{\boldsymbol{y}} \mathcal{L}_\beta(\boldsymbol{x}_{k+1}, \boldsymbol{y}_k, \boldsymbol{\lambda}_k), \tag{ADMM}$$

$$\boldsymbol{\lambda}_{k+1} = \boldsymbol{\lambda}_k + \beta(\boldsymbol{A}\boldsymbol{x}_{k+1} + \boldsymbol{B}\boldsymbol{y}_{k+1} - \boldsymbol{b}).$$

## 4 *ACVI*: FIRST-ORDER ADMM-BASED IP METHOD FOR CONSTRAINED VIs

### 4.1 DERIVING THE ACVI ALGORITHM

In this section, we derive an interior-point method for the cVI problem that we refer to as ACVI (**ADMM**-based **i**nterior problem for constrained **VI**s). We first restate the cVI problem in a form that will allow us to derive an interior-point procedure. By the definition of cVI it follows (see §1.3 in Facchinei & Pang, 2003) that:

$$
\boldsymbol{x} \in \mathcal{S}^{\star}_{C,F} \Leftrightarrow
\begin{cases}
\boldsymbol{w} = \boldsymbol{x} \\
\boldsymbol{x} = \underset{\boldsymbol{z}}{argmin}\, F(\boldsymbol{w})^{\mathsf{T}}\boldsymbol{z} \\
s.t. \quad \varphi(\boldsymbol{z}) \leq \boldsymbol{0} \\
\quad\quad \boldsymbol{C}\boldsymbol{z} = \boldsymbol{d}
\end{cases}
\Leftrightarrow
\begin{cases}
F(\boldsymbol{x}) + \nabla\varphi^{\mathsf{T}}(\boldsymbol{x})\boldsymbol{\lambda} + \boldsymbol{C}^{\mathsf{T}}\boldsymbol{\nu} = \boldsymbol{0} \\
\boldsymbol{C}\boldsymbol{x} = \boldsymbol{d} \\
\boldsymbol{0} \leq \boldsymbol{\lambda} \perp \varphi(\boldsymbol{x}) \leq \boldsymbol{0},
\end{cases}
\tag{KKT}
$$

where $\boldsymbol{\lambda} \in \mathbb{R}^m$ and $\boldsymbol{\nu} \in \mathbb{R}^p$ are dual variables, and $\perp$ denotes perpendicular. Recall that we assume that $int\, \mathcal{C} \neq \emptyset$, thus, by the Slater condition (using the fact that $\varphi_i(\boldsymbol{x}), i \in [m]$ are convex) and the KKT conditions, the second equivalence holds, yielding the KKT system of cVI. Note that the above equivalence also guarantees the two solutions coincide; see Facchinei & Pang (2003, Prop. 1.3.4 (b)). Analogous to the method described in § 3, we add a log-barrier term to the objective to remove the inequality constraints and obtain the following modified version of (KKT):

$$
\begin{cases}
\boldsymbol{w} = \boldsymbol{x} \\
\boldsymbol{x} = \underset{\boldsymbol{z}}{argmin}\; F(\boldsymbol{w})^{\mathsf{T}}\boldsymbol{z} - \mu \sum_{i=1}^{m} \log\left(-\varphi_i(\boldsymbol{z})\right) \\
s.t. \quad \boldsymbol{C}\boldsymbol{z} = \boldsymbol{d}
\end{cases}
\Leftrightarrow
\begin{cases}
F(\boldsymbol{x}) + \nabla\varphi^{\mathsf{T}}(\boldsymbol{x})\boldsymbol{\lambda} + \boldsymbol{C}^{\mathsf{T}}\boldsymbol{\nu} = \boldsymbol{0} \\
\boldsymbol{\lambda} \odot \varphi(\boldsymbol{x}) + \mu\boldsymbol{e} = \boldsymbol{0} \\
\boldsymbol{C}\boldsymbol{x} - \boldsymbol{d} = \boldsymbol{0} \\
\varphi(\boldsymbol{x}) < \boldsymbol{0}, \boldsymbol{\lambda} > \boldsymbol{0},
\end{cases}
\tag{KKT-2}
$$

with $\mu > 0$, $\boldsymbol{e} \triangleq [1, \dots, 1]^{\mathsf{T}} \in \mathbb{R}^m$. Again, the equivalence holds by the KKT and the Slater condition. We derive the update rule at step $k$ via the following subproblem: $\min_{\boldsymbol{x}} F(\boldsymbol{w}_k)^{\mathsf{T}}\boldsymbol{x} - \mu \sum_{i=1}^{m} \log\left(-\varphi_i(\boldsymbol{x})\right), s.t.\, \boldsymbol{C}\boldsymbol{x} = \boldsymbol{d}$, where we fix $\boldsymbol{w} = \boldsymbol{w}_k$. Directly projecting on the equality constraint may cause the vectors to fall out of the domain of the log term. On the other hand, (i) $\boldsymbol{w}_k$ is a constant vector in this subproblem, and (ii) the objective is split, making ADMM a natural choice to solve the subproblem. Hence, we introduce a new variable $\boldsymbol{y} \in \mathbb{R}^n$ yielding:

$$
\begin{cases}
\underset{\boldsymbol{x},\boldsymbol{y}}{\min} F(\boldsymbol{w}_k)^{\mathsf{T}}\boldsymbol{x} + \mathbb{1}[\boldsymbol{C}\boldsymbol{x} = \boldsymbol{d}] - \mu \sum_{i=1}^{m} \log\left(-\varphi_i(\boldsymbol{y})\right), \\
s.t. \quad \boldsymbol{x} = \boldsymbol{y}
\end{cases}
\mathbb{1}[\boldsymbol{C}\boldsymbol{x} = \boldsymbol{d}] \triangleq
\begin{cases}
0, & \text{if } \boldsymbol{C}\boldsymbol{x} = \boldsymbol{d} \\
+\infty, & \text{if } \boldsymbol{C}\boldsymbol{x} \neq \boldsymbol{d}.
\end{cases}
\tag{1}
$$

Note that $\mathbb{1}[\boldsymbol{C}\boldsymbol{x} = \boldsymbol{d}]$ is a generalized real-valued convex function of $\boldsymbol{x}$. We introduce the following:

$$
\boldsymbol{P}_c \triangleq \boldsymbol{I} - \boldsymbol{C}^{\mathsf{T}}(\boldsymbol{C}\boldsymbol{C}^{\mathsf{T}})^{-1}\boldsymbol{C}, \tag{$\boldsymbol{P}_c$} \quad \text{and} \quad \boldsymbol{d}_c \triangleq \boldsymbol{C}^{\mathsf{T}}(\boldsymbol{C}\boldsymbol{C}^{\mathsf{T}})^{-1}\boldsymbol{d}, \tag{$d_c$-EQ}
$$

where $\boldsymbol{P}_c \in \mathbb{R}^{n \times n}$ and $\boldsymbol{d}_c \in \mathbb{R}^n$. The augmented Lagrangian of (1) is thus:

$$
\mathcal{L}_\beta(\boldsymbol{x}, \boldsymbol{y}, \boldsymbol{\lambda}) = F(\boldsymbol{w}_k)^{\mathsf{T}}\boldsymbol{x} + \mathbb{1}(\boldsymbol{C}\boldsymbol{x} = \boldsymbol{d}) - \mu \sum_{i=1}^{m} \log(-\varphi_i(\boldsymbol{y})) + \langle \boldsymbol{\lambda}, \boldsymbol{x} - \boldsymbol{y} \rangle + \frac{\beta}{2}\|\boldsymbol{x} - \boldsymbol{y}\|^2, \tag{AL}
$$

where $\beta > 0$ is the penalty parameter. Finally, using ADMM, we have the following update rule for $\boldsymbol{x}$ at step $k$:

$$
\boldsymbol{x}_{k+1} = \arg\min_{\boldsymbol{x} \in \mathcal{C}_=} \mathcal{L}_\beta(\boldsymbol{x}, \boldsymbol{y}_k, \boldsymbol{\lambda}_k) = \arg\min_{\boldsymbol{x} \in \mathcal{C}_=} \frac{\beta}{2}\left\|\boldsymbol{x} - \boldsymbol{y}_k + \frac{1}{\beta}(F(\boldsymbol{w}_k) + \boldsymbol{\lambda}_k)\right\|^2. \tag{2}
$$

This yields the following update for $\boldsymbol{x}$:

$$
\boldsymbol{x}_{k+1} = \boldsymbol{P}_c\left(\boldsymbol{y}_k - \frac{1}{\beta}(F(\boldsymbol{w}_k) + \boldsymbol{\lambda}_k)\right) + \boldsymbol{d}_c. \tag{X-EQ}
$$

For $\boldsymbol{y}$ and the dual variable $\boldsymbol{\lambda}$, we have:

$$
\boldsymbol{y}_{k+1} = \arg\min_{\boldsymbol{y}} \mathcal{L}_\beta(\boldsymbol{x}_{k+1}, \boldsymbol{y}, \boldsymbol{\lambda}_k) = \arg\min_{\boldsymbol{y}}\left(-\mu \sum_{i=1}^{m} \log\left(-\varphi_i(\boldsymbol{y})\right) + \frac{\beta}{2}\left\|\boldsymbol{y} - \boldsymbol{x}_{k+1} - \frac{1}{\beta}\boldsymbol{\lambda}_k\right\|^2\right),
$$
$$
\tag{Y-EQ}
$$

$$
\boldsymbol{\lambda}_{k+1} = \boldsymbol{\lambda}_k + \beta(\boldsymbol{x}_{k+1} - \boldsymbol{y}_{k+1}). \tag{$\lambda$-EQ}
$$

Next, we derive the update rule for $\boldsymbol{w}$. We set $\boldsymbol{w}_k$ to be the solution of the following equation:

$$\boldsymbol{w} + \frac{1}{\beta}\boldsymbol{P}_c F(\boldsymbol{w}) - \boldsymbol{P}_c\boldsymbol{y}_k + \frac{1}{\beta}\boldsymbol{P}_c\boldsymbol{\lambda}_k - \boldsymbol{d}_c = \boldsymbol{0}. \tag{W-EQ}$$

The following theorem ensures the solution of (W-EQ) exists and is unique, see App. B.1 for proof.

**Theorem 1** (W-EQ: solution uniqueness). *If $F$ is monotone on $\mathcal{C}_=$, the following statements hold true for the solution of (W-EQ): (i) it always exists, (ii) it is unique, and (iii) it is contained in $\mathcal{C}_=$.*

**Remark 1.** *Note that when there are no equality constraints, $\mathcal{C}_=$ becomes the entire space $\mathbb{R}^n$. Further notice that $\boldsymbol{w}_k = \boldsymbol{x}_{k+1}$, thus it is redundant to state it in the algorithm, and we remove $\boldsymbol{w}$.*

We summarize the full algorithm as Algorithm 1. For problems such as affine or low-dimensional VIs, or optimization over the probability simplex, (W-EQ) can be solved analytically, such that step 8 is fast to compute. For problems where (W-EQ) is cumbersome to solve analytically—e.g., in GANs—one could use optimization methods for the *unconstrained* case, e.g., EG and GDA, among others. See App. B.5 for further discussion. In the remaining discussion, where clear from context, we drop the superscript from the iterate $\boldsymbol{x}_k^{(t)}$.

---

**Algorithm 1** ACVI pseudocode.

---

1: **Input:** operator $F : \mathcal{X} \to \mathbb{R}^n$, constraints $\boldsymbol{Cx} = \boldsymbol{d}$ and $\varphi_i(\boldsymbol{x}) \leq 0, i = [m]$, hyperparameters $\mu_{-1}, \beta > 0, \delta \in (0,1)$, number of outer and inner loop iterations $T$ and $K$, resp.
2: **Initialize:** $\boldsymbol{y}_0^{(0)} \in \mathbb{R}^n, \boldsymbol{\lambda}_0^{(0)} \in \mathbb{R}^n$
3: $\boldsymbol{P}_c \triangleq \boldsymbol{I} - \boldsymbol{C}^\intercal(\boldsymbol{CC}^\intercal)^{-1}\boldsymbol{C}$          where $\boldsymbol{P}_c \in \mathbb{R}^{n \times n}$
4: $\boldsymbol{d}_c \triangleq \boldsymbol{C}^\intercal(\boldsymbol{CC}^\intercal)^{-1}\boldsymbol{d}$              where $\boldsymbol{d}_c \in \mathbb{R}^n$
5: **for** $t = 0, \ldots, T - 1$ **do**
6:      $\mu_t = \delta\mu_{t-1}$
7:      **for** $k = 0, \ldots, K - 1$ **do**
8:          Set $\boldsymbol{x}_{k+1}^{(t)}$ to be the solution of: $\boldsymbol{x} + \frac{1}{\beta}\boldsymbol{P}_c F(\boldsymbol{x}) - \boldsymbol{P}_c\boldsymbol{y}_k^{(t)} + \frac{1}{\beta}\boldsymbol{P}_c\boldsymbol{\lambda}_k^{(t)} - \boldsymbol{d}_c = \boldsymbol{0}$ (w.r.t. $\boldsymbol{x}$)
9:          $\boldsymbol{y}_{k+1}^{(t)} = \underset{\boldsymbol{y}}{argmin} -\mu_t \sum_{i=1}^m \log\big(-\varphi_i(\boldsymbol{y})\big) + \frac{\beta}{2}\left\| \boldsymbol{y} - \boldsymbol{x}_{k+1}^{(t)} - \frac{1}{\beta}\boldsymbol{\lambda}_k^{(t)} \right\|^2$
10:          $\boldsymbol{\lambda}_{k+1}^{(t)} = \boldsymbol{\lambda}_k^{(t)} + \beta(\boldsymbol{x}_{k+1}^{(t)} - \boldsymbol{y}_{k+1}^{(t)})$
11:      **end for**
12:      $(\boldsymbol{y}_0^{(t+1)}, \boldsymbol{\lambda}_0^{(t+1)}) \triangleq (\boldsymbol{y}_K^{(t)}, \boldsymbol{\lambda}_K^{(t)})$
13: **end for**

---

### 4.2 CONVERGENCE ANALYSIS

We consider two broad classes of problems. The first class assumes that $F$ is $\xi$-monotone on $\mathcal{C}_=$—a stronger assumption than monotonicity, yet weaker than strong monotonicity. The second setup requires that (i) $F$ is monotone, (ii) the constraints are active at the solution, and (iii) $F$ is not purely rotational. Note that (iii) is weaker than requiring that the active constraints at the solution form an acute angle with the operator; in other words, given the latter, the former holds due to monotonicity of $F$. (See App. B). Note that (iii) is not strong, as purely rotational games occur "almost never" in a Baire category sense (Kupka, 1963; Smale, 1963; Balduzzi et al., 2018; Hsieh et al., 2021). The proofs of the main theorems use the following lemma.

**Lemma 1** (Upper bound for $\mathcal{G}(\cdot)$). *When $F$ is L-Lipschitz on $\mathcal{C}_=$—as per Assumption 1—we have that any iterate $\boldsymbol{x}_k$ produced by Algorithm 1 satisfies $\mathcal{G}(\boldsymbol{x}_k, \mathcal{C}) \leq M_0 \|\boldsymbol{x}_k - \boldsymbol{x}^\star\|$, where $M_0 > 0$ depends linearly on L, and $\boldsymbol{x}^\star \in \mathcal{S}_{\mathcal{C},F}^\star$.*

To state the results we define the following sets. For $r, s > 0$, let $\hat{\mathcal{C}}_r \triangleq \{\boldsymbol{x} \in \mathbb{R}^n | \boldsymbol{Cx} = \boldsymbol{d}, \varphi(\boldsymbol{x}) \leq r\boldsymbol{e}\}$, and similarly let $\tilde{\mathcal{C}}_s \triangleq \{\boldsymbol{x} \in \mathbb{R}^n | \|\boldsymbol{Cx} - \boldsymbol{d}\| \leq s, \varphi(\boldsymbol{x}) \leq \boldsymbol{0}\}$. We have the following.

**Theorem 2** (Last and average iterate convergence for star-$\xi$-monotone operator). *Given an operator $F : \mathcal{X} \to \mathbb{R}^n$ monotone on $\mathcal{C}_=$ (Def. 1), assume that either $F$ is strictly monotone on $\mathcal{C}$ or one of $\varphi_i$ is strictly convex. Assume there exists $r > 0$ or $s > 0$ such that $F$ is star-$\xi$-monotone on either $\hat{\mathcal{C}}_r$ or $\tilde{\mathcal{C}}_s$. Let $\boldsymbol{x}_K^{(t)}$ and $\hat{\boldsymbol{x}}_K^{(t)} \triangleq \frac{1}{K}\sum_{k=1}^K \boldsymbol{x}_k^{(t)}$ denote the last and average iterate of Algorithm 1, respectively, run with sufficiently small $\mu_{-1}$. Then for all $t \in [T]$, we have that:*

1. $\left\|\boldsymbol{x}_K^{(t)} - \boldsymbol{x}^\star\right\| \le \mathcal{O}(\frac{1}{K^{1/(2\xi)}})$.

2. If in addition $F$ is $\xi$-monotone on $C_=$, we have $\left\|\hat{\boldsymbol{x}}_K^{(t)} - \boldsymbol{x}^\star\right\| \le \mathcal{O}(\frac{1}{K^{1/\xi}})$.

3. Moreover, if $F$ is $L$-Lipschitz on $C_=$—as per Assumption 1—the same corresponding upper bounds hold for $\mathcal{G}(\boldsymbol{x}_K^{(t)}, \mathcal{C})$ and $\mathcal{G}(\hat{\boldsymbol{x}}_K^{(t)}, \mathcal{C})$; that is, $\mathcal{G}(\boldsymbol{x}_K^{(t)}, \mathcal{C}) \le \mathcal{O}(\frac{L}{K^{1/(2\xi)}})$ and $\mathcal{G}(\hat{\boldsymbol{x}}_K^{(t)}, \mathcal{C}) \le \mathcal{O}(\frac{L}{K^{1/\xi}})$.

**Remark 2.** *Note that the convergence guarantee does not rely on Assumption 1, and it is solely used to relate the rate to the gap function. Also, note that $\mu_{-1}$ does not impact the convergence rate. Moreover, for simplicity we state the result with sufficiently small $\mu_{-1}$, however, the proof extends to any $\mu_{-1} > 0$. That is, the above result can be made parameter-free; see App. B.4.*

**Theorem 3** (Last and average iterate convergence for monotone operator). *Given an operator $F : \mathcal{X} \to \mathbb{R}^n$, assume (i) $F$ is monotone on $C_=$, and (ii) either $F$ is strictly monotone on $\mathcal{C}$ or one of $\varphi_i$ is strictly convex, and (iii) $\inf\limits_{\boldsymbol{x} \in S \setminus \{\boldsymbol{x}^\star\}} F(\boldsymbol{x})^\intercal \frac{\boldsymbol{x} - \boldsymbol{x}^\star}{\|\boldsymbol{x} - \boldsymbol{x}^\star\|} > 0$, where $S \equiv \hat{\mathcal{C}}_r$ or $\tilde{\mathcal{C}}_s$. Let $\boldsymbol{x}_K^{(t)}$ and $\hat{\boldsymbol{x}}_K^{(t)} \triangleq \frac{1}{K} \sum_{k=1}^K \boldsymbol{x}_k^{(t)}$ denote the last and average iterate of Algorithm 1, respectively, run with sufficiently small $\mu_{-1}$. Then for all $t \in [T]$, we have that:*

1. $\left\|\boldsymbol{x}_K^{(t)} - \boldsymbol{x}^\star\right\| \le \mathcal{O}(\frac{1}{\sqrt{K}})$.

2. If in addition $\inf\limits_{\boldsymbol{x} \in S \setminus \{\boldsymbol{x}^\star\}} F(\boldsymbol{x}^\star)^\intercal \frac{\boldsymbol{x} - \boldsymbol{x}^\star}{\|\boldsymbol{x} - \boldsymbol{x}^\star\|} > 0$ (with $S \equiv \hat{\mathcal{C}}_r$ or $\tilde{\mathcal{C}}_s$), then $\left\|\hat{\boldsymbol{x}}_K^{(t)} - \boldsymbol{x}^\star\right\| \le \mathcal{O}(\frac{1}{K})$.

3. Moreover, if $F$ is $L$-Lipschitz on $C_=$—as per Assumption 1—the same corresponding upper bounds hold for $\mathcal{G}(\boldsymbol{x}_K^{(t)}, \mathcal{C})$ and $\mathcal{G}(\hat{\boldsymbol{x}}_K^{(t)}, \mathcal{C})$, that is, $\mathcal{G}(\boldsymbol{x}_K^{(t)}, \mathcal{C}) \le \mathcal{O}(\frac{L}{\sqrt{K}})$ and $\mathcal{G}(\hat{\boldsymbol{x}}_K^{(t)}, \mathcal{C}) \le \mathcal{O}(\frac{L}{K})$.

Assumption (iii) in Theorem 3 requires the angle of $F(\boldsymbol{x})$ and $\boldsymbol{x} - \boldsymbol{x}^\star$ to be acute on $\mathcal{S} \setminus \{\boldsymbol{x}^\star\}$, where $\mathcal{S} = \hat{\mathcal{C}}_r$ or $\tilde{\mathcal{C}}_s$. For example, when there are no equality constraints, Assumption (iii) becomes $\inf\limits_{\boldsymbol{x} \in \mathcal{C} \setminus \{\boldsymbol{x}^\star\}} F(\boldsymbol{x})^\intercal \frac{\boldsymbol{x} - \boldsymbol{x}^\star}{\|\boldsymbol{x} - \boldsymbol{x}^\star\|} > 0$. From (cVI) and by the monotonicity of $F$, we can see that for any point $x \in \mathcal{C} \setminus \{\boldsymbol{x}^\star\}$, the angle between $F(\boldsymbol{x})$ and $\boldsymbol{x} - \boldsymbol{x}^\star$ is always less than or equal to $\pi/2$. And assumption (iii) requires that $F(\boldsymbol{x}^\star) \ne \boldsymbol{0}$, which means some constraints are active at $\boldsymbol{x}^\star$, and $\exists \theta \in (0, \pi/2)$ s.t. for any $\boldsymbol{x} \in \mathcal{C} \setminus \{\boldsymbol{x}^\star\}$, the angle between $F(\boldsymbol{x})$ and $\boldsymbol{x} - \boldsymbol{x}^\star$ is upper bounded by $\theta$.

**Remark 3.** *Our proofs rely on the existence of the central path—see Appendix A. Note that since $\mathcal{C}$ is compact, it suffices that either: (i) $F$ is strictly monotone on $\mathcal{C}$, or that (ii) one of the inequality constraints $\varphi_i$ is strictly convex for the central path to exist (Facchinei & Pang, 2003, Corollary 11.4.24). Thus, if $F$ is $\xi$-monotone on $\mathcal{C}$, then the central path exists. However, to relax the former assumption, notice that—by the compactness of $\mathcal{C}$—there exists a sufficiently large $M$ such that for any $\boldsymbol{x} \in \mathcal{C}$, $\boldsymbol{x}^\intercal \boldsymbol{x} \le M$. Thus, one can add a strictly convex inequality constraint $\varphi_{m+1}(\boldsymbol{x})$—e.g., $\boldsymbol{x}^\intercal \boldsymbol{x} - M \le 0$—and the solution set remains intact. That is, as $\mu$ tends to 0 the original problem is recovered. This ensures the existence of the central path without changing the original problem.*

## 5 EXPERIMENTS

**Problems.** To study the empirical performance of ACVI we use the following 2D problems: (i) *cBG*: the common bilinear game, constrained on $\mathbb{R}_+$ for the two players, stated in Fig. 1, (ii) *Von Neumann's ratio game* (Von Neumann, 1971; Daskalakis et al., 2020; Diakonikolas et al., 2021), (iii) *Forsaken* game (Hsieh et al., 2021)–which exhibits *limit cycles*, as well as (iv) *toy GAN*—used in (Daskalakis et al., 2018; Antonakopoulos et al., 2021). Note that these are known to be challenging problems in the literature, and interestingly the latter three are non-monotone, going beyond the assumptions that we made in our theoretical results. We also consider the following higher-dimension bilinear game on the probability simplex, with $\eta \in (0, 1)$, $n = 1000$:

$$\min_{\boldsymbol{x}_1 \in \triangle} \max_{\boldsymbol{x}_2 \in \triangle} \eta \boldsymbol{x}_1^\intercal \boldsymbol{x}_1 + (1 - \eta) \boldsymbol{x}_1^\intercal \boldsymbol{x}_2 - \eta \boldsymbol{x}_2^\intercal \boldsymbol{x}_2 ; \quad \triangle = \{\boldsymbol{x}_i \in \mathbb{R}^{500} | \boldsymbol{x}_i \ge \boldsymbol{0}, \text{ and }, \boldsymbol{e}^\intercal \boldsymbol{x}_i = 1\}. \quad \text{(HBG)}$$

As GANs on MNIST (Lecun & Cortes, 1998) enjoy well-established metrics, we use this setup and augment it solely with linear inequalities. We implement the baselines with the greedy projection algorithm—see App. D.3 for details—hence these baselines will be slower when equality constraints are also given. App. E lists additional experiments, including on Fashion-MNIST (Xiao et al., 2017).

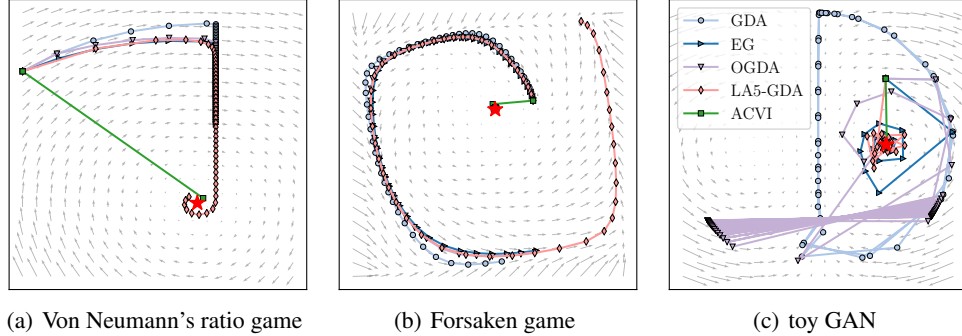

(a) Von Neumann's ratio game     (b) Forsaken game     (c) toy GAN

Figure 2: **Convergence of GDA, EG, OGDA, LA-GDA, and ACVI on three different 2d problems**, for a *fixed* number of *total* iterations, where markers denote the *iterates* of the respective method.

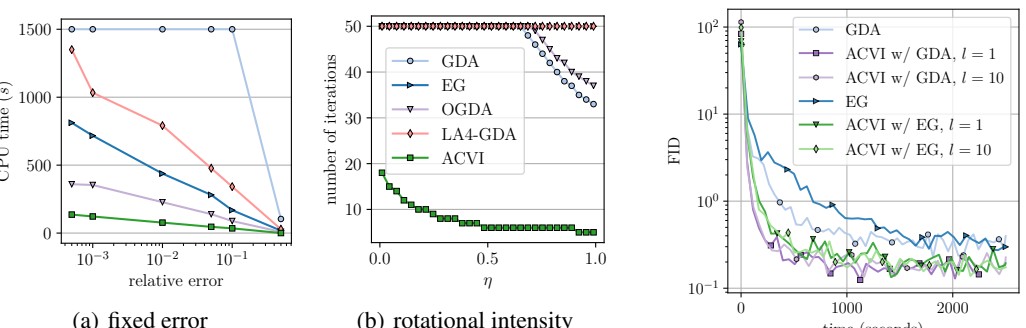

(a) fixed error        (b) rotational intensity

Figure 3: **Comparison on the** (HBG) **problem:** (a)–CPU time given fixed error, (b)–number of iterations needed to reach $\epsilon$-distance to solution for varying intensity of the rotational component $(1 - \eta)$. For both plots, we set maximum iterations/time to run. See § 5 for discussion.

Figure 4: FID (lower is better) on **MNIST** with added constraints, over wall-clock time; averaged over 3 seeds. See § 5 and App. D for discussion and implementation, resp.

**Methods.** We compare ACVI with the projected variants of the common saddle point optimizers (fully described in App. A.4): (i) **GDA**, (ii) **EG** (Korpelevich, 1976), (iii) **OGDA** (Popov, 1980), and (iv) **LA$\tilde{k}$-GDA** (Chavdarova et al., 2021b; Zhang et al., 2019), where $\tilde{k}$ is the hyperparameter of LA. For ACVI on MNIST, $l$ denotes the number of steps to solve the subproblems; see Algorithm 4.

**Results.** From Fig. 1, we observe that projection-based algorithms may zigzag when hitting a constraint due to the rotational nature of $F$, behavior that ACVI avoids because it incorporates the constraints in its update rule; see Fig. 5 for the remaining baselines. Fig. 2 shows that even with problems that go beyond our theoretical assumptions, a single step of ACVI significantly reduces the distance to the solution. Moreover, from Fig. 2(b), we observe that ACVI escapes the limit cycles; see also Fig. 6. Fig. 3 shows results for (HBG), indicating that ACVI is time efficient, and that ACVI performs well relative to projection-based methods for varying rotational intensity $(1 - \eta)$. Fig. 4 summarizes the experiments on MNIST with linear inequality constraints; we observe that ACVI converges significantly faster than the corresponding baseline.

## 6 CONCLUSION

Motivated by the lack of a *first*-order method to solve constrained VI (cVI) problems with general constraints, we proposed a framework that combines *(i) interior-point* methods—needed to be able to handle general constraints—with *(ii)* the ADMM method—designed to deal with *separable* objectives. The combination yields *ACVI*—a first-order ADMM-based interior point method for cVIs. We proved convergence for two broad classes of problems and derived the corresponding convergence rates. Numerical experiments showed that while projection-based methods zigzag when hitting a constraint due to the rotational vector field, ACVI avoids this by incorporating the constraints in the update rule.

ACKNOWLEDGMENTS

TC thanks the support of the Swiss National Science Foundation (SNSF), grant P2ELP2_199740. The authors thank Matteo Pagliardini and Tianyi Lin for insightful discussions and feedback.

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

## A    BACKGROUND: ADDITIONAL DETAILS

This section lists additional background such as omitted definitions and a description of the used baseline methods.

### A.1    ADDITIONAL VI DEFINITIONS & EQUIVALENT FORMULATIONS

Seeing an operator $F : \mathcal{X} \to \mathbb{R}^n$ as the graph $GrF = \{(\boldsymbol{x}, \boldsymbol{y}) | \boldsymbol{x} \in \mathcal{X}, \boldsymbol{y} = F(\boldsymbol{x})\}$, its inverse $F^{-1}$ is defined as $GrF^{-1} = \{(\boldsymbol{y}, \boldsymbol{x}) | (\boldsymbol{x}, \boldsymbol{y}) \in GrF\}$. See, for example, (Ryu & Yin, 2022) for further discussion. We denote the projection to the set $\mathcal{X}$ with $\Pi_{\mathcal{X}}$.

**Definition 3** ($\frac{1}{\mu}$-cocoercive operator). *An operator $F : \mathcal{X} \supseteq \mathcal{S} \to \mathbb{R}^n$ is $\frac{1}{\mu}$-cocoercive (or $\frac{1}{\mu}$-inverse strongly monotone) on $\mathcal{S}$ if its inverse (graph) $F^{-1}$ is $\mu$-strongly monotone on $\mathcal{S}$, that is,*

$$\exists \mu > 0, \quad s.t. \quad \langle \boldsymbol{x} - \boldsymbol{x}', F(\boldsymbol{x}) - F(\boldsymbol{x}') \rangle \geq \mu \left\| F(\boldsymbol{x}) - F(\boldsymbol{x}') \right\|^2, \forall \boldsymbol{x}, \boldsymbol{x}' \in \mathcal{S}.$$

*It is star $\frac{1}{\mu}$-cocoercive if the above holds when setting $\boldsymbol{x}' \equiv \boldsymbol{x}^{\star}$ where $\boldsymbol{x}^{\star}$ denotes a solution, that is:*

$$\exists \mu > 0, \quad s.t. \quad \langle \boldsymbol{x} - \boldsymbol{x}^{\star}, F(\boldsymbol{x}) - F(\boldsymbol{x}^{\star}) \rangle \geq \mu \left\| F(\boldsymbol{x}) - F(\boldsymbol{x}^{\star}) \right\|^2, \forall \boldsymbol{x} \in \mathcal{S}, \boldsymbol{x}^{\star} \in \mathcal{S}_{\mathcal{X}, F}^{\star}.$$

Note from Def. 3 that cocoercivity is a strict subclass of monotone and L-Lipschitz operators, thus is it is a stronger assumption. See Chapter 4.2 of (Bauschke & Combettes, 2017) for further relations of cocoercivity with other properties of operators.

In the following, we will make use of the natural and normal mappings of an operator $F : \mathcal{X} \to \mathbb{R}^n$, where $\mathcal{X} \subset \mathbb{R}^n$. Following the notation of (Facchinei & Pang, 2003), the natural map $F_{\mathcal{X}}^{NAT} : \mathcal{X} \to \mathbb{R}^n$ is:

$$F_{\mathcal{X}}^{NAT} \triangleq \boldsymbol{x} - \Pi_{\mathcal{X}}\big(\boldsymbol{x} - F(\boldsymbol{x})\big), \qquad \forall \boldsymbol{x} \in \mathcal{X}, \tag{F-NAT}$$

whereas the normal map $F_{\mathcal{X}}^{NOR} : \mathbb{R}^n \to \mathbb{R}^n$ is:

$$F_{\mathcal{X}}^{NOR} \triangleq F\big(\Pi_{\mathcal{X}}(\boldsymbol{x})\big) + \boldsymbol{x} - \Pi_{\mathcal{X}}(\boldsymbol{x}), \qquad \forall \boldsymbol{x} \in \mathbb{R}^n. \tag{F-NOR}$$

Moreover, we have the following solution characterizations:

(i)  $\boldsymbol{x}^{\star} \in \mathcal{S}_{\mathcal{X}, F}^{\star}$    iff    $F_{\mathcal{X}}^{NAT}(\boldsymbol{x}^{\star}) = \boldsymbol{0}$, and

(ii)  $\boldsymbol{x}^{\star} \in \mathcal{S}_{\mathcal{X}, F}^{\star}$    iff    $\exists \boldsymbol{x}' \in R^n$ s.t. $\boldsymbol{x}^{\star} = \Pi_{\mathcal{X}}(\boldsymbol{x}')$ and $F_{\mathcal{X}}^{NOR}(\boldsymbol{x}') = \boldsymbol{0}$.

**Remark 4** ("rotational" component of the vector field). *The rotational trajectories in parameter space are induced by the fact that the eigenvalues of the Jacobian of the vector field $F$ (the second-order derivative matrix) belong to the complex set $\mathbb{C}$; that is $\lambda_i \in \mathbb{C}$. In contrast, when $F \equiv \nabla f$ the second-order derivative matrix known as the Hessian is always symmetric, and thus the eigenvalues are real.*

### A.2    EXISTENCE OF SOLUTION

We provide brief informal summary of some sufficient conditions for solution existence, that $\mathcal{S}_{\cdot, F}^{\star} \neq \emptyset$. See Chapter 2 of (Facchinei & Pang, 2003) for a full treatment of the topic.

The common underlying tool to establish that a solution to the VI($\mathcal{X}$, $F$) problem exists is using *topological degree*. The topological degree tool is *designed* so as to satisfy the so-called *homotopy invariance* axiom, which in turn allows for reducing a solution existence question of a complicated map to a simpler one (which is homotopy-invariant to the original one) for which we can more easily show that it has a solution (e.g., the identity map on a closed domain). It can be used (as one way) to prove the celebrated *Brouwer fixed-point theorem*, which states that any continuous map $\Phi : \mathcal{S} \to \mathcal{S}$, where $\mathcal{S}$ is a nonempty convex compact set, has a fixed point in $\mathcal{S}$.

We have the following sufficient condition, see (Cor. 2.2.5, Facchinei & Pang, 2003)

**Theorem 4** (sufficient condition for existence of the solution, Cor. 2.2.5, (Facchinei & Pang, 2003))**.** *If $\mathcal{X} \subseteq \mathbb{R}^n$ is compact and convex, and $F : \mathcal{X} \to \mathbb{R}^n$ is continuous, then the solution set is nonempty and compact.*

It can also be shown that when $\mathcal{X}$ is closed convex (and $F$ continuous), if one can find $\boldsymbol{x}' \in \mathcal{X}$ s.t. $\langle F(\boldsymbol{x}), \boldsymbol{x} - \boldsymbol{x}' \rangle \geq 0, \forall \boldsymbol{x} \in \mathcal{X}$, then $\mathcal{S}^\star_{\mathcal{X}, F} \neq \emptyset$. The same conclusion follows if one can show that $\exists \boldsymbol{x}' \in \mathcal{X}$ and the set $\{\boldsymbol{x} \in \mathcal{X} | \langle F(\boldsymbol{x}), \boldsymbol{x} - \boldsymbol{x}' \rangle < 0\}$ is bounded (possibly empty, see Prop. 2.2.3 in (Facchinei & Pang, 2003)).

Sufficient conditions can also be established via the natural and the normal map due to the above solution characterizations. In this case, we require that $F$ is continuous on an open set $\mathcal{S}$ and we are interested if $\mathcal{S}^\star_{\mathcal{X}, F} \neq \emptyset$, where $\mathcal{X}$ is assumed closed and convex and subset of $\mathcal{S}$, $\mathcal{X} \subseteq \mathcal{S}$. If one establishes that a solution exists for $F^{NAT}_{\mathcal{X}}$ on a bounded open set $\mathcal{U}$, and if $cl\,\mathcal{U} \subseteq \mathcal{S}$, then it follows that $\mathcal{S}^\star_{\mathcal{X}, F} \neq \emptyset$. A similar implication holds when we have such a guarantee for $F^{NOR}_{\mathcal{X}}$. See Theorem 2.2.1 of (Facchinei & Pang, 2003).

In summary, the solution existence guarantee follows from the boundness of some set which includes $\mathcal{X}$, the boundness of the set of potential solutions (if we can construct such set), or the compactness of $\mathcal{X}$ itself.

### A.3 EXISTENCE OF CENTRAL PATH

In this section, we discuss the results that establish guarantees of the existence of the central path. Let $L(\boldsymbol{x}, \boldsymbol{\lambda}, \boldsymbol{\nu}) \triangleq F(\boldsymbol{x}) + \nabla\varphi^{\mathsf{T}}(\boldsymbol{x})\boldsymbol{\lambda} + \boldsymbol{C}^{\mathsf{T}}\boldsymbol{\nu}$, $h(\boldsymbol{x}) = \boldsymbol{C}^{\mathsf{T}}\boldsymbol{x} - \boldsymbol{d}$. For $(\boldsymbol{\lambda}, \boldsymbol{w}, \boldsymbol{x}, \boldsymbol{\nu}) \in \mathbb{R}^{2m+n+p}$, let

$$G(\boldsymbol{\lambda}, \boldsymbol{w}, \boldsymbol{x}, \boldsymbol{\nu}) \triangleq \begin{pmatrix} \boldsymbol{w} \circ \boldsymbol{\lambda} \\ \boldsymbol{w} + \varphi(\boldsymbol{x}) \\ L(\boldsymbol{x}, \boldsymbol{\lambda}, \boldsymbol{\nu}) \\ h(\boldsymbol{x}) \end{pmatrix} \in \mathbb{R}^{2m+n+p},$$

and

$$H(\boldsymbol{\lambda}, \boldsymbol{w}, \boldsymbol{x}, \boldsymbol{\nu}) \triangleq \begin{pmatrix} \boldsymbol{w} + \varphi(\boldsymbol{x}) \\ L(\boldsymbol{x}, \boldsymbol{\lambda}, \boldsymbol{\nu}) \\ h(\boldsymbol{x}) \end{pmatrix} \in \mathbb{R}^{m+n+p}.$$

Let $H_{++} \triangleq H(\mathbb{R}^{2m}_{++} \times \mathbb{R}^n \times \mathbb{R}^p)$. By (Corollary 11.4.24, Facchinei & Pang, 2003) we have the following proposition.

**Proposition 1** (sufficient condition for the existence of the central path.)**.** *If $F$ is monotone, either $F$ is strictly monotone or one of $\varphi_i$ is strictly convex, and $\mathcal{C}$ is bounded. The following four statements hold for the functions $G$ and $H$:*

1. *$G$ maps $\mathbb{R}^{2m}_{++} \times \mathbb{R}^{n+p}$ homeomorphically onto $\mathbb{R}^m_{++} \times H_{++}$;*

2. *$\mathbb{R}^m_{++} \times H_{++} \subseteq G(\mathbb{R}^{2m}_+ \times \mathbb{R}^{n+p})$;*

3. *for every vector $\boldsymbol{a} \in \mathbb{R}^m_+$, the system*

   $$H(\boldsymbol{\lambda}, \boldsymbol{w}, \boldsymbol{x}, \boldsymbol{\nu}) = \boldsymbol{0}, \quad \boldsymbol{w} \circ \boldsymbol{\lambda} = \boldsymbol{a}$$

   *has a solution $(\boldsymbol{\lambda}, \boldsymbol{w}, \boldsymbol{x}, \boldsymbol{\nu}) \in \mathbb{R}^{2m}_+ \times \mathbb{R}^{n+p}$; and*

4. *the set $H_{++}$ is convex.*

### A.4 SADDLE-POINT OPTIMIZATION METHODS

In this section, we describe in detail the saddle point methods that we compare within the main part (in § 5). We denote the projection to the set $\mathcal{X}$ with $\Pi_{\mathcal{X}}$, and when the method is applied in the unconstrained setting $\Pi_{\mathcal{X}} \equiv \boldsymbol{I}$.

For an example of the associated vector field and its Jacobian, consider the following constrained zero-sum game:

$$\min_{\boldsymbol{x}_1 \in \mathcal{X}_1} \max_{\boldsymbol{x}_2 \in \mathcal{X}_2} f(\boldsymbol{x}_1, \boldsymbol{x}_2), \tag{ZS-G}$$

where $f : \mathcal{X}_1 \times \mathcal{X}_2 \to \mathbb{R}$ is smooth and convex in $\boldsymbol{x}_1$ and concave in $\boldsymbol{x}_2$. As in the main paper, we write $\boldsymbol{x} \triangleq (\boldsymbol{x}_1, \boldsymbol{x}_2) \in \mathbb{R}^n$. The vector field $F : \mathcal{X} \to \mathbb{R}^n$ and its Jacobian $J$ are defined as:

$$F(\boldsymbol{x}) = \begin{bmatrix} \nabla_{\boldsymbol{x}_1} f(\boldsymbol{x}) \\ -\nabla_{\boldsymbol{x}_2} f(\boldsymbol{x}) \end{bmatrix}, \qquad J(\boldsymbol{x}) = \begin{bmatrix} \nabla^2_{\boldsymbol{x}_1} f(\boldsymbol{x}) & \nabla_{\boldsymbol{x}_2} \nabla_{\boldsymbol{x}_1} f(\boldsymbol{x}) \\ -\nabla_{\boldsymbol{x}_1} \nabla_{\boldsymbol{x}_2} f(\boldsymbol{x}) & -\nabla^2_{\boldsymbol{x}_2} f(\boldsymbol{x}) \end{bmatrix}.$$

In the remainder of this section, we will only refer to the joint variable $\boldsymbol{x}$, and (with abuse of notation) the subscript will denote the step. Let $\gamma \in [0, 1]$ denote the step size.

**(Projected) Gradient Descent Ascent (GDA).** The extension of gradient descent for the cVI problem is *gradient descent ascent* (GDA). The GDA update at step $k$ is then:

$$\boldsymbol{x}_{k+1} = \Pi_{\mathcal{X}} \big( \boldsymbol{x}_k - \gamma F(\boldsymbol{x}_k) \big). \tag{GDA}$$

**(Projected) Extragradient (EG).** EG (Korpelevich, 1976) uses a "prediction" step to obtain an extrapolated point $\boldsymbol{x}_{k+\frac{1}{2}}$ using GDA: $\boldsymbol{x}_{k+\frac{1}{2}} = \Pi_{\mathcal{X}} \big( \boldsymbol{x}_k - \gamma F(\boldsymbol{x}_k) \big)$, and the gradients at the *extrapolated* point are then applied to the *current* iterate $\boldsymbol{x}_t$:

$$\boldsymbol{x}_{k+1} = \Pi_{\mathcal{X}} \left( \boldsymbol{x}_k - \gamma F \Big( \Pi_{\mathcal{X}} \big( \boldsymbol{x}_k - \gamma F(\boldsymbol{x}_k) \big) \Big) \right). \tag{EG}$$

In the original EG paper, (Korpelevich, 1976) proved that the EG method (with a fixed step size) converges for monotone VIs, as follows.

**Theorem 5** (Korpelevich (1976))**.** *Given a map $F : \mathcal{X} \mapsto \mathbb{R}^n$, if the following is satisfied:*

1. *the set $\mathcal{X}$ is closed and convex,*

2. *$F$ is single-valued, definite, and monotone on $\mathcal{X}$–as per Def. 1,*

3. *$F$ is L-Lipschitz–as per Asm. 1.*

*then there exists a solution $\boldsymbol{x}^\star \in \mathcal{X}$, such that the iterates $\boldsymbol{x}_k$ produced by the EG update rule with a fixed step size $\gamma \in (0, \frac{1}{L})$ converge to it, that is $\boldsymbol{x}_k \to \boldsymbol{x}^\star$, as $k \to \infty$.*

Facchinei & Pang (2003) also show that for any *convex-concave* function $f$ and any closed convex sets $\boldsymbol{x}_1 \in \mathcal{X}_1$ and $\boldsymbol{x}_2 \in \mathcal{X}_2$, the EG method converges (Facchinei & Pang, 2003, Theorem 12.1.11).

**(Projected) Optimistic Gradient Descent Ascent (OGDA).** The update rule of Optimistic Gradient Descent Ascent OGDA ((OGDA) Popov, 1980) is:

$$\boldsymbol{x}_{n+1} = \Pi_{\mathcal{X}} \big( \boldsymbol{x}_n - 2\gamma F(\boldsymbol{x}_n) + \gamma F(\boldsymbol{x}_{n-1}) \big). \tag{OGDA}$$

**(Projected) Lookahead–Minmax (LA).** The LA algorithm for min-max optimization (Chavdarova et al., 2021b), originally proposed for minimization by Zhang et al. (2019), is a general wrapper of a "base" optimizer where, at every step $t$: (i) a copy of the current iterate $\tilde{\boldsymbol{x}}_n$ is made: $\tilde{\boldsymbol{x}}_n \leftarrow \boldsymbol{x}_n$, (ii) $\tilde{\boldsymbol{x}}_n$ is updated $k \geq 1$ times, yielding $\tilde{\boldsymbol{\omega}}_{n+k}$, and finally (iii) the actual update $\boldsymbol{x}_{n+1}$ is obtained as a *point that lies on a line between* the current $\boldsymbol{x}_n$ iterate and the predicted one $\tilde{\boldsymbol{x}}_{n+k}$:

$$\boldsymbol{x}_{n+1} \leftarrow \boldsymbol{x}_n + \alpha(\tilde{\boldsymbol{x}}_{n+k} - \boldsymbol{x}_n), \quad \alpha \in [0, 1]. \tag{LA}$$

In this work, we use solely GDA as a base optimizer for LA and thus denote it with *LAk-GDA*.

**The projection-free Frank-Wolfe (FW).** FW (Frank & Wolfe, 1956) is an IP-type method for solving constrained smooth zero-sum games (ZS-G). It avoids the projection operator by ensuring we never leave the constraint set. To do so, it finds the intersection points of the inequality constraints—hence, it requires that the inequality constraints satisfy certain structures (such as linear) in order for this operation to be computationally cheap. We state FW for zero-sum games in Algorithm 2 as proposed by Gidel et al. (2017a, Alg. 2) for completeness. It requires access to a linear minimization oracle (LMO) over the constraint set—to minimize the linear function in line 5 in Algorithm 2. Currently, FW-style algorithms only have convergence guarantees when the constraint set is a polytope ((Gidel et al., 2017a)) with additional assumptions (which we listed in § 2).

---

**Algorithm 2** Frank-Wolfe algorithm for zero-sum games.

1: **Input:** $C, \boldsymbol{\nu} > 0$
2: **Initialize:** $\boldsymbol{z}^{(0)} = (\boldsymbol{x}_1^{(0)}, \boldsymbol{x}_2^{(0)}) \in \mathcal{X}_1 \times \mathcal{X}_2$
3: **for** $t = 0, \dots, T$ **do**
4:     Compute $\boldsymbol{r}^{(t)} \triangleq \begin{bmatrix} \nabla_{\boldsymbol{x}_1} f(\boldsymbol{x}_1^{(t)}, \boldsymbol{x}_2^{(t)}) \\ -\nabla_{\boldsymbol{x}_2} f(\boldsymbol{x}_1^{(t)}, \boldsymbol{x}_2^{(t)}) \end{bmatrix}$
5:     $\boldsymbol{s}^{(t)} \triangleq \arg\min_{\boldsymbol{z} \in \mathcal{X}_1 \times \mathcal{X}_2} \langle \boldsymbol{z}, \boldsymbol{r}^{(t)} \rangle$
6:     Compute $g_t \triangleq \langle \boldsymbol{z}^{(t)} - \boldsymbol{s}^{(t)}, \boldsymbol{r}^{(t)} \rangle$
7:     **if** $g_t \leq \varepsilon$ **then**
8:         **return** $\boldsymbol{z}^{(t)}$
9:     **end if**
10:     Let $\gamma = \min\left(1, \frac{\boldsymbol{\nu}}{2C} g_t\right)$ or $\gamma = \frac{2}{2+t}$
11:     Update $\boldsymbol{z}^{(t+1)} = (1 - \gamma)\boldsymbol{z}^{(t)} + \gamma \boldsymbol{s}^{(t)}$
12: **end for**

---

## B  OMITTED PROOFS AND DISCUSSIONS CONCERNING ALGORITHM 1

This section provides the proofs of the theoretical results in § 4.2.

### B.1  PROOF OF THEOREM 1: UNIQUENESS OF THE SOLUTION OF EQ. W-EQ

Recall first that Eq. W-EQ is as follows:

$$\boldsymbol{x} + \frac{1}{\beta}\boldsymbol{P}_c F(\boldsymbol{x}) - \boldsymbol{P}_c \boldsymbol{y}_k + \frac{1}{\beta}\boldsymbol{P}_c \boldsymbol{\lambda}_k - \boldsymbol{d}_c = \boldsymbol{0},$$

since $\boldsymbol{w} = \boldsymbol{x}$.

*Proof of Theorem 1: uniqueness of the solution of* (W-EQ). Let $G(\boldsymbol{x})$ denote the LHS of (W-EQ), that is:

$$G(\boldsymbol{x}) \triangleq \boldsymbol{x} + \frac{1}{\beta}\boldsymbol{P}_c F(\boldsymbol{x}) - \boldsymbol{P}_c \boldsymbol{y}_k + \frac{1}{\beta}\boldsymbol{P}_c \boldsymbol{\lambda}_k - \boldsymbol{d}_c \tag{3}$$

We claim that $G(\boldsymbol{x})$ is strongly monotone on $\mathcal{C}_=$. In fact, $\forall \boldsymbol{x}, \boldsymbol{y} \in \mathcal{C}_=$, $\boldsymbol{P}_c(\boldsymbol{x} - \boldsymbol{y}) = \boldsymbol{x} - \boldsymbol{y}$. Note that $\boldsymbol{P}_c$ is symmetric, thus we have:

$$\langle G(\boldsymbol{x}) - G(\boldsymbol{y}), \boldsymbol{x} - \boldsymbol{y}\rangle = \|\boldsymbol{x} - \boldsymbol{y}\|^2 + \frac{1}{\beta}\langle \boldsymbol{P}_c F(\boldsymbol{x}) - \boldsymbol{P}_c F(\boldsymbol{y}), \boldsymbol{x} - \boldsymbol{y}\rangle$$

$$= \|\boldsymbol{x} - \boldsymbol{y}\|^2 + \frac{1}{\beta}\langle \boldsymbol{x} - \boldsymbol{y}, F(\boldsymbol{x}) - F(\boldsymbol{y})\rangle$$

$$\geq \|\boldsymbol{x} - \boldsymbol{y}\|^2.$$

Therefore, according to Theorem 2.3.3 (b) in (Facchinei & Pang, 2003), $\mathcal{S}^{\star}_{\mathcal{C}_=,G}$ has a unique solution $\tilde{\boldsymbol{x}} \in \mathcal{C}_=$. Thus, we have:

$$G(\tilde{\boldsymbol{x}})^{\mathsf{T}}(\boldsymbol{x} - \tilde{\boldsymbol{x}}) = 0, \ \forall \boldsymbol{x} \in \mathcal{C}_= \,.$$

From the above, we deduce that $G(\boldsymbol{x}) \in Span\{\boldsymbol{c}_1, \cdots, \boldsymbol{c}_p\}$, where $\boldsymbol{c}_i$ is the row vectors of $\boldsymbol{C}$, $i \in [p]$.

Suppose that $G(\tilde{\boldsymbol{x}}) = \sum_{i=1}^p \alpha_i \boldsymbol{c}_i$. Notice that $\boldsymbol{C}G(\boldsymbol{x}) = \boldsymbol{0}, \forall \boldsymbol{x} \in \mathcal{C}_=$. Thus, we have that:

$$\boldsymbol{c}_j^{\mathsf{T}} G(\tilde{\boldsymbol{x}}) = \boldsymbol{c}_j^{\mathsf{T}} \sum_{i=1}^p \alpha_i \boldsymbol{c}_i = 0, \ \forall j \in [p] \,.$$

Hence,

$$\langle G(\tilde{\boldsymbol{x}}), G(\tilde{\boldsymbol{x}})\rangle = \langle \sum_{i=1}^p \alpha_i \boldsymbol{c}_i, \sum_{i=1}^p \alpha_i \boldsymbol{c}_i \rangle = 0 \,,$$

which indicates that $G(\tilde{\boldsymbol{x}}) = \boldsymbol{0}$. Hence, $\tilde{\boldsymbol{x}}$ is a solution of (W-EQ) in $\mathcal{C}_=$.

On the other hand, $\forall x \in \mathbb{R}^n$, if $\boldsymbol{x}$ is a solution of (W-EQ), i.e. $G(\boldsymbol{x}) = \boldsymbol{0}$, then $\boldsymbol{x} \in \mathcal{C}_=$. By the uniqueness of $\tilde{\boldsymbol{x}}$ in $\mathcal{C}_=$ we have that $\boldsymbol{x} = \tilde{\boldsymbol{x}}$, which means $\tilde{\boldsymbol{x}}$ is unique in $\mathbb{R}^n$. $\qquad\square$

### B.2  PROOF OF LEMMA 1: UPPER BOUND ON THE GAP FUNCTION

*Proof of Lemma 1: Upper bound on the gap function.* Let $\boldsymbol{x}_k$ denote an iterate produced by Algorithm 1, and let $\boldsymbol{x} \in \mathcal{C}$. Note that we always have $\boldsymbol{x}_k \in \mathcal{C}_=$. We have that:

$$\langle F(\boldsymbol{x}_k), \boldsymbol{x}_k - \boldsymbol{x}\rangle = \langle F(\boldsymbol{x}^{\star}), \boldsymbol{x}^{\star} - \boldsymbol{x}\rangle + \langle F(\boldsymbol{x}_k), \boldsymbol{x}_k - \boldsymbol{x}\rangle - \langle F(\boldsymbol{x}^{\star}), \boldsymbol{x}^{\star} - \boldsymbol{x}\rangle$$

$$= \langle F(\boldsymbol{x}^{\star}), \boldsymbol{x}^{\star} - \boldsymbol{x}\rangle + \langle F(\boldsymbol{x}_k), \boldsymbol{x}_k - \boldsymbol{x}^{\star} + \boldsymbol{x}^{\star} - \boldsymbol{x}\rangle - \langle F(\boldsymbol{x}^{\star}), \boldsymbol{x}^{\star} - \boldsymbol{x}\rangle \,.$$

From the proof of Theorem 6 we know that $\boldsymbol{x}_k$ is bounded, which gives: $\langle F(\boldsymbol{x}_k), \boldsymbol{x}_k - \boldsymbol{x}^{\star}\rangle \leq M\|\boldsymbol{x}_k - \boldsymbol{x}^{\star}\|$, with $M > 0$, as well as that $\langle F(\boldsymbol{x}^{\star}), \boldsymbol{x}^{\star} - \boldsymbol{x}\rangle \leq 0$. Thus, for the above we get:

$$\langle F(\boldsymbol{x}_k), \boldsymbol{x}_k - \boldsymbol{x}\rangle \leq \langle F(\boldsymbol{x}_k) - F(\boldsymbol{x}^{\star}), \boldsymbol{x}^{\star} - \boldsymbol{x}\rangle + M\|\boldsymbol{x}_k - \boldsymbol{x}^{\star}\|$$

$$\leq D\|F(\boldsymbol{x}_k) - F(\boldsymbol{x}^{\star})\| + M\|\boldsymbol{x}_k - \boldsymbol{x}^{\star}\|$$

$$\leq (DL + M)\|\boldsymbol{x}_k - \boldsymbol{x}^{\star}\| \,,$$

where for the second row we used $\boldsymbol{x}^\star - \boldsymbol{x} \leq D$ where $D \triangleq \max_{\boldsymbol{x}' \in \mathcal{C}} \|\boldsymbol{x}^\star - \boldsymbol{x}'\|$ is the largest distance between any point in $\mathcal{C}$ and $\boldsymbol{x}^\star$. For the last row we used that $F$ is L-Lipschitz—Assumption 1—which concludes the proof. $\square$

The proof is analogous for the $\boldsymbol{y}_k \in \mathcal{C}_\leq$ iterates produced by Algorithm 1.

### B.3 PROOFS OF THE CONVERGENCE RATE: THEOREMS 2 AND 3

Let
$$f^{(t)}(\boldsymbol{x}) \triangleq F(\boldsymbol{x}^{\mu_t})^\mathsf{T}\boldsymbol{x} + \mathbb{1}(\boldsymbol{C}\boldsymbol{x} = \boldsymbol{d}),$$
$$f_k^{(t)}(\boldsymbol{x}) \triangleq F(\boldsymbol{x}_{k+1}^{(t)})^\mathsf{T}\boldsymbol{x} + \mathbb{1}(\boldsymbol{C}\boldsymbol{x} = \boldsymbol{d}),$$
$$g^{(t)}(\boldsymbol{y}) \triangleq -\mu_t \sum_{i=1}^m \log\left(-\varphi_i(\boldsymbol{y})\right),$$

where $\boldsymbol{x}^{\mu_t}$ is a solution of (KKT-2) when $\mu = \mu_t$. Note that the existence of $\boldsymbol{x}^{\mu_t}$ is guaranteed by the existence of the central path-see App. A, and that $f^{(t)}, f_k^{(t)}$ and $g^{(t)}$ are all convex.

In the following proofs, unless causing confusion, we drop the subscript $t$ to simplify notations.

Let $\boldsymbol{y}^\mu = \boldsymbol{x}^\mu$, from (KKT-2) we can see that $(\boldsymbol{x}^\mu, \boldsymbol{y}^\mu)$ is an optimal solution of

$$\begin{cases} \min f(\boldsymbol{x}) + g(\boldsymbol{y}) \\ \quad s.t. \quad \boldsymbol{x} = \boldsymbol{y} \end{cases}. \tag{4}$$

There exists $\boldsymbol{\lambda}^\mu \in \mathbb{R}^n$ such that $(\boldsymbol{x}^\mu, \boldsymbol{y}^\mu, \boldsymbol{\lambda}^\mu)$ is a KKT point of (4). We give the following proposition which we will repeatedly use in the proofs:

**Proposition 2.** *If $F$ is monotone, then $\forall k \in \mathbb{N}$,*

$$f_k(\boldsymbol{x}_{k+1}) - f_k(\boldsymbol{x}^\mu) \geq f(\boldsymbol{x}_{k+1}) - f(\boldsymbol{x}^\mu).$$

*Furthermore, if $F$ is $\xi$-monotone, as per Def. 1*

$$f_k(\boldsymbol{x}_{k+1}) - f_k(\boldsymbol{x}^\mu) \geq f(\boldsymbol{x}_{k+1}) - f(\boldsymbol{x}^\mu) + c\|\boldsymbol{x}_{k+1} - \boldsymbol{x}^\mu\|_2^\xi.$$

*Proof of Proposition 2.* It suffices to note that:

$$f_k(\boldsymbol{x}_{k+1}) - f_k(\boldsymbol{x}^\mu) - (f(\boldsymbol{x}_{k+1}) - f(\boldsymbol{x}^\mu)) = (F(\boldsymbol{x}_{k+1}) - F(\boldsymbol{x}^\mu))^\mathsf{T}(\boldsymbol{x}_{k+1} - \boldsymbol{x}^\mu).$$

$\square$

Some of our proofs that follow are inspired by some convergence proofs in ADMM (Gabay, 1983; Eckstein & Bertsekas, 1992; Davis & Yin, 2016; He & Yuan, 2012; 2015; Lin et al., 2022). However, although Algorithm 1 adopts the high level idea of ADMM, we can not directly refer to the convergence proofs of ADMM, but need to substantially modify these.

We will use the following lemma.

**Lemma 2.** $f(\boldsymbol{x}) + g(\boldsymbol{y}) - f(\boldsymbol{x}^\mu) - g(\boldsymbol{y}^\mu) + \langle \boldsymbol{\lambda}^\mu, \boldsymbol{x} - \boldsymbol{y} \rangle \geq 0, \forall \boldsymbol{x}, \boldsymbol{y}.$

*Proof.* The Lagrange function of (4) is

$$L(\boldsymbol{x}, \boldsymbol{y}, \boldsymbol{\lambda}) = f(\boldsymbol{x}) + g(\boldsymbol{y}) + \boldsymbol{\lambda}^\mathsf{T}(\boldsymbol{x} - \boldsymbol{y}).$$

And by the property of KKT point, we have

$$L(\boldsymbol{x}^\mu, \boldsymbol{y}^\mu, \boldsymbol{\lambda}) \leq L(\boldsymbol{x}^\mu, \boldsymbol{y}^\mu, \boldsymbol{\lambda}^\mu) \leq L(\boldsymbol{x}, \boldsymbol{y}, \boldsymbol{\lambda}^\mu), \ \forall(\boldsymbol{x}, \boldsymbol{y}, \boldsymbol{\lambda}),$$

from which the conclusion follows. $\square$

The following lemma is straightforward to verify:

**Lemma 3.** *If*

$$f(\boldsymbol{x}) + g(\boldsymbol{y}) - f(\boldsymbol{x}^{\mu}) - g(\boldsymbol{y}^{\mu}) + \langle \boldsymbol{\lambda}^{\mu}, \boldsymbol{x} - \boldsymbol{y} \rangle \leq \alpha_1,$$
$$\|\boldsymbol{x} - \boldsymbol{y}\| \leq \alpha_2$$

*then we have*

$$-\|\boldsymbol{\lambda}^{\mu}\|\alpha_2 \leq f(\boldsymbol{x}) + g(\boldsymbol{y}) - f(\boldsymbol{x}^{\mu}) - g(\boldsymbol{y}^{\mu}) \leq \|\boldsymbol{\lambda}^{\mu}\|\alpha_2 + \alpha_1.$$

The following lemma lists some simple but useful facts that we will use in the following proofs.

**Lemma 4.** *For* (4) *and Algorithm* 1*, we have*

$$\boldsymbol{0} \in \partial f_k(\boldsymbol{x}_{k+1}) + \boldsymbol{\lambda}_k + \beta(\boldsymbol{x}_{k+1} - \boldsymbol{y}_{k+1}) \tag{5}$$
$$\boldsymbol{0} \in \partial g(\boldsymbol{y}_{k+1}) - \boldsymbol{\lambda}_k - \beta(\boldsymbol{x}_{k+1} - \boldsymbol{y}_{k+1}), \tag{6}$$
$$\boldsymbol{\lambda}_{k+1} - \boldsymbol{\lambda}_k = \beta(\boldsymbol{x}_{k+1} - \boldsymbol{y}_{k+1}), \tag{7}$$
$$\boldsymbol{0} \in \partial f(\boldsymbol{x}^{\mu}) + \boldsymbol{\lambda}^{\mu}, \tag{8}$$
$$\boldsymbol{0} \in \partial g(\boldsymbol{y}^{\mu}) - \boldsymbol{\lambda}^{\mu}, \tag{9}$$
$$\boldsymbol{x}^{\mu} = \boldsymbol{y}^{\mu}. \tag{10}$$

We define:

$$\hat{\nabla} f_k(\boldsymbol{x}_{k+1}) \triangleq -\boldsymbol{\lambda}_k - \beta(\boldsymbol{x}_{k+1} - \boldsymbol{y}_k), \tag{11}$$
$$\hat{\nabla} g(\boldsymbol{y}_{k+1}) \triangleq \boldsymbol{\lambda}_k + \beta(\boldsymbol{x}_{k+1} - \boldsymbol{y}_{k+1}). \tag{12}$$

Then from (5) and (6) we can see that

$$\hat{\nabla} f_k(\boldsymbol{x}_{k+1}) \in \partial f_k(\boldsymbol{x}_{k+1}) \text{ and } \hat{\nabla} g(\boldsymbol{y}_{k+1}) \in \partial g(\boldsymbol{y}_{k+1}). \tag{13}$$

**Lemma 5.** *For Algorithm* 1*, we have*

$$\langle \hat{\nabla} g(\boldsymbol{y}_{k+1}), \boldsymbol{y}_{k+1} - \boldsymbol{y} \rangle = -\langle \boldsymbol{\lambda}_{k+1}, \boldsymbol{y} - \boldsymbol{y}_{k+1} \rangle, \tag{14}$$

*and*

$$\langle \hat{\nabla} f_k(\boldsymbol{x}_{k+1}), \boldsymbol{x}_{k+1} - \boldsymbol{x} \rangle + \langle \hat{\nabla} g(\boldsymbol{y}_{k+1}), \boldsymbol{y}_{k+1} - \boldsymbol{y} \rangle$$
$$= -\langle \boldsymbol{\lambda}_{k+1}, \boldsymbol{x}_{k+1} - \boldsymbol{y}_{k+1} - \boldsymbol{x} + \boldsymbol{y} \rangle + \beta \langle -\boldsymbol{y}_{k+1} + \boldsymbol{y}_k, \boldsymbol{x}_{k+1} - \boldsymbol{x} \rangle. \tag{15}$$

*Proof of Lemma 5.* From (7), (11) and (12) we have:

$$\langle \hat{\nabla} f_k(\boldsymbol{x}_{k+1}), \boldsymbol{x}_{k+1} - \boldsymbol{x} \rangle$$
$$= -\langle \boldsymbol{\lambda}_k + \beta(\boldsymbol{x}_{k+1} - \boldsymbol{y}_k), \boldsymbol{x}_{k+1} - \boldsymbol{x} \rangle$$
$$= -\langle \boldsymbol{\lambda}_{k+1}, \boldsymbol{x}_{k+1} - \boldsymbol{x} \rangle + \beta \langle -\boldsymbol{y}_{k+1} + \boldsymbol{y}_k, \boldsymbol{x}_{k+1} - \boldsymbol{x} \rangle,$$

and

$$\langle \hat{\nabla} g(\boldsymbol{y}_{k+1}), \boldsymbol{y}_{k+1} - \boldsymbol{y} \rangle = -\langle \boldsymbol{\lambda}_{k+1}, \boldsymbol{y} - \boldsymbol{y}_{k+1} \rangle.$$

Adding these together yields (15). □

**Lemma 6.** *For Algorithm* 1*, we have*

$$\langle \hat{\nabla} f_k(\boldsymbol{x}_{k+1}), \boldsymbol{x}_{k+1} - \boldsymbol{x}^{\mu} \rangle + \langle \hat{\nabla} g(\boldsymbol{y}_{k+1}), \boldsymbol{y}_{k+1} - \boldsymbol{y}^{\mu} \rangle + \langle \boldsymbol{\lambda}^{\mu}, \boldsymbol{x}_{k+1} - \boldsymbol{y}_{k+1} \rangle$$
$$\leq \frac{1}{2\beta} \|\boldsymbol{\lambda}_k - \boldsymbol{\lambda}^{\mu}\|^2 - \frac{1}{2\beta} \|\boldsymbol{\lambda}_{k+1} - \boldsymbol{\lambda}^{\mu}\|^2$$
$$+ \frac{\beta}{2} \|\boldsymbol{y}^{\mu} - \boldsymbol{y}_k\|^2 - \frac{\beta}{2} \|\boldsymbol{y}^{\mu} - \boldsymbol{y}_{k+1}\|^2$$
$$- \frac{1}{2\beta} \|\boldsymbol{\lambda}_{k+1} - \boldsymbol{\lambda}_k\|^2 - \frac{\beta}{2} \|\boldsymbol{y}_k - \boldsymbol{y}_{k+1}\|^2$$

*Proof of Lemma 6.* Letting $(\boldsymbol{x}, \boldsymbol{y}, \boldsymbol{\lambda}) = (\boldsymbol{x}^{\mu}, \boldsymbol{y}^{\mu}, \boldsymbol{\lambda}^{\mu})$ in(15), adding $\langle \boldsymbol{\lambda}^{\mu}, \boldsymbol{x}_{k+1} - \boldsymbol{y}_{k+1} \rangle$ to both sides, and using (7) and (10), we have:

$$
\begin{aligned}
&\langle \hat{\nabla} f_k(\boldsymbol{x}_{k+1}), \boldsymbol{x}_{k+1} - \boldsymbol{x}^{\mu} \rangle + \langle \hat{\nabla} g(\boldsymbol{y}_{k+1}), \boldsymbol{y}_{k+1} - \boldsymbol{y}^{\mu} \rangle + \langle \boldsymbol{\lambda}^{\mu}, \boldsymbol{x}_{k+1} - \boldsymbol{y}_{k+1} \rangle \\
&= - \langle \boldsymbol{\lambda}_{k+1} - \boldsymbol{\lambda}^{\mu}, \boldsymbol{x}_{k+1} - \boldsymbol{y}_{k+1} \rangle + \beta \langle -\boldsymbol{y}_{k+1} + \boldsymbol{y}_k, \boldsymbol{x}_{k+1} - \boldsymbol{x}^{\mu} \rangle \\
&= - \frac{1}{\beta} \langle \boldsymbol{\lambda}_{k+1} - \boldsymbol{\lambda}^{\mu}, \boldsymbol{\lambda}_{k+1} - \boldsymbol{\lambda}_k \rangle + \langle -\boldsymbol{y}_{k+1} + \boldsymbol{y}_k, \boldsymbol{\lambda}_{k+1} - \boldsymbol{\lambda}_k \rangle \\
&\quad - \beta \langle -\boldsymbol{y}_{k+1} + \boldsymbol{y}_k, -\boldsymbol{y}_{k+1} + \boldsymbol{y}^{\mu} \rangle \\
&= \frac{1}{2\beta} \|\boldsymbol{\lambda}_k - \boldsymbol{\lambda}^{\mu}\|^2 - \frac{1}{2\beta} \|\boldsymbol{\lambda}_{k+1} - \boldsymbol{\lambda}^{\mu}\|^2 - \frac{1}{2\beta} \|\boldsymbol{\lambda}_{k+1} - \boldsymbol{\lambda}_k\|^2 \\
&\quad + \frac{\beta}{2} \|-\boldsymbol{y}_k + \boldsymbol{y}^{\star}\|^2 - \frac{\beta}{2} \|-\boldsymbol{y}_{k+1} + \boldsymbol{y}^{\star}\|^2 - \frac{\beta}{2} \|-\boldsymbol{y}_{k+1} + \boldsymbol{y}_k\|^2 \\
&\quad + \langle -\boldsymbol{y}_{k+1} + \boldsymbol{y}_k, \boldsymbol{\lambda}_{k+1} - \boldsymbol{\lambda}_k \rangle .
\end{aligned}
$$
(16)

(17)

On the other hand, (14) gives

$$
\langle \hat{\nabla} g(\boldsymbol{y}_k), \boldsymbol{y}_k - \boldsymbol{y} \rangle + \langle \boldsymbol{\lambda}_k, -\boldsymbol{y}_k + \boldsymbol{y} \rangle = 0 .
$$
(18)

Letting $\boldsymbol{y} = \boldsymbol{y}_k$ in (14) and $\boldsymbol{y} = \boldsymbol{y}_{k+1}$ in (18), and adding them together, we have:

$$
\langle \hat{\nabla} g(\boldsymbol{y}_{k+1}) - \hat{\nabla} g(\boldsymbol{y}_k), \boldsymbol{y}_{k+1} - \boldsymbol{y}_k \rangle + \langle \boldsymbol{\lambda}_{k+1} - \boldsymbol{\lambda}_k, -\boldsymbol{y}_{k+1} + \boldsymbol{y}_k \rangle = 0 .
$$

By the monotonicity of $\partial g$ we know that the first term of the above equality is non-negative. Thus, we have:

$$
\langle \boldsymbol{\lambda}_{k+1} - \boldsymbol{\lambda}_k, -\boldsymbol{y}_{k+1} + \boldsymbol{y}_k \rangle \le 0 .
$$
(19)

Plugging it into (17), we have the conclusion. $\qquad\square$

**Lemma 7.** *For Algorithm 1, we have*

$$
\begin{aligned}
&f(\boldsymbol{x}_{k+1}) + g(\boldsymbol{y}_{k+1}) - f(\boldsymbol{x}^{\mu}) - g(\boldsymbol{y}^{\mu}) + \langle \boldsymbol{\lambda}^{\mu}, \boldsymbol{x}_{k+1} - \boldsymbol{y}_{k+1} \rangle \\
&\le \frac{1}{2\beta} \|\boldsymbol{\lambda}_k - \boldsymbol{\lambda}^{\mu}\|^2 - \frac{1}{2\beta} \|\boldsymbol{\lambda}_{k+1} - \boldsymbol{\lambda}^{\mu}\|^2 \\
&\quad + \frac{\beta}{2} \|-\boldsymbol{y}_k + \boldsymbol{y}^{\mu}\|^2 - \frac{\beta}{2} \|-\boldsymbol{y}_{k+1} + \boldsymbol{y}^{\mu}\|^2 \\
&\quad - \frac{1}{2\beta} \|\boldsymbol{\lambda}_{k+1} - \boldsymbol{\lambda}_k\|^2 - \frac{\beta}{2} \|-\boldsymbol{y}_{k+1} + \boldsymbol{y}_k\|^2
\end{aligned}
$$
(20)

*Furthermore, if $F$ is $\xi$-monotone on $\mathcal{C}_=$, we have*

$$
\begin{aligned}
&c \|\boldsymbol{x}_{k+1} - \boldsymbol{x}^{\mu}\|_2^{\xi} + f(\boldsymbol{x}_{k+1}) + g(\boldsymbol{y}_{k+1}) - f(\boldsymbol{x}^{\mu}) - g(\boldsymbol{y}^{\mu}) + \langle \boldsymbol{\lambda}^{\mu}, \boldsymbol{x}_{k+1} - \boldsymbol{y}_{k+1} \rangle \\
&\le \frac{1}{2\beta} \|\boldsymbol{\lambda}_k - \boldsymbol{\lambda}^{\mu}\|^2 - \frac{1}{2\beta} \|\boldsymbol{\lambda}_{k+1} - \boldsymbol{\lambda}^{\mu}\|^2 \\
&\quad + \frac{\beta}{2} \|-\boldsymbol{y}_k + \boldsymbol{y}^{\mu}\|^2 - \frac{\beta}{2} \|-\boldsymbol{y}_{k+1} + \boldsymbol{y}^{\mu}\|^2 \\
&\quad - \frac{1}{2\beta} \|\boldsymbol{\lambda}_{k+1} - \boldsymbol{\lambda}_k\|^2 - \frac{\beta}{2} \|-\boldsymbol{y}_{k+1} + \boldsymbol{y}_k\|^2 .
\end{aligned}
$$
(21)

*Proof.* From the convexity of $f_k(\boldsymbol{x})$ and $g(\boldsymbol{y})$ and using Proposition 2 and (13), we have

$$
\begin{aligned}
&f(\boldsymbol{x}_{k+1}) + g(\boldsymbol{y}_{k+1}) - f(\boldsymbol{x}^\mu) - g(\boldsymbol{y}^\mu) + \langle \boldsymbol{\lambda}^\mu, \boldsymbol{x}_{k+1} - \boldsymbol{y}_{k+1} \rangle \\
\leq & f_k(\boldsymbol{x}_{k+1}) + g(\boldsymbol{y}_{k+1}) - f_k(\boldsymbol{x}^\mu) - g(\boldsymbol{y}^\mu) + \langle \boldsymbol{\lambda}^\mu, \boldsymbol{x}_{k+1} - \boldsymbol{y}_{k+1} \rangle \\
\leq & \langle \hat{\nabla} f_k(\boldsymbol{x}_{k+1}), \boldsymbol{x}_{k+1} - \boldsymbol{x}^\mu \rangle + \langle \hat{\nabla} g(\boldsymbol{y}_{k+1}), \boldsymbol{y}_{k+1} - \boldsymbol{y}^\mu \rangle \\
& + \langle \boldsymbol{\lambda}^\mu, \boldsymbol{x}_{k+1} - \boldsymbol{y}_{k+1} \rangle \\
\leq & \frac{1}{2\beta} \|\boldsymbol{\lambda}_k - \boldsymbol{\lambda}^\mu\|^2 - \frac{1}{2\beta} \|\boldsymbol{\lambda}_{k+1} - \boldsymbol{\lambda}^\mu\|^2 \\
& + \frac{\beta}{2} \|-\boldsymbol{y}_k + \boldsymbol{y}^\mu\|^2 - \frac{\beta}{2} \|-\boldsymbol{y}_{k+1} + \boldsymbol{y}^\mu\|^2 \\
& - \frac{1}{2\beta} \|\boldsymbol{\lambda}_{k+1} - \boldsymbol{\lambda}_k\|^2 - \frac{\beta}{2} \|-\boldsymbol{y}_{k+1} + \boldsymbol{y}_k\|^2.
\end{aligned}
\tag{22}
$$

If $F$ is $\xi$-monotone on $\mathcal{C}_=$, again by Proposition 2, we can add the term $c\|\boldsymbol{x}_{k+1} - \boldsymbol{x}^\mu\|_2^\xi$ to the first line and the inequality still holds. $\square$

**Theorem 6.** *For Algorithm 1, we have*

$$
\begin{aligned}
f(\boldsymbol{x}_{k+1}) - f(\boldsymbol{x}^\mu) + g(\boldsymbol{y}_{k+1}) - g(\boldsymbol{y}^\mu) &\to 0, \\
f_k(\boldsymbol{x}_{k+1}) - f_k(\boldsymbol{x}^\mu) + g(\boldsymbol{y}_{k+1}) - g(\boldsymbol{y}^\mu) &\to 0, \\
\boldsymbol{x}_{k+1} - \boldsymbol{y}_{k+1} &\to \boldsymbol{0},
\end{aligned}
$$

*as $k \to \infty$. Furthermore, if $F$ is $\xi$-monotone on $\mathcal{C}_=$, we have*

$$
\boldsymbol{x}_{k+1} \to \boldsymbol{x}^\mu, \ k \to \infty
$$

*Proof of Theorem 6.* Proof From Lemma 2 and (20), we have

$$
\begin{aligned}
& \frac{1}{2\beta} \|\boldsymbol{\lambda}_{k+1} - \boldsymbol{\lambda}_k\|^2 + \frac{\beta}{2} \|-\boldsymbol{y}_{k+1} + \boldsymbol{y}_k\|^2 \\
\leq & \frac{1}{2\beta} \|\boldsymbol{\lambda}_k - \boldsymbol{\lambda}^\mu\|^2 - \frac{1}{2\beta} \|\boldsymbol{\lambda}_{k+1} - \boldsymbol{\lambda}^\mu\|^2 \\
& + \frac{\beta}{2} \|-\boldsymbol{y}_k + \boldsymbol{y}^\mu\|^2 - \frac{\beta}{2} \|-\boldsymbol{y}_{k+1} + \boldsymbol{y}^\mu\|^2.
\end{aligned}
\tag{23}
$$

Summing over $k = 0, \cdots, \infty$, we have

$$
\begin{aligned}
& \sum_{k=0}^{\infty} \left( \frac{1}{2\beta} \|\boldsymbol{\lambda}_{k+1} - \boldsymbol{\lambda}_k\|^2 + \frac{\beta}{2} \|-\boldsymbol{y}_{k+1} + \boldsymbol{y}_k\|^2 \right) \\
\leq & \frac{1}{2\beta} \|\boldsymbol{\lambda}_0 - \boldsymbol{\lambda}^\mu\|^2 + \frac{\beta}{2} \|-\boldsymbol{y}_0 + \boldsymbol{y}^\mu\|^2.
\end{aligned}
$$

from which we deduce that $\boldsymbol{\lambda}_{k+1} - \boldsymbol{\lambda}_k \to \boldsymbol{0}$ and $-\boldsymbol{y}_{k+1} + \boldsymbol{y}_k \to \boldsymbol{0}$. Moreover, $\|\boldsymbol{\lambda}_k - \boldsymbol{\lambda}^\mu\|^2$ and $\|-\boldsymbol{y}_k + \boldsymbol{y}^\mu\|^2$ are bounded for all $k$, as well as $\|\boldsymbol{\lambda}_k\|$. Since

$$
\boldsymbol{\lambda}_{k+1} - \boldsymbol{\lambda}_k = \beta(\boldsymbol{x}_{k+1} - \boldsymbol{y}_{k+1}) = \beta(\boldsymbol{x}_{k+1} - \boldsymbol{x}^\mu) + \beta(-\boldsymbol{y}_{k+1} + \boldsymbol{y}^\mu)
$$

we deduce that $\boldsymbol{x}_{k+1} - \boldsymbol{y}_{k+1} \to \boldsymbol{0}$ and $\boldsymbol{x}_{k+1} - \boldsymbol{x}^\mu$ is also bounded.

From (15) and the convexity of $f$ and $g$, and using Proposition 2, we have:

$$
\begin{aligned}
& f(\boldsymbol{x}_{k+1}) - f(\boldsymbol{x}^\mu) + g(\boldsymbol{y}_{k+1}) - g(\boldsymbol{y}^\mu) \\
\leq & f_k(\boldsymbol{x}_{k+1}) - f_k(\boldsymbol{x}^\mu) + g(\boldsymbol{y}_{k+1}) - g(\boldsymbol{y}^\mu) \\
\leq & -\langle \boldsymbol{\lambda}_{k+1}, \boldsymbol{x}_{k+1} - \boldsymbol{y}_{k+1} \rangle + \beta \langle -\boldsymbol{y}_{k+1} + \boldsymbol{y}_k, \boldsymbol{x}_{k+1} - \boldsymbol{x}^\mu \rangle \to 0.
\end{aligned}
$$

On the other hand, from (8), (9), and (10), we have:

$$
\begin{aligned}
& f_k(\boldsymbol{x}_{k+1}) - f_k(\boldsymbol{x}^\mu) + g(\boldsymbol{y}_{k+1}) - g(\boldsymbol{y}^\mu) \\
\geq & f(\boldsymbol{x}_{k+1}) - f(\boldsymbol{x}^\mu) + g(\boldsymbol{y}_{k+1}) - g(\boldsymbol{y}^\mu) \\
\geq & \langle -\boldsymbol{\lambda}^\mu, \boldsymbol{x}_{k+1} - \boldsymbol{x}^\mu \rangle + \langle \boldsymbol{\lambda}^\mu, \boldsymbol{y}_{k+1} - \boldsymbol{y}^\mu \rangle \\
= & -\langle \boldsymbol{\lambda}^\mu, \boldsymbol{x}_{k+1} - \boldsymbol{y}_{k+1} \rangle \to 0.
\end{aligned}
$$

Thus, we have $f(\boldsymbol{x}_{k+1}) - f(\boldsymbol{x}^\mu) + g(\boldsymbol{y}_{k+1}) - g(\boldsymbol{y}^\mu) \to 0$ and $f_k(\boldsymbol{x}_{k+1}) - f_k(\boldsymbol{x}^\mu) + g(\boldsymbol{y}_{k+1}) - g(\boldsymbol{y}^\mu) \to 0, k \to \infty$.

If $F$ is $\xi$-monotone on $\mathcal{C}_=$, from Lemma 2 and (21) we have

$$
\begin{aligned}
&c\|\boldsymbol{x}_{k+1} - \boldsymbol{x}^\mu\|_2^\xi + \frac{1}{2\beta}\|\boldsymbol{\lambda}_{k+1} - \boldsymbol{\lambda}_k\|^2 + \frac{\beta}{2}\|-\boldsymbol{y}_{k+1} + \boldsymbol{y}_k\|^2 \\
\leq& \frac{1}{2\beta}\|\boldsymbol{\lambda}_k - \boldsymbol{\lambda}^\mu\|^2 - \frac{1}{2\beta}\|\boldsymbol{\lambda}_{k+1} - \boldsymbol{\lambda}^\mu\|^2 \\
&+ \frac{\beta}{2}\|-\boldsymbol{y}_k + \boldsymbol{y}^\mu\|^2 - \frac{\beta}{2}\|-\boldsymbol{y}_{k+1} + \boldsymbol{y}^\mu\|^2
\end{aligned}
\tag{24}
$$

From this we deduce that:

$$
\begin{aligned}
&c\|\boldsymbol{x}_{k+1} - \boldsymbol{x}^\mu\|_2^\xi + \sum_{k=0}^\infty \left(\frac{1}{2\beta}\|\boldsymbol{\lambda}_{k+1} - \boldsymbol{\lambda}_k\|^2 + \frac{\beta}{2}\|-\boldsymbol{y}_{k+1} + \boldsymbol{y}_k\|^2\right) \\
\leq& \frac{1}{2\beta}\|\boldsymbol{\lambda}^0 - \boldsymbol{\lambda}^\mu\|^2 + \frac{\beta}{2}\|-\boldsymbol{y}^0 + \boldsymbol{y}^\mu\|^2.
\end{aligned}
$$

Therefore, $\|\boldsymbol{x}_{k+1} - \boldsymbol{x}^\mu\|_2 \to 0, k \to \infty$. □

**Lemma 8.** *For Algorithm 1, we have*

$$
\begin{aligned}
&\frac{1}{2\beta}\|\boldsymbol{\lambda}_{k+1} - \boldsymbol{\lambda}_k\|^2 + \frac{\beta}{2}\|-\boldsymbol{y}_{k+1} + \boldsymbol{y}_k\|^2 \\
\leq& \frac{1}{2\beta}\|\boldsymbol{\lambda}_k - \boldsymbol{\lambda}_{k-1}\|^2 + \frac{\beta}{2}\|-\boldsymbol{y}_k + \boldsymbol{y}_{k-1}\|^2.
\end{aligned}
\tag{25}
$$

*Furthermore, if $F$ is $\xi$-monotone on $\mathcal{C}_=$, we have*

$$
\begin{aligned}
&c\|\boldsymbol{x}_{k+1} - \boldsymbol{x}_k\|^2 + \frac{1}{2\beta}\|\boldsymbol{\lambda}_{k+1} - \boldsymbol{\lambda}_k\|^2 + \frac{\beta}{2}\|-\boldsymbol{y}_{k+1} + \boldsymbol{y}_k\|^2 \\
\leq& \frac{1}{2\beta}\|\boldsymbol{\lambda}_k - \boldsymbol{\lambda}_{k-1}\|^2 + \frac{\beta}{2}\|-\boldsymbol{y}_k + \boldsymbol{y}_{k-1}\|^2.
\end{aligned}
\tag{26}
$$

*Proof of Lemma 8.* (15) gives:

$$
\begin{aligned}
&\langle\hat{\nabla}f_{k-1}(\boldsymbol{x}_k), \boldsymbol{x}_k - \boldsymbol{x}\rangle + \langle\hat{\nabla}g(\boldsymbol{y}_k), \boldsymbol{y}_k - \boldsymbol{y}\rangle \\
=& -\langle\boldsymbol{\lambda}_k, \boldsymbol{x}_k - \boldsymbol{y}_k - \boldsymbol{x} + \boldsymbol{y}\rangle + \beta\langle-\boldsymbol{y}_k + \boldsymbol{y}_{k-1}, \boldsymbol{x}_k - \boldsymbol{x}\rangle.
\end{aligned}
\tag{27}
$$

Letting $(\boldsymbol{x}, \boldsymbol{y}, \boldsymbol{\lambda}) = (\boldsymbol{x}_k, \boldsymbol{y}_k, \boldsymbol{\lambda}_k)$ in (15) and $(\boldsymbol{x}, \boldsymbol{y}, \boldsymbol{\lambda}) = (\boldsymbol{x}_{k+1}, \boldsymbol{y}_{k+1}, \boldsymbol{\lambda}_{k+1})$ in (27), and adding them together, and using (7), we have

$$
\begin{aligned}
& \langle \hat{\nabla} f_k\left(\boldsymbol{x}_{k+1}\right) - \hat{\nabla} f_{k-1}\left(\boldsymbol{x}_k\right), \boldsymbol{x}_{k+1} - \boldsymbol{x}_k \rangle + \langle \hat{\nabla} g\left(\boldsymbol{y}_{k+1}\right) - \hat{\nabla} g\left(\boldsymbol{y}_k\right), \boldsymbol{y}_{k+1} - \boldsymbol{y}_k \rangle \\
& = -\langle \boldsymbol{\lambda}_{k+1} - \boldsymbol{\lambda}_k, \boldsymbol{x}_{k+1} - \boldsymbol{y}_{k+1} - \boldsymbol{x}_k + \boldsymbol{y}_k \rangle + \beta\langle -\boldsymbol{y}_{k+1} + \boldsymbol{y}_k - (-\boldsymbol{y}_k + \boldsymbol{y}_{k-1}), \boldsymbol{x}_{k+1} - \boldsymbol{x}_k \rangle \\
& = -\frac{1}{\beta}\langle \boldsymbol{\lambda}_{k+1} - \boldsymbol{\lambda}_k, \boldsymbol{\lambda}_{k+1} - \boldsymbol{\lambda}_k - (\boldsymbol{\lambda}_k - \boldsymbol{\lambda}_{k-1}) \rangle \\
& \quad +\langle -\boldsymbol{y}_{k+1} + \boldsymbol{y}_k + (\boldsymbol{y}_k - \boldsymbol{y}_{k-1}), \boldsymbol{\lambda}_{k+1} - \boldsymbol{\lambda}_k + \beta\boldsymbol{y}_{k+1} - (\boldsymbol{\lambda}_k - \boldsymbol{\lambda}_{k-1} + \beta\boldsymbol{y}_k) \rangle \\
& = \frac{1}{2\beta}\left[ \|\boldsymbol{\lambda}_k - \boldsymbol{\lambda}_{k-1}\|^2 - \|\boldsymbol{\lambda}_{k+1} - \boldsymbol{\lambda}_k\|^2 - \|\boldsymbol{\lambda}_{k+1} - \boldsymbol{\lambda}_k - (\boldsymbol{\lambda}_k - \boldsymbol{\lambda}_{k-1})\|^2 \right] \\
& \quad + \frac{\beta}{2}\left[ \|-\boldsymbol{y}_k + \boldsymbol{y}_{k-1}\|^2 - \|-\boldsymbol{y}_{k+1} + \boldsymbol{y}_k\|^2 - \|-\boldsymbol{y}_{k+1} + \boldsymbol{y}_k - (-\boldsymbol{y}_k + \boldsymbol{y}_{k-1})\|^2 \right] \\
& \quad + \langle -\boldsymbol{y}_{k+1} + \boldsymbol{y}_k - (-\boldsymbol{y}_k + \boldsymbol{y}_{k-1}), \boldsymbol{\lambda}_{k+1} - \boldsymbol{\lambda}_k - (\boldsymbol{\lambda}_k - \boldsymbol{\lambda}_{k-1}) \rangle \\
& = \frac{1}{2\beta}\left( \|\boldsymbol{\lambda}_k - \boldsymbol{\lambda}_{k-1}\|^2 - \|\boldsymbol{\lambda}_{k+1} - \boldsymbol{\lambda}_k\|^2 \right) + \frac{\beta}{2}\left( \|-\boldsymbol{y}_k + \boldsymbol{y}_{k-1}\|^2 - \|-\boldsymbol{y}_{k+1} + \boldsymbol{y}_k\|^2 \right) \\
& \quad - \frac{1}{2\beta}\|\boldsymbol{\lambda}_{k+1} - \boldsymbol{\lambda}_k - (\boldsymbol{\lambda}_k - \boldsymbol{\lambda}_{k-1})\|^2 - \frac{\beta}{2}\|-\boldsymbol{y}_{k+1} + \boldsymbol{y}_k - (-\boldsymbol{y}_k + \boldsymbol{y}_{k-1})\|^2 \\
& \quad + \langle -\boldsymbol{y}_{k+1} + \boldsymbol{y}_k - (-\boldsymbol{y}_k + \boldsymbol{y}_{k-1}), \boldsymbol{\lambda}_{k+1} - \boldsymbol{\lambda}_k - (\boldsymbol{\lambda}_k - \boldsymbol{\lambda}_{k-1}) \rangle \\
& \leq \frac{1}{2\beta}\left( \|\boldsymbol{\lambda}_k - \boldsymbol{\lambda}_{k-1}\|^2 - \|\boldsymbol{\lambda}_{k+1} - \boldsymbol{\lambda}_k\|^2 \right) + \frac{\beta}{2}\left( \|-\boldsymbol{y}_k + \boldsymbol{y}_{k-1}\|^2 - \|-\boldsymbol{y}_{k+1} + \boldsymbol{y}_k\|^2 \right)
\end{aligned}
$$

By the convexity of $f_k$ and $f_{k-1}$, we have

$$
\begin{aligned}
\langle \hat{\nabla} f_k\left(\boldsymbol{x}_{k+1}\right), \boldsymbol{x}_{k+1} - \boldsymbol{x}_k \rangle &\geq f_k(\boldsymbol{x}_{k+1}) - f_k(\boldsymbol{x}_k), \\
-\langle \hat{\nabla} f_{k-1}\left(\boldsymbol{x}_k\right), \boldsymbol{x}_{k+1} - \boldsymbol{x}_k \rangle &\geq f_{k-1}(\boldsymbol{x}_k) - f_{k-1}(\boldsymbol{x}_{k+1}).
\end{aligned}
$$

Adding them together gives that:

$$
\begin{aligned}
\langle \hat{\nabla} f_k\left(\boldsymbol{x}_{k+1}\right) - \hat{\nabla} f_{k-1}\left(\boldsymbol{x}_k\right), \boldsymbol{x}_{k+1} - \boldsymbol{x}_k \rangle &\geq f_k(\boldsymbol{x}_{k+1}) - f_{k-1}(\boldsymbol{x}_{k+1}) - f_k(\boldsymbol{x}_k) + f_{k-1}(\boldsymbol{x}_k) \\
&= \langle F(\boldsymbol{x}_{k+1}) - F(\boldsymbol{x}_k), \boldsymbol{x}_{k+1} - \boldsymbol{x}_k \rangle \geq 0.
\end{aligned}
$$

Thus by the monotonicity of $F$ and $\hat{\nabla} g$, (25) follows. $\qquad\square$

**Theorem 7.** *If $F$ is monotone on $\mathcal{C}_=$, then for Algorithm 1, we have*

$$
-\|\boldsymbol{\lambda}^\mu\|\sqrt{\frac{\Delta^\mu}{\beta(K+1)}} \leq f\left(\boldsymbol{x}_{K+1}\right) + g\left(\boldsymbol{y}_{K+1}\right) - f\left(\boldsymbol{x}^\mu\right) - g\left(\boldsymbol{y}^\mu\right)
$$

$$
\leq f_K\left(\boldsymbol{x}_{K+1}\right) + g\left(\boldsymbol{y}_{K+1}\right) - f_K\left(\boldsymbol{x}^\mu\right) - g\left(\boldsymbol{y}^\mu\right) \tag{28}
$$

$$
\leq \frac{\Delta^\mu}{K+1} + \frac{2\Delta^\mu}{\sqrt{K+1}} + \|\boldsymbol{\lambda}^\mu\|\sqrt{\frac{\Delta^\mu}{\beta(K+1)}},
$$

$$
\|\boldsymbol{x}_{K+1} - \boldsymbol{y}_{K+1}\| \leq \sqrt{\frac{\Delta^\mu}{\beta(K+1)}}, \tag{29}
$$

*where $\Delta^\mu \triangleq \frac{1}{\beta}\|\boldsymbol{\lambda}_0 - \boldsymbol{\lambda}^\mu\|^2 + \beta\|\boldsymbol{y}_0 - \boldsymbol{y}^\mu\|^2$.*

*Furthermore, if $F$ is $\xi$-monotone on $\mathcal{C}_=$, we have*

$$
c\|\hat{\boldsymbol{x}}_{K+1} - \boldsymbol{x}^\mu\|^\xi \leq \frac{\Delta^\mu}{K+1}, \tag{30}
$$

$$
c\|\boldsymbol{x}_{K+1} - \boldsymbol{x}^\mu\|_2^\xi \leq \frac{\Delta^\mu}{K+1} + \frac{2\Delta^\mu}{\sqrt{K+1}}. \tag{31}
$$

*where $\hat{\boldsymbol{x}}_{K+1} = \frac{1}{K+1}\sum_{k=1}^{K+1}\boldsymbol{x}_K$.*

*Proof of Theorem 7.* Summing (23) over $k = 0, 1, \ldots, K$ and using the monotonicity of $\frac{1}{2\beta}\|\boldsymbol{\lambda}_{k+1} - \boldsymbol{\lambda}_k\|^2 + \frac{\beta}{2}\|-\boldsymbol{y}_{k+1} + \boldsymbol{y}_k\|^2$ from Lemma 8, we have:

$$\frac{1}{\beta}\|\boldsymbol{\lambda}_{K+1} - \boldsymbol{\lambda}_K\|^2 + \beta\|-\boldsymbol{y}_{K+1} + \boldsymbol{y}_K\|^2 \leq \frac{1}{K+1}\left(\frac{1}{\beta}\|\boldsymbol{\lambda}_0 - \boldsymbol{\lambda}^\mu\|^2 + \beta\|-\boldsymbol{y}_0 + \boldsymbol{y}^\mu\|^2\right) \quad (32)$$

From the above we deduce that

$$\beta\|\boldsymbol{x}_{K+1} - \boldsymbol{y}_{K+1}\| = \|\boldsymbol{\lambda}_{K+1} - \boldsymbol{\lambda}_K\| \leq \sqrt{\frac{\beta\Delta^\mu}{(K+1)}},$$

$$\|-\boldsymbol{y}_{K+1} + \boldsymbol{y}_K\| \leq \sqrt{\frac{\Delta^\mu}{\beta(K+1)}}.$$

On the other hand, (23) gives:

$$\frac{1}{2\beta}\|\boldsymbol{\lambda}_{k+1} - \boldsymbol{\lambda}^\mu\|^2 + \frac{\beta}{2}\|-\boldsymbol{y}_{k+1} + \boldsymbol{y}^\mu\|^2$$

$$\leq \frac{1}{2\beta}\|\boldsymbol{\lambda}_k - \boldsymbol{\lambda}^\mu\|^2 + \frac{\beta}{2}\|-\boldsymbol{y}_k + \boldsymbol{y}^\mu\|^2$$

$$\leq \frac{1}{2\beta}\|\boldsymbol{\lambda}_0 - \boldsymbol{\lambda}^\mu\|^2 + \frac{\beta}{2}\|-\boldsymbol{y}_0 + \boldsymbol{y}^\mu\|^2 = \frac{1}{2}\Delta^\mu.$$

Hence, we have:

$$\|\boldsymbol{\lambda}_{K+1} - \boldsymbol{\lambda}^{\mu_t}\| \leq \sqrt{\beta\Delta^{\mu_t}}, \quad (33)$$

$$\|-\boldsymbol{y}_{K+1} + \boldsymbol{y}^{\mu_t}\| \leq \sqrt{\frac{\Delta^{\mu_t}}{\beta}}. \quad (34)$$

Then from (16) and the convexity of $f$ and $g$, we have:

$$f(\boldsymbol{x}_{K+1}) - f(\boldsymbol{x}^\mu) + g(\boldsymbol{y}_{K+1}) - g(\boldsymbol{y}^\mu) + \langle\boldsymbol{\lambda}^\mu, \boldsymbol{x}_{K+1} - \boldsymbol{y}_{K+1}\rangle$$

$$\leq f_K(\boldsymbol{x}_{K+1}) - f_K(\boldsymbol{x}^\mu) + g(\boldsymbol{y}_{K+1}) - g(\boldsymbol{y}^\mu) + \langle\boldsymbol{\lambda}^\mu, \boldsymbol{x}_{K+1} - \boldsymbol{y}_{K+1}\rangle$$

$$\leq \frac{1}{\beta}\|\boldsymbol{\lambda}_{K+1} - \boldsymbol{\lambda}^\mu\|\|\boldsymbol{\lambda}_{K+1} - \boldsymbol{\lambda}_K\| + \|-\boldsymbol{y}_{K+1} + \boldsymbol{y}_K\|\|\boldsymbol{\lambda}_{K+1} - \boldsymbol{\lambda}_K\| \quad (35)$$

$$+ \beta\|-\boldsymbol{y}_{K+1} + \boldsymbol{y}_K\|\|-\boldsymbol{y}_{K+1} + \boldsymbol{y}^\mu\|$$

$$\leq \frac{\Delta^\mu}{K+1} + \frac{2\Delta^\mu}{\sqrt{K+1}}.$$

From Lemma 3, we have (28).

If in addition $F$ is $\xi$-monotone on $\mathcal{C}_=$, using (24), we can obtain the following inequality similar to (32):

$$c\frac{\sum_{k=0}^K\|\boldsymbol{x}_{k+1} - \boldsymbol{x}^\mu\|_2^\xi}{K+1} + \frac{1}{\beta}\|\boldsymbol{\lambda}_{K+1} - \boldsymbol{\lambda}_K\|^2 + \beta\|-\boldsymbol{y}_{K+1} + \boldsymbol{y}_K\|^2$$

$$\leq \frac{1}{K+1}\left(\frac{1}{\beta}\|\boldsymbol{\lambda}_0 - \boldsymbol{\lambda}^\mu\|^2 + \beta\|-\boldsymbol{y}_0 + \boldsymbol{y}^\mu\|^2\right)$$

By the convexity of $\|\cdot\|_2^\xi$ we have $c\|\hat{\boldsymbol{x}}_{K+1} - \boldsymbol{x}^\mu\|_2^\xi \leq \frac{\Delta^\mu}{K+1}$. And from (35) we can see that $c\|\boldsymbol{x}_{K+1} - \boldsymbol{x}^\mu\|_2^\xi \leq \frac{\Delta^\mu}{K+1} + \frac{2\Delta^\mu}{\sqrt{K+1}}$. $\square$

**Theorem 8.** *If $F$ is monotone on $\mathcal{C}_=$, then for Algorithm 1, we have*

$$|f(\hat{\boldsymbol{x}}_{K+1}) + g(\hat{\boldsymbol{y}}_{K+1}) - f(\boldsymbol{x}^\mu) - g(\boldsymbol{y}^\mu)| \leq \frac{\Delta^\mu}{2(K+1)} + \frac{2\sqrt{\beta\Delta^\mu}\|\boldsymbol{\lambda}^\mu\|}{\beta(K+1)},$$

$$\|\hat{\boldsymbol{x}}_{K+1} - \hat{\boldsymbol{y}}_{K+1}\| \leq \frac{2\sqrt{\beta\Delta^\mu}}{\beta(K+1)} \quad (36)$$

*where $\hat{\boldsymbol{x}}_{K+1} = \frac{1}{K+1}\sum_{k=1}^{K+1}\boldsymbol{x}_k, \hat{\boldsymbol{y}}_{K+1} = \frac{1}{K+1}\sum_{k=1}^{K+1}\boldsymbol{y}_k$, and $\Delta^\mu \triangleq \frac{1}{\beta}\|\boldsymbol{\lambda}_0 - \boldsymbol{\lambda}^\mu\|^2 + \beta\|\boldsymbol{y}_0 - \boldsymbol{y}^\mu\|^2$.*

*Proof of Theorem 8.* Summing (20) over $k = 0, 1, \ldots, K$, dividing both sides with $K+1$, and using the definitions of $\hat{\boldsymbol{x}}_{K+1}$ and $\hat{\boldsymbol{y}}_{K+1}$ and the convexity of $f$ and $g$, we have

$$f(\hat{\boldsymbol{x}}_{K+1}) + g(\hat{\boldsymbol{y}}_{K+1}) - f(\boldsymbol{x}^\mu) - g(\boldsymbol{y}^\mu) + \langle \boldsymbol{\lambda}^\mu, \hat{\boldsymbol{x}}_{K+1} - \hat{\boldsymbol{y}}_{K+1}\rangle \leq \frac{\Delta^\mu}{2(K+1)}.$$

From (7) and (33), we have:

$$\begin{aligned}
\|\hat{\boldsymbol{x}}_{K+1} - \hat{\boldsymbol{y}}_{K+1}\| &= \frac{1}{\beta(K+1)}\|\sum_{k=0}^{K}(\boldsymbol{\lambda}_{k+1} - \boldsymbol{\lambda}_k)\| \\
&= \frac{1}{\beta(K+1)}\|\boldsymbol{\lambda}_{K+1} - \boldsymbol{\lambda}_0\| \\
&\leq \frac{1}{\beta(K+1)}(\|\boldsymbol{\lambda}_0 - \boldsymbol{\lambda}^\mu\| + \|\boldsymbol{\lambda}_{K+1} - \boldsymbol{\lambda}^\mu\|) \\
&\leq \frac{2\sqrt{\beta\Delta^\mu}}{\beta(K+1)}
\end{aligned}$$

Finally, from Lemma 3, the conclusion follows. $\qquad\square$

From Proposition 1 and the fact that $\mathcal{C}$ is compact we can see that $\lim_{\mu \to 0} \boldsymbol{x}^\mu$ and $\lim_{\mu \to 0} \boldsymbol{\lambda}^\mu$ exist and are finite. Let $\boldsymbol{x}^\star = \lim_{\mu \to 0} \boldsymbol{x}^\mu$ and $\boldsymbol{\lambda}^\star = \lim_{\mu \to 0} \boldsymbol{\lambda}^\mu$, then $\boldsymbol{x}^\star \in \mathcal{S}^\star_{\mathcal{C}, F}$.

**Theorem 9.** $\exists \tilde{\mu} > 0$, *s.t.* *if* $\mu_t < \tilde{\mu}$, *then* $\left|F(\boldsymbol{x}^{(t)}_{K+1})^\mathsf{T}(\boldsymbol{x}^{(t)}_{K+1} - \boldsymbol{x}^\star)\right| \leq 2(\frac{\Delta^\mu}{K+1} + \frac{2\Delta^\mu}{\sqrt{K+1}} + \|\boldsymbol{\lambda}^\star\|\sqrt{\frac{\Delta^\mu}{\beta(K+1)}})$, $\left|F(\boldsymbol{x}^\star)^\mathsf{T}(\boldsymbol{x}^{(t)}_{K+1} - \boldsymbol{x}^\star)\right| \leq 2(\frac{\Delta^\mu}{K+1} + \frac{2\Delta^\mu}{\sqrt{K+1}} + \|\boldsymbol{\lambda}^\star\|\sqrt{\frac{\Delta^\mu}{\beta(K+1)}})$ *and* $\left|F(\boldsymbol{x}^\star)^\mathsf{T}(\hat{\boldsymbol{x}}^{(t)}_{K+1} - \boldsymbol{x}^\star)\right| \leq 2(\frac{\Delta^\mu}{2(K+1)} + \frac{2\sqrt{\beta\Delta^\mu}\|\boldsymbol{\lambda}^\star\|}{\beta(K+1)})$, $\forall K \geq 0$.

*Proof of Theorem 9.* For simplicity we let $B(\boldsymbol{x})$ denote the $log$-barrier term $-\sum_{i=1}^{m} log(-\varphi_i(\boldsymbol{x}))$. From Theorem 7 and 8 we have

$$\begin{aligned}
&\left|F(\boldsymbol{x}^\mu)^\mathsf{T}(\boldsymbol{x}^{(t)}_{K+1} - \boldsymbol{x}^\mu) + \mu(B(\boldsymbol{y}^{(t)}_{K+1}) - B(\boldsymbol{y}^\mu))\right| \\
&= \left|f(\boldsymbol{x}^{(t)}_{K+1}) - f(\boldsymbol{x}^\mu) + g(\boldsymbol{y}^{(t)}_{K+1}) - g(\boldsymbol{y}^\mu)\right| \\
&\leq \frac{\Delta^\mu}{K+1} + \frac{2\Delta^\mu}{\sqrt{K+1}} + \|\boldsymbol{\lambda}^\mu\|\sqrt{\frac{\Delta^\mu}{\beta(K+1)}} \quad\quad (37)
\end{aligned}$$

$$\begin{aligned}
&\left|F(\boldsymbol{x}^{K+1})^\mathsf{T}(\boldsymbol{x}^{(t)}_{K+1} - \boldsymbol{x}^\mu) + \mu(B(\boldsymbol{y}^{(t)}_{K+1}) - B(\boldsymbol{y}^\mu))\right| \\
&= \left|f_K(\boldsymbol{x}^{(t)}_{K+1}) - f_K(\boldsymbol{x}^\mu) + g(\boldsymbol{y}^{(t)}_{K+1}) - g(\boldsymbol{y}^\mu)\right| \\
&\leq \frac{\Delta^\mu}{K+1} + \frac{2\Delta^\mu}{\sqrt{K+1}} + \|\boldsymbol{\lambda}^\mu\|\sqrt{\frac{\Delta^\mu}{\beta(K+1)}} \quad\quad (38)
\end{aligned}$$

$$\begin{aligned}
&\left|F(\boldsymbol{x}^\mu)^\mathsf{T}(\hat{\boldsymbol{x}}^{(t)}_{K+1} - \boldsymbol{x}^\mu) + \mu(B(\hat{\boldsymbol{y}}^{(t)}_{K+1}) - B(\boldsymbol{y}^\mu))\right| \\
&= \left|f(\hat{\boldsymbol{x}}^{(t)}_{K+1}) - f(\boldsymbol{x}^\mu) + g(\hat{\boldsymbol{y}}^{(t)}_{K+1}) - g(\boldsymbol{y}^\mu)\right| \\
&\leq \frac{\Delta^\mu}{2(K+1)} + \frac{2\sqrt{\beta\Delta^\mu}\|\boldsymbol{\lambda}^\mu\|}{\beta(K+1)}. \quad\quad (39)
\end{aligned}$$

From Proposition 1 in App. A we know that when $\mu \to 0$, $\boldsymbol{x}^\mu \to \boldsymbol{x}^\star$, $\boldsymbol{\lambda}^\mu \to \boldsymbol{\lambda}^\star$ and $\mu(B(\boldsymbol{y}_{K+1}^{(t)}) - B(\boldsymbol{y}^\mu)) \to 0$, so $\exists \tilde{\mu} > 0$, s.t. if $\mu_t < \tilde{\mu}$, then we have

$$\left| F(\boldsymbol{x}^\star)^\intercal(\boldsymbol{x}_{K+1}^{(t)} - \boldsymbol{x}^\star) \right| \leq 2 \left( \frac{\Delta^\mu}{K+1} + \frac{2\Delta^\mu}{\sqrt{K+1}} + \|\boldsymbol{\lambda}^\star\| \sqrt{\frac{\Delta^\mu}{\beta(K+1)}} \right)$$

$$\left| F(\boldsymbol{x}_{K+1}^{(t)})^\intercal(\boldsymbol{x}_{K+1}^{(t)} - \boldsymbol{x}^\star) \right| \leq 2 \left( \frac{\Delta^\mu}{K+1} + \frac{2\Delta^\mu}{\sqrt{K+1}} + \|\boldsymbol{\lambda}^\star\| \sqrt{\frac{\Delta^\mu}{\beta(K+1)}} \right)$$

$$\left| F(\boldsymbol{x}^\star)^\intercal(\hat{\boldsymbol{x}}_{K+1}^{(t)} - \boldsymbol{x}^\star) \right| \leq 2 \left( \frac{\Delta^\mu}{2(K+1)} + \frac{2\sqrt{\beta\Delta^\mu}\|\boldsymbol{\lambda}^\star\|}{\beta(K+1)} \right).$$

$\square$

To make the dependencies of the constants clear, here we restate Theorem 2 and Theorem 3 and provide their proofs.

**Theorem 10** (restatement of Theorem 2). *Given an operator $F : \mathcal{X} \to \mathbb{R}^n$ monotone on $\mathcal{C}_=$ (Def. 1), and either $F$ is strictly monotone on $\mathcal{C}$ or one of $\varphi_i$ is strictly convex. Assume there exists $r > 0$ or $s > 0$ such that $F$ is star-$\xi$-monotone—as per Def. 1—on either $\hat{\mathcal{C}}_r$ or $\tilde{\mathcal{C}}_s$, resp. Let $\Delta \triangleq \frac{1}{\beta}\|\boldsymbol{\lambda}_0 - \boldsymbol{\lambda}^\star\|^2 + \beta\|\boldsymbol{y}_0 - \boldsymbol{y}^\star\|^2$.*

*Let $\boldsymbol{x}_K^{(t)}$ and $\hat{\boldsymbol{x}}_K^{(t)} \triangleq \frac{1}{K}\sum_{k=1}^K \boldsymbol{x}_k^{(t)}$ denote the last and average iterate of Algorithm 1, respectively, run with sufficiently small $\mu_{-1}$. Then there exists $K_0 \in \mathbb{N}$, $K_0$ depends on $r$ or $s$, s.t.$\forall K > K_0$, for all $t \in [T]$, we have that:*

1. *$\|\boldsymbol{x}_K^{(t)} - \boldsymbol{x}^\star\| \leq (\frac{4}{c}(\frac{\Delta}{K} + \frac{2\Delta}{\sqrt{K}} + \|\boldsymbol{\lambda}^\star\|\sqrt{\frac{\Delta}{\beta K}}))^{1/\xi}$.*

2. *If in addition $F$ is $\xi$-monotone on $C_=$, we have $\|\hat{\boldsymbol{x}}_K^{(t)} - \boldsymbol{x}^\star\| \leq (\frac{2\Delta}{cK})^{1/\xi}$.*

3. *Moreover, if $F$ is $L$-Lipschitz on $\mathcal{C}_=$—as per Assumption 1— then $\mathcal{G}(\boldsymbol{x}_K^{(t)}, \mathcal{C}) \leq M_0(\frac{4}{c}(\frac{\Delta}{K} + \frac{2\Delta}{\sqrt{K}} + \|\boldsymbol{\lambda}^\star\|\sqrt{\frac{\Delta}{\beta K}}))^{1/\xi}$ and $\mathcal{G}(\hat{\boldsymbol{x}}_K^{(t)}, \mathcal{C}) \leq M_0 \left( \frac{2\Delta}{c(K+1)} \right)^{1/\xi}$,*

*where $M_0 = DL + M$ is a linear function of $L$, see the proof of Lemma 1 in App. B.*

*Proof of Theorem 10.* Without loss of generality, we suppose that $F$ is star-$\xi$-monotone on $\hat{\mathcal{C}}_r$. Since you $\phi(\boldsymbol{y}_k) \leq \boldsymbol{0}$, from (29) we can see that $\exists K_0 \in \mathbb{N}$, $K_0$ depends on $r$, s.t.$\forall K \geq K_0$, $\boldsymbol{x}_k \in \hat{\mathcal{C}}_r$, so if $\mu_t < \tilde{\mu}$ ($\tilde{\mu}$ is defined in Theorem 9), by Theorem 9 we have

$$\begin{aligned} c\|\boldsymbol{x}_{K+1}^{(t)} - \boldsymbol{x}^\star\|^\xi &\leq \left| (F(\boldsymbol{x}_{K+1}^{(t)})^\intercal - F(\boldsymbol{x}^\star)^\intercal)(\boldsymbol{x}_{K+1}^{(t)} - \boldsymbol{x}^\star) \right| \\ &\leq \left| F(\boldsymbol{x}^\star)^\intercal(\boldsymbol{x}_{K+1}^{(t)} - \boldsymbol{x}^\star) \right| + \left| F(\boldsymbol{x}_{K+1}^{(t)})^\intercal(\boldsymbol{x}_{K+1}^{(t)} - \boldsymbol{x}^\star) \right| \\ &\leq 4 \left( \frac{\Delta}{K+1} + \frac{2\Delta}{\sqrt{K+1}} + \|\boldsymbol{\lambda}^\star\| \sqrt{\frac{\Delta}{\beta(K+1)}} \right). \end{aligned}$$

So we have

$$\|\boldsymbol{x}_{K+1}^{(t)} - \boldsymbol{x}^\star\| \leq \left( \frac{4}{c} \left( \frac{\Delta}{K+1} + \frac{2\Delta}{\sqrt{K+1}} + \|\boldsymbol{\lambda}^\star\| \sqrt{\frac{\Delta}{\beta(K+1)}} \right) \right)^{1/\xi}.$$

If in addition $F$ is $\xi$-monotone on $C_=$, then from (30) we know that when $\hat{\mu}$ is small enough, we have

$$\|\hat{\boldsymbol{x}}_{K+1}^{(t)} - \boldsymbol{x}^\star\| \leq \left( \frac{2\Delta}{c(K+1)} \right)^{1/\xi}.$$

If $F$ is $L$-Lipschitz on $\mathcal{C}_=$, then from Lemma 1 we can see that

$$\mathcal{G}(\boldsymbol{x}_K^{(t)}, \mathcal{C}) \le M_0 \left( \frac{4}{c} \left( \frac{\Delta}{K+1} + \frac{2\Delta}{\sqrt{K+1}} + \|\boldsymbol{\lambda}^\star\| \sqrt{\frac{\Delta}{\beta(K+1)}} \right) \right)^{1/\xi},$$

$$\mathcal{G}(\hat{\boldsymbol{x}}_K^{(t)}, \mathcal{C}) \le M_0 \left( \frac{2\Delta}{c(K+1)} \right)^{1/\xi}.$$

$\square$

**Theorem 11** (restatement of Theorem 3). *Given an operator $F : \mathcal{X} \to \mathbb{R}^n$, assume: (i) $F$ is monotone on $\mathcal{C}_=$, as per Def. 1; (ii) either $F$ is strictly monotone on $\mathcal{C}$ or one of $\varphi_i$ is strictly convex; and (iii) $\inf\limits_{\boldsymbol{x} \in S \setminus \{\boldsymbol{x}^\star\}} F(\boldsymbol{x})^\mathsf{T} \frac{\boldsymbol{x} - \boldsymbol{x}^\star}{\|\boldsymbol{x} - \boldsymbol{x}^\star\|} = a > 0$, where $S \equiv \hat{\mathcal{C}}_r$ or $\tilde{\mathcal{C}}_s$. Let $\Delta \triangleq \frac{1}{\beta} \|\boldsymbol{\lambda}_0 - \boldsymbol{\lambda}^\star\|^2 + \beta \|\boldsymbol{y}_0 - \boldsymbol{y}^\star\|^2$. Let $\boldsymbol{x}_K^{(t)}$ and $\hat{\boldsymbol{x}}_K^{(t)} \triangleq \frac{1}{K} \sum_{k=1}^K \boldsymbol{x}_k^{(t)}$ denote the last and average iterate of Algorithm 1, respectively, run with sufficiently small $\mu_{-1}$. Then there exists $K_0 \in \mathbb{N}$, $K_0$ depends on $r$ or $s$, s.t. $\forall t \in [T]$, $\forall K > K_0$, we have that:*

1. $\left\| \boldsymbol{x}_K^{(t)} - \boldsymbol{x}^\star \right\| \le \frac{2}{a} \left( \frac{\Delta}{K} + \frac{2\Delta}{\sqrt{K}} + \|\boldsymbol{\lambda}^\star\| \sqrt{\frac{\Delta}{\beta K}} \right).$

2. *If in addition* $\inf\limits_{\boldsymbol{x} \in S \setminus \{\boldsymbol{x}^\star\}} F(\boldsymbol{x}^\star)^\mathsf{T} \frac{\boldsymbol{x} - \boldsymbol{x}^\star}{\|\boldsymbol{x} - \boldsymbol{x}^\star\|} = b > 0$ *(with $S \equiv \hat{\mathcal{C}}_r$ or $\tilde{\mathcal{C}}_s$), then* $\left\| \hat{\boldsymbol{x}}_K^{(t)} - \boldsymbol{x}^\star \right\| \le \frac{2}{b} \left( \frac{\Delta}{2K} + \frac{2\sqrt{\beta\Delta}\|\boldsymbol{\lambda}^\star\|}{\beta K} \right).$

3. *Moreover, if $F$ is $L$-Lipschitz on $\mathcal{C}_=$—as per Assumption 1—then* $\mathcal{G}(\boldsymbol{x}_K^{(t)}, \mathcal{C}) \le \frac{2M_0}{a} \left( \frac{\Delta}{K} + \frac{2\Delta}{\sqrt{K}} + \|\boldsymbol{\lambda}^\star\| \sqrt{\frac{\Delta}{\beta K}} \right)$ *and* $\mathcal{G}(\hat{\boldsymbol{x}}_K^{(t)}, \mathcal{C}) \le \frac{2M_0}{b} \left( \frac{\Delta}{2K} + \frac{2\sqrt{\beta\Delta}\|\boldsymbol{\lambda}^\star\|}{\beta K} \right),$

*where $M_0 = DL + M$ is a linear function of $L$, see the proof of Lemma 1 in App. B.*

*Proof of Theorem 11.* Without loss of generality, we suppose $\inf\limits_{\boldsymbol{x} \in \hat{\mathcal{C}}_r \setminus \{\boldsymbol{x}^\star\}} F(\boldsymbol{x})^\mathsf{T} \frac{\boldsymbol{x} - \boldsymbol{x}^\star}{\|\boldsymbol{x} - \boldsymbol{x}^\star\|} = a > 0$. When $K \ge K_0$ ($K_0$ and $\tilde{\mu}$ are defined in the proof of Theorem 10 and in Theorem 9, resp.), by Theorem 9 we have that

$$\|\boldsymbol{x}_{K+1}^{(t)} - \boldsymbol{x}^\star\| \le \frac{1}{a} \left| F(\boldsymbol{x}_{K+1}^{(t)})^\mathsf{T} (\boldsymbol{x}_{K+1}^{(t)} - \boldsymbol{x}^\star) \right|$$

$$\le \frac{2}{a} \left( \frac{\Delta}{K+1} + \frac{2\Delta}{\sqrt{K+1}} + \|\boldsymbol{\lambda}^\star\| \sqrt{\frac{\Delta}{\beta(K+1)}} \right).$$

Similarly, if $\inf\limits_{\boldsymbol{x} \in \hat{\mathcal{C}}_r \setminus \{\boldsymbol{x}^\star\}} F(\boldsymbol{x}^\star)^\mathsf{T} \frac{\boldsymbol{x} - \boldsymbol{x}^\star}{\|\boldsymbol{x} - \boldsymbol{x}^\star\|} = b > 0$, we have that

$$\|\hat{\boldsymbol{x}}_{K+1}^{(t)} - \boldsymbol{x}^\star\| \le \frac{2}{b} \left( \frac{\Delta}{2(K+1)} + \frac{2\sqrt{\beta\Delta}\|\boldsymbol{\lambda}^\star\|}{\beta(K+1)} \right).$$

If $F$ is $L$-Lipschitz on $\mathcal{C}_=$, then from Lemma 1 we can see that

$$\mathcal{G}(\boldsymbol{x}_K^{(t)}, \mathcal{C}) \le \frac{2M_0}{a} \left( \frac{\Delta}{K+1} + \frac{2\Delta}{\sqrt{K+1}} + \|\boldsymbol{\lambda}^\star\| \sqrt{\frac{\Delta}{\beta(K+1)}} \right),$$

$$\mathcal{G}(\hat{\boldsymbol{x}}_K^{(t)}, \mathcal{C}) \le \frac{2M_0}{b} \left( \frac{\Delta}{2(K+1)} + \frac{2\sqrt{\beta\Delta}\|\boldsymbol{\lambda}^\star\|}{\beta(K+1)} \right).$$

$\square$

From the above proofs, we can see that by setting $\mu_{-1}$ small enough, Theorem 2 and 3 hold true. But since we do not know exactly how small $\mu_{-1}$ should be, in practice, we can set a small $\mu_{-1}$, and

$\mu_t$ could be smaller than any fixed positive number after a very small number of outer loops, as $\mu_t$ decays exponentially. Thus, Algorithm 1 is actually parameter-free.

Note that the assumption (iii) in Theorem 3 is the weakening of the assumption $\inf\limits_{\boldsymbol{x} \in S \setminus \{\boldsymbol{x}^\star\}} F(\boldsymbol{x}^\star)^\mathsf{T} \frac{\boldsymbol{x} - \boldsymbol{x}^\star}{\|\boldsymbol{x} - \boldsymbol{x}^\star\|} > 0$, where $S = \hat{\mathcal{C}}_r$ or $\tilde{\mathcal{C}}_s$. In fact, by the monotonicity of $F$, we have $F(\boldsymbol{x})^\mathsf{T}(\boldsymbol{x} - \boldsymbol{x}^\star) \geq F(\boldsymbol{x}^\star)^\mathsf{T}(\boldsymbol{x} - \boldsymbol{x}^\star)$, so if $\inf\limits_{\boldsymbol{x} \in S \setminus \{\boldsymbol{x}^\star\}} F(\boldsymbol{x}^\star)^\mathsf{T} \frac{\boldsymbol{x} - \boldsymbol{x}^\star}{\|\boldsymbol{x} - \boldsymbol{x}^\star\|} > 0$, there must be $\inf\limits_{\boldsymbol{x} \in S \setminus \{\boldsymbol{x}^\star\}} F(\boldsymbol{x})^\mathsf{T} \frac{\boldsymbol{x} - \boldsymbol{x}^\star}{\|\boldsymbol{x} - \boldsymbol{x}^\star\|} > 0$.

### B.4 PARAMETER-FREE CONVERGENCE RATE

In this section, we give a convergence rate taking into account both the inner and the outer loop when the operator $F$ is $L$-Lipschitz.

First, we bound $\|\boldsymbol{x}^\star - \boldsymbol{x}^\mu\|$ under the assumptions in Theorem 2 and Theorem 3, resp, and give the following lemma:

**Lemma 9.** *For monotone $F$, then for any $\mu > 0$, we have:*

*(i) If $F$ is star-$\xi$-monotone (Def. 1), then $\|\boldsymbol{x}^\star - \boldsymbol{x}^\mu\| \leq \left(\frac{m}{c}\mu\right)^{\frac{1}{\xi}}$;*

*(ii) If $a \triangleq \inf\limits_{\boldsymbol{x} \in S \setminus \{\boldsymbol{x}^\star\}} F(\boldsymbol{x})^\mathsf{T} \frac{\boldsymbol{x} - \boldsymbol{x}^\star}{\|\boldsymbol{x} - \boldsymbol{x}^\star\|} > 0$, where $S \equiv \hat{\mathcal{C}}_r$ or $\tilde{\mathcal{C}}_s$, then $\|\boldsymbol{x}^\star - \boldsymbol{x}^\mu\| \leq \frac{m}{a}\mu$.*

*Proof.* Consider convex problem

$$\begin{cases} \min_{\boldsymbol{x}} \ F(\boldsymbol{x}^\mu)^\mathsf{T}\boldsymbol{x} \\ s.t. \quad \varphi(\boldsymbol{x}) \leq \boldsymbol{0} \\ \qquad \boldsymbol{C}\boldsymbol{x} = \boldsymbol{d} \end{cases} \tag{$\mathcal{P}$}$$

The Lagrangian of ($\mathcal{P}$) is

$$L(\boldsymbol{x}, \boldsymbol{\lambda}, \boldsymbol{\nu}) = F(\boldsymbol{x}^\mu)^\mathsf{T}\boldsymbol{x} + \boldsymbol{\lambda}^\mathsf{T}\varphi(\boldsymbol{x}) + \boldsymbol{\nu}^\mathsf{T}(\boldsymbol{C}\boldsymbol{x} - \boldsymbol{d}).$$

There exists $\bar{\boldsymbol{\lambda}}^\mu > \boldsymbol{0}$ and $\boldsymbol{\nu}^\mu$, s.t. $(\boldsymbol{x}^\mu, \bar{\boldsymbol{\lambda}}^\mu, \boldsymbol{\nu}^\mu)$ is a KKT point of (KKT-2). By the stationarity condition in (KKT-2), we have that

$$g(\bar{\boldsymbol{\lambda}}^\mu, \boldsymbol{\nu}^\mu) = \inf_{\boldsymbol{x}} L(\boldsymbol{x}^\mu, \bar{\boldsymbol{\lambda}}^\mu, \boldsymbol{\nu}^\mu) = F(\boldsymbol{x}^\mu)^\mathsf{T}\boldsymbol{x}^\mu + \bar{\boldsymbol{\lambda}}^\mathsf{T}\varphi(\boldsymbol{x}^\mu) + \boldsymbol{\nu}^\mathsf{T}\underbrace{(\boldsymbol{C}\boldsymbol{x}^\mu - \boldsymbol{d})}_{=\boldsymbol{0}}.$$

Note that by the complementarity slackness condition in (KKT-2), we have $\bar{\boldsymbol{\lambda}}^\mathsf{T}\varphi(\boldsymbol{x}^\mu) = -m\mu$.

Since $\boldsymbol{x}^\star$ is a feasible point of ($\mathcal{P}$), $(\bar{\boldsymbol{\lambda}}^\mu, \boldsymbol{\nu}^\mu)$ dual feasible, thus by the duality theory, we have:

$$F(\boldsymbol{x}^\mu)^\mathsf{T}\boldsymbol{x}^\star \geq g(\bar{\boldsymbol{\lambda}}^\mu, \boldsymbol{\nu}^\mu) = F(\boldsymbol{x}^\mu)^\mathsf{T}\boldsymbol{x}^\mu - m\mu,$$

from where we deduce that

$$F(\boldsymbol{x}^\mu)^\mathsf{T}(\boldsymbol{x}^\mu - \boldsymbol{x}^\star) \leq m\mu. \tag{40}$$

Therefore,

(i) If $F$ is star-$\xi$-monotone:

Since $\boldsymbol{x}^\star \in \mathcal{S}_{\mathcal{C},F}^\star$, we have

$$F(\boldsymbol{x}^\star)^\mathsf{T}(\boldsymbol{x}^\mu = \boldsymbol{x}^\star) \geq 0. \tag{41}$$

Subtract (40) by (41) and using the star-$\xi$-monotonicity of $F$, we obtain

$$c\|\boldsymbol{x}^\mu - \boldsymbol{x}^\star\|^\xi \leq (F(\boldsymbol{x}^\mu) - F(\boldsymbol{x}^\star))^\mathsf{T}(\boldsymbol{x}^\mu - \boldsymbol{x}^\star) \leq m\mu.$$

(ii) If $a \triangleq \inf\limits_{\boldsymbol{x} \in S \setminus \{\boldsymbol{x}^\star\}} F(\boldsymbol{x})^\mathsf{T} \frac{\boldsymbol{x} - \boldsymbol{x}^\star}{\|\boldsymbol{x} - \boldsymbol{x}^\star\|} > 0$, where $S \equiv \hat{\mathcal{C}}_r$ or $\tilde{\mathcal{C}}_s$, since $\boldsymbol{x}^\mu \in \mathcal{S}$, by (40) we have that $a\|\boldsymbol{x}^\mu - \boldsymbol{x}^\star\| \leq m\mu$.

$\square$

We give another lemma that would be used in the proofs of our main results in this section.

**Lemma 10.** *Assume $F$ is monotone and $L$-smooth (Assumption 1), then we have*

$$\left| F(\boldsymbol{x}^\star)^\mathsf{T}(\boldsymbol{x}_{K+1}^{(t)} - \boldsymbol{x}^\star) + \mu_t(B(\boldsymbol{y}_{K+1}^{(t)}) - B(\boldsymbol{y}^{\mu_t})) \right|$$

$$\leq \frac{\Delta^{\mu_t}}{K+1} + \frac{2\Delta^{\mu_t}}{\sqrt{K+1}} + \|\boldsymbol{\lambda}^{\mu_t}\|\sqrt{\frac{\Delta^{\mu_t}}{\beta(K+1)}} + m\mu_t \tag{42}$$

$$+ L\|\boldsymbol{x}^{\mu_t} - \boldsymbol{x}^\star\|\left( \|\boldsymbol{x}^{\mu_t} - \boldsymbol{x}^\star\| + \sqrt{\frac{\Delta^{\mu_t}}{\beta(K+1)}} + \sqrt{\frac{\Delta^{\mu_t}}{\beta}} \right),$$

$$\left| F(\boldsymbol{x}_K^{(t)})^\mathsf{T}(\boldsymbol{x}_{K+1}^{(t)} - \boldsymbol{x}^\star) + \mu_t(B(\boldsymbol{y}_{K+1}^{(t)}) - B(\boldsymbol{y}^{\mu_t})) \right|$$

$$\leq \frac{\Delta^{\mu_t}}{K+1} + \frac{2\Delta^{\mu_t}}{\sqrt{K+1}} + \|\boldsymbol{\lambda}^{\mu_t}\|\sqrt{\frac{\Delta^{\mu_t}}{\beta(K+1)}} + M\|\boldsymbol{x}^{\mu_t} - \boldsymbol{x}^\star\|, \tag{43}$$

*and*

$$\left| F(\boldsymbol{x}^\star)^\mathsf{T}(\boldsymbol{x}_{K+1}^{(t)} - \boldsymbol{x}^\star) + \mu_t(B(\boldsymbol{y}_{K+1}^{(t)}) - B(\boldsymbol{y}^{\mu_t})) \right|$$

$$\leq \frac{\Delta^{\mu_t}}{2(K+1)} + \frac{2\sqrt{\beta\Delta^{\mu_t}}\|\boldsymbol{\lambda}^{\mu_t}\|}{\beta(K+1)} + m\mu_t \tag{44}$$

$$+ L\|\boldsymbol{x}^{\mu_t} - \boldsymbol{x}^\star\|\left( \|\boldsymbol{x}^{\mu_t} - \boldsymbol{x}^\star\| + \frac{2\sqrt{\beta\Delta^{\mu_t}}}{\beta(K+1)} + \sqrt{\frac{\Delta^{\mu_t}}{\beta}} \right),$$

*where $M = \sup_{\boldsymbol{x} \in \mathcal{C}} \|F(\boldsymbol{x})\|$, and $B(\boldsymbol{x}) = -\sum_{i=1}^m \log(-\varphi_i(\boldsymbol{x}))$.*

*Proof.* Note that

$$\left\| \boldsymbol{x}_{K+1}^{(t)} - \boldsymbol{x}^{\mu_t} \right\| = \left\| \boldsymbol{x}_{K+1}^{(t)} - \boldsymbol{y}^{\mu_t} \right\| \leq \left\| \boldsymbol{x}_{K+1}^{(t)} - \boldsymbol{y}_{K+1}^{(t)} \right\| + \left\| \boldsymbol{y}_{K+1}^{(t)} - \boldsymbol{y}^{\mu_t} \right\|. \tag{45}$$

Recall that (29) gives

$$\|\boldsymbol{x}_{K+1}^{(t)} - \boldsymbol{y}_{K+1}^{(t)}\| \leq \sqrt{\frac{\Delta^{\mu_t}}{\beta(K+1)}},$$

and (34) gives

$$\left\| \boldsymbol{y}_{K+1}^{(t)} - \boldsymbol{y}^{\mu_t} \right\| = \sqrt{\frac{\Delta^{\mu_t}}{\beta}}.$$

Plugging (29) and (34) into (45), we have

$$\left\| \boldsymbol{x}_{K+1}^{(t)} - \boldsymbol{x}^{\mu_t} \right\| \leq \sqrt{\frac{\Delta^{\mu_t}}{\beta(K+1)}} + \sqrt{\frac{\Delta^{\mu_t}}{\beta}}. \tag{46}$$

Note that

$$F(\boldsymbol{x}^\star)^\mathsf{T}(\boldsymbol{x}_{K+1}^{(t)} - \boldsymbol{x}^\star) + \mu_t(B(\boldsymbol{y}_{K+1}^{(t)}) - B(\boldsymbol{y}^{\mu_t}))$$

$$= F(\boldsymbol{x}^{\mu_t})^\mathsf{T}(\boldsymbol{x}_{K+1}^{(t)} - \boldsymbol{x}_t^\mu) + \mu_t(B(\boldsymbol{y}_{K+1}^{(t)}) - B(\boldsymbol{y}^{\mu_t})) + F(\boldsymbol{x}^\mu)^\mathsf{T}(\boldsymbol{x}^\mu - \boldsymbol{x}^\star)$$

$$+ (F(\boldsymbol{x}^\star) - F(\boldsymbol{x}^{\mu_t}))^\mathsf{T}(\boldsymbol{x}_{K+1}^{(t)} - \boldsymbol{x}^\star).$$

Thus by the $L$-Lipschitzness of $F$, using (40), (46) and (37) in the proof of Thm. 9, we have

$$
\begin{aligned}
& \left| F(\boldsymbol{x}^\star)^\intercal (\boldsymbol{x}^{(t)}_{K+1} - \boldsymbol{x}^\star) + \mu_t (B(\boldsymbol{y}^{(t)}_{K+1}) - B(\boldsymbol{y}^{\mu_t})) \right| \\
& \leq \left| F(\boldsymbol{x}^{\mu_t})^\intercal (\boldsymbol{x}^{(t)}_{K+1} - \boldsymbol{x}^{\mu_t}) + \mu_t (B(\boldsymbol{y}^{(t)}_{K+1}) - B(\boldsymbol{y}^{\mu_t})) \right| + |F(\boldsymbol{x}^\mu_t)^\intercal (\boldsymbol{x}^\mu_t - \boldsymbol{x}^\star)| \\
& \quad + \left| (F(\boldsymbol{x}^\star) - F(\boldsymbol{x}^{\mu_t}))^\intercal (\boldsymbol{x}^{(t)}_{K+1} - \boldsymbol{x}^\star) \right| \\
& \leq \left| F(\boldsymbol{x}^{\mu_t})^\intercal (\boldsymbol{x}^{(t)}_{K+1} - \boldsymbol{x}^{\mu_t}) + \mu_t (B(\boldsymbol{y}^{(t)}_{K+1}) - B(\boldsymbol{y}^{\mu_t})) \right| + |F(\boldsymbol{x}^{\mu_t})^\intercal (\boldsymbol{x}^{\mu_t} - \boldsymbol{x}^\star)| \\
& \quad + L \left\| \boldsymbol{x}^{\mu_t} - \boldsymbol{x}^\star \right\| \left( \left\| \boldsymbol{x}^{\mu_t} - \boldsymbol{x}^{(t)}_{K+1} \right\| + \left\| \boldsymbol{x}^{\mu_t} - \boldsymbol{x}^\star \right\| \right) \\
& \leq \frac{\Delta^{\mu_t}}{K+1} + \frac{2\Delta^{\mu_t}}{\sqrt{K+1}} + \left\| \boldsymbol{\lambda}^{\mu_t} \right\| \sqrt{\frac{\Delta^{\mu_t}}{\beta (K+1)}} + m\mu_t \\
& \quad + L \left\| \boldsymbol{x}^{\mu_t} - \boldsymbol{x}^\star \right\| \left( \left\| \boldsymbol{x}^{\mu_t} - \boldsymbol{x}^\star \right\| + \sqrt{\frac{\Delta^{\mu_t}}{\beta (K+1)}} + \sqrt{\frac{\Delta^{\mu_t}}{\beta}} \right).
\end{aligned}
$$

Using (38), we have

$$
\begin{aligned}
& \left| F(\boldsymbol{x}^{(t)}_{K+1})^\intercal (\boldsymbol{x}^{(t)}_{K+1} - \boldsymbol{x}^\star) + \mu_t (B(\boldsymbol{y}^{(t)}_{K+1}) - B(\boldsymbol{y}^{\mu_t})) \right| \\
& = \left| F(\boldsymbol{x}^{\mu_t})^\intercal (\boldsymbol{x}^{(t)}_{K+1} - \boldsymbol{x}^\star) + \mu_t (B(\boldsymbol{y}^{(t)}_{K+1}) - B(\boldsymbol{y}^{\mu_t})) + F(\boldsymbol{x}^{(t)}_{K+1})^\intercal (\boldsymbol{x}^{\mu_t} - \boldsymbol{x}^\star) \right| \\
& \leq \left| F(\boldsymbol{x}^{\mu_t})^\intercal (\boldsymbol{x}^{(t)}_{K+1} - \boldsymbol{x}^\star) + \mu_t (B(\boldsymbol{y}^{(t)}_{K+1}) - B(\boldsymbol{y}^{\mu_t})) \right| + M \left\| \boldsymbol{x}^{\mu_t} - \boldsymbol{x}^\star \right\| \\
& \leq \frac{\Delta^{\mu_t}}{K+1} + \frac{2\Delta^{\mu_t}}{\sqrt{K+1}} + \left\| \boldsymbol{\lambda}^{\mu_t} \right\| \sqrt{\frac{\Delta^{\mu_t}}{\beta (K+1)}} + M \left\| \boldsymbol{x}^{\mu_t} - \boldsymbol{x}^\star \right\|.
\end{aligned}
$$

Similarly, recall that (36) gives

$$
\left\| \hat{\boldsymbol{x}}_{K+1} - \hat{\boldsymbol{y}}_{K+1} \right\| \leq \frac{2\sqrt{\beta \Delta^{\mu_t}}}{\beta (K+1)}.
$$

By (34) and the convexity of $\left\| \cdot \right\|$, we have

$$
\left\| \hat{\boldsymbol{y}}^{(t)}_{K+1} - \boldsymbol{y}^{\mu_t} \right\| = \left\| \frac{1}{K+1} \sum_{k=1}^{K+1} (\boldsymbol{y}^{(t)}_{k+1} - \boldsymbol{y}^{\mu_t}) \right\| \leq \sqrt{\frac{\Delta^{\mu_t}}{\beta}}. \tag{47}
$$

Note that Thus we have

$$
\left\| \hat{\boldsymbol{x}}^{(t)}_{K+1} - \boldsymbol{x}^{\mu_t} \right\| = \left\| \hat{\boldsymbol{x}}^{(t)}_{K+1} - \boldsymbol{y}^{\mu_t} \right\| \leq \left\| \hat{\boldsymbol{x}}^{(t)}_{K+1} - \hat{\boldsymbol{y}}^{(t)}_{K+1} \right\| + \left\| \hat{\boldsymbol{y}}^{(t)}_{K+1} - \boldsymbol{y}^{\mu_t} \right\|. \tag{48}
$$

Plugging (36) and (47) into (48), we have

$$
\left\| \hat{\boldsymbol{x}}^{(t)}_{K+1} - \boldsymbol{x}^{\mu_t} \right\| \leq \frac{2\sqrt{\beta \Delta^{\mu_t}}}{\beta (K+1)} + \sqrt{\frac{\Delta^{\mu_t}}{\beta}}. \tag{49}
$$

Using the $L$-Lipschitzness of $F$, (39) and (49), we have

$$
\begin{aligned}
&\left| F(\boldsymbol{x}^\star)^\intercal (\hat{\boldsymbol{x}}^{(t)}_{K+1} - \boldsymbol{x}^\star) + \mu_t (B(\hat{\boldsymbol{y}}^{(t)}_{K+1}) - B(\boldsymbol{y}^{\mu_t})) \right| \\
&\leq \left| F(\boldsymbol{x}^{\mu_t})^\intercal (\hat{\boldsymbol{x}}^{(t)}_{K+1} - \boldsymbol{x}^{\mu_t}) + \mu_t (B(\hat{\boldsymbol{y}}^{(t)}_{K+1}) - B(\boldsymbol{y}^{\mu_t})) \right| + |F(\boldsymbol{x}^{\mu_t})^\intercal (\boldsymbol{x}^{\mu_t} - \boldsymbol{x}^\star)| \\
&\quad + \left| (F(\boldsymbol{x}^\star) - F(\boldsymbol{x}^{\mu_t}))^\intercal (\hat{\boldsymbol{x}}^{(t)}_{K+1} - \boldsymbol{x}^\star) \right| \\
&\leq \left| F(\boldsymbol{x}^{\mu_t})^\intercal (\hat{\boldsymbol{x}}^{(t)}_{K+1} - \boldsymbol{x}^\mu_t) + \mu_t (B(\hat{\boldsymbol{y}}^{(t)}_{K+1}) - B(\boldsymbol{y}^{\mu_t})) \right| + |F(\boldsymbol{x}^{\mu_t})^\intercal (\boldsymbol{x}^{\mu_t} - \boldsymbol{x}^\star)| \\
&\quad + L \left\| \boldsymbol{x}^{\mu_t} - \boldsymbol{x}^\star \right\| \left( \left\| \boldsymbol{x}^{\mu_t} - \hat{\boldsymbol{x}}^{(t)}_{K+1} \right\| + \| \boldsymbol{x}^{\mu_t} - \boldsymbol{x}^\star \| \right) \\
&\leq \frac{\Delta^{\mu_t}}{2(K+1)} + \frac{2\sqrt{\beta \Delta^{\mu_t}} \| \boldsymbol{\lambda}^{\mu_t} \|}{\beta(K+1)} + m\mu_t \\
&\quad + L \| \boldsymbol{x}^{\mu_t} - \boldsymbol{x}^\star \| \left( \| \boldsymbol{x}^{\mu_t} - \boldsymbol{x}^\star \| + \frac{2\sqrt{\beta \Delta^{\mu_t}}}{\beta(K+1)} + \sqrt{\frac{\Delta^{\mu_t}}{\beta}} \right).
\end{aligned}
$$

$\square$

Now we are ready to give our main theorems in this section.

**Theorem 12** (Complete convergence rate for star-$\xi$-monotone operator). *Given an operator $F : \mathcal{X} \to \mathbb{R}^n$ monotone and $L$-Lipschitz on $\mathcal{C}_=$ (Def. 1, 1), and either $F$ is strictly monotone on $\mathcal{C}$ or one of $\varphi_i$ is strictly convex. Assume there exists $r > 0$ or $s > 0$ such that $F$ is star-$\xi$-monotone—as per Def. 1—on either $\hat{\mathcal{C}}_r$ or $\tilde{\mathcal{C}}_s$, resp. Let $\Delta^{\mu_t} \triangleq \frac{1}{\beta} \| \boldsymbol{\lambda}_0 - \boldsymbol{\lambda}^{\mu_t} \|^2 + \beta \| \boldsymbol{y}_0 - \boldsymbol{y}^{\mu_t} \|^2$,*

*Let $\boldsymbol{x}^{(t)}_K$ and $\hat{\boldsymbol{x}}^{(t)}_K \triangleq \frac{1}{K} \sum_{k=1}^K \boldsymbol{x}^{(t)}_k$ denote the last and average iterate of Algorithm 1, respectively, run with sufficiently small $\mu_{t-1}$. Then there exists $K_0 \in \mathbb{N}$, $K_0$ depends on $r$ or $s$, s.t. $\forall K > K_0$, for all $t \in [T]$, we have that:*

1. $$
\left\| \boldsymbol{x}^{(t)}_K - \boldsymbol{x}^\star \right\| \leq \left( \frac{2}{c} \left( \frac{\Delta^{\mu_t}}{K} + \frac{2\Delta^{\mu_t}}{\sqrt{K}} + \| \boldsymbol{\lambda}^{\mu_t} \| \sqrt{\frac{\Delta^{\mu_t}}{\beta K}} \right) + \frac{LM_1}{c} \left( \frac{m}{c} \mu_{-1} \right)^{\frac{1}{\xi}} \delta^{\frac{t+1}{\xi}} \right)^{\frac{1}{\xi}}, \quad \text{and}
$$
$$
\mathcal{G}(\boldsymbol{x}^{(t)}_K, \mathcal{C}) \leq M_0 \left( \frac{2}{c} \left( \frac{\Delta^{\mu_t}}{K} + \frac{2\Delta^{\mu_t}}{\sqrt{K}} + \| \boldsymbol{\lambda}^{\mu_t} \| \sqrt{\frac{\Delta^{\mu_t}}{\beta K}} \right) + \frac{LM_1}{c} \left( \frac{m}{c} \mu_{-1} \right)^{\frac{1}{\xi}} \delta^{\frac{t+1}{\xi}} \right)^{\frac{1}{\xi}}.
$$

2. *If in addition $F$ is $\xi$-monotone on $C_=$, we have $\| \hat{\boldsymbol{x}}^{(t)}_K - \boldsymbol{x}^\star \| \leq \left( \frac{\Delta^{\mu_t}}{cK} \right)^{\frac{1}{\xi}} + \left( \frac{m}{c} \mu_t \right)^{\frac{1}{\xi}}$, and*
$$
\mathcal{G}(\hat{\boldsymbol{x}}^{(t)}_K, \mathcal{C}) \leq M_0 \left( \left( \frac{\Delta^{\mu_t}}{cK} \right)^{1/\xi} + \left( \frac{m}{c} \mu_t \right)^{\frac{1}{\xi}} \right)^{\frac{1}{\xi}},
$$

*where $M_0 = DL + M$ is a linear function of $L$, see the proof of Lemma 1 in App. B, and $M_1 \triangleq \left( \frac{m}{c} \mu_{-1} \right)^{\frac{1}{\xi}} + 2\sqrt{\frac{\Delta^{\mu_t}}{\beta}} + \frac{M}{L}$.*

*Proof.* By the star-$\xi$-monotonicity of course $F$, using (42) and (43) in Lemma 10, we have

$$c\left\|\boldsymbol{x}_{K+1}^{(t)} - \boldsymbol{x}^{\star}\right\|^{\xi}$$

$$\leq (F(\boldsymbol{x}_K^{(t)}) - F(\boldsymbol{x}^{\star}))^{\mathsf{T}}(\boldsymbol{x}_{K+1}^{(t)} - \boldsymbol{x}^{\star})$$

$$= F(\boldsymbol{x}_K^{(t)})^{\mathsf{T}}(\boldsymbol{x}_{K+1}^{(t)} - \boldsymbol{x}^{\star}) + \mu_t(B(\boldsymbol{y}_{K+1}^{(t)}) - B(\boldsymbol{y}^{\mu_t}))$$

$$\quad - (F(\boldsymbol{x}^{\star})^{\mathsf{T}}(\boldsymbol{x}_{K+1}^{(t)} - \boldsymbol{x}^{\star}) + \mu_t(B(\boldsymbol{y}_{K+1}^{(t)}) - B(\boldsymbol{y}^{\mu_t})))$$

$$\leq \|F(\boldsymbol{x}_K^{(t)})^{\mathsf{T}}(\boldsymbol{x}_{K+1}^{(t)} - \boldsymbol{x}^{\star}) + \mu_t(B(\boldsymbol{y}_{K+1}^{(t)}) - B(\boldsymbol{y}^{\mu_t}))\|$$

$$\quad + \|F(\boldsymbol{x}^{\star})^{\mathsf{T}}(\boldsymbol{x}_{K+1}^{(t)} - \boldsymbol{x}^{\star}) + \mu_t(B(\boldsymbol{y}_{K+1}^{(t)}) - B(\boldsymbol{y}^{\mu_t}))\|$$

$$\leq \frac{\Delta^{\mu_t}}{K+1} + \frac{2\Delta^{\mu_t}}{\sqrt{K+1}} + \|\boldsymbol{\lambda}^{\mu_t}\|\sqrt{\frac{\Delta^{\mu_t}}{\beta(K+1)}} + M\|\boldsymbol{x}^{\mu_t} - \boldsymbol{x}^{\star}\|$$

$$\quad + \frac{\Delta^{\mu_t}}{K+1} + \frac{2\Delta^{\mu_t}}{\sqrt{K+1}} + \|\boldsymbol{\lambda}^{\mu_t}\|\sqrt{\frac{\Delta^{\mu_t}}{\beta(K+1)}} + m\mu_t$$

$$\quad + L\|\boldsymbol{x}^{\mu_t} - \boldsymbol{x}^{\star}\|\left(\|\boldsymbol{x}^{\mu_t} - \boldsymbol{x}^{\star}\| + \sqrt{\frac{\Delta^{\mu_t}}{\beta(K+1)}} + \sqrt{\frac{\Delta^{\mu_t}}{\beta}}\right)$$

$$\leq 2\left(\frac{\Delta^{\mu_t}}{K+1} + \frac{2\Delta^{\mu_t}}{\sqrt{K+1}} + \|\boldsymbol{\lambda}^{\mu_t}\|\sqrt{\frac{\Delta^{\mu_t}}{\beta(K+1)}}\right)$$

$$\quad + L\|\boldsymbol{x}^{\mu_t} - \boldsymbol{x}^{\star}\|\left(\|\boldsymbol{x}^{\mu_t} - \boldsymbol{x}^{\star}\| + \sqrt{\frac{\Delta^{\mu_t}}{\beta(K+1)}} + \sqrt{\frac{\Delta^{\mu_t}}{\beta}} + \frac{M}{L}\right) + m\mu_t$$

$$\leq 2\left(\frac{\Delta^{\mu_t}}{K+1} + \frac{2\Delta^{\mu_t}}{\sqrt{K+1}} + \|\boldsymbol{\lambda}^{\mu_t}\|\sqrt{\frac{\Delta^{\mu_t}}{\beta(K+1)}}\right)$$

$$\quad + L\left(\frac{m}{c}\mu_t\right)^{\frac{1}{\xi}}\left(\left(\frac{m}{c}\mu_t\right)^{\frac{1}{\xi}} + \sqrt{\frac{\Delta^{\mu_t}}{\beta(K+1)}} + \sqrt{\frac{\Delta^{\mu_t}}{\beta}} + \frac{M}{L}\right)$$

where in the last inequality we use Lemma 9 (i).

We let $M_1 \triangleq \left(\frac{m}{c}\mu_{-1}\right)^{\frac{1}{\xi}} + 2\sqrt{\frac{\Delta^{\mu_t}}{\beta}} + \frac{M}{L}$, then we have

$$\left\|\boldsymbol{x}_{K+1}^{(t)} - \boldsymbol{x}^{\star}\right\| \leq \left(\frac{2}{c}\left(\frac{\Delta^{\mu_t}}{K+1} + \frac{2\Delta^{\mu_t}}{\sqrt{K+1}} + \|\boldsymbol{\lambda}^{\mu_t}\|\sqrt{\frac{\Delta^{\mu_t}}{\beta(K+1)}}\right) + \frac{LM_1}{c}\left(\frac{m}{c}\mu_t\right)^{\frac{1}{\xi}}\right)^{\frac{1}{\xi}}.$$

By Lemma 1, we obtain

$$\mathcal{G}(\boldsymbol{x}_{K+1}^{(t)}, \mathcal{C}) \leq M_0\left(\frac{2}{c}\left(\frac{\Delta^{\mu_t}}{K+1} + \frac{2\Delta^{\mu_t}}{\sqrt{K+1}} + \|\boldsymbol{\lambda}^{\mu_t}\|\sqrt{\frac{\Delta^{\mu_t}}{\beta(K+1)}}\right) + \frac{LM_1}{c}\left(\frac{m}{c}\mu_{-1}\right)^{\frac{1}{\xi}}\delta^{\frac{t+1}{\xi}}\right)^{\frac{1}{\xi}}.$$

If in addition $F$ is $\xi$-monotone on $\mathcal{C}_=$, then from (30) we have:

$$\|\hat{\boldsymbol{x}}_{K+1}^{(t)} - \boldsymbol{x}^{\star}\| \leq \left(\frac{\Delta^{\mu_t}}{c(K+1)}\right)^{\frac{1}{\xi}} + \left(\frac{m}{c}\mu_t\right)^{\frac{1}{\xi}}.$$

Again by Lemma 1, we have $\mathcal{G}(\hat{\boldsymbol{x}}_{K+1}^{(t)}, \mathcal{C}) \leq M_0\left(\left(\frac{\Delta^{\mu_t}}{c(K+1)}\right)^{\frac{1}{\xi}} + \left(\frac{m}{c}\mu_t\right)^{\frac{1}{\xi}}\right).$ □

**Theorem 13** (Complete convergence rate for star-$\xi$-monotone operator)**.** *Given an operator $F$ : $\mathcal{X} \to \mathbb{R}^n$, assume: (i) $F$ is monotone and $L$-smooth on $\mathcal{C}_=$, as per Def. 1, 1; (ii) either $F$ is strictly*

*monotone on $\mathcal{C}$ or one of $\varphi_i$ is strictly convex; and (iii)* $\inf\limits_{\boldsymbol{x}\in S\setminus\{\boldsymbol{x}^\star\}} F(\boldsymbol{x})^\mathsf{T}\frac{\boldsymbol{x}-\boldsymbol{x}^\star}{\|\boldsymbol{x}-\boldsymbol{x}^\star\|} = a > 0$, *where*

$S \equiv \hat{\mathcal{C}}_r$ *or* $\tilde{\mathcal{C}}_s$. *Let* $\Delta^{\mu_t} \triangleq \frac{1}{\beta}\|\boldsymbol{\lambda}_0 - \boldsymbol{\lambda}^{\mu_t}\|^2 + \beta\|\boldsymbol{y}_0 - \boldsymbol{y}^{\mu_t}\|^2$.

*Let* $\boldsymbol{x}_K^{(t)}$ *and* $\hat{\boldsymbol{x}}_K^{(t)} \triangleq \frac{1}{K}\sum_{k=1}^{K}\boldsymbol{x}_k^{(t)}$ *denote the last and average iterate of Algorithm 1, respectively. Then there exists* $K_0 \in \mathbb{N}$, $K_0$ *depends on* $r$ *or* $s$, *s.t.* $\forall t \in [T]$, $\forall K > K_0$, *we have that:*

1. $\left\|\boldsymbol{x}_K^{(t)} - \boldsymbol{x}^\star\right\| = \frac{1}{a}\left(\frac{\Delta^{\mu_t}}{K} + \frac{2\Delta^{\mu_t}}{\sqrt{K}} + \|\boldsymbol{\lambda}^{\mu_t}\|\sqrt{\frac{\Delta^{\mu_t}}{\beta K}}\right) + \left(\frac{mM}{a^2} + \frac{B(\boldsymbol{y}^{\mu_t}) - B^\star}{a}\right)\mu_{-1}\delta^{t+1}$, *and*

   $\mathcal{G}(\boldsymbol{x}_K^{(t)}, \mathcal{C}) \le M_0\left(\frac{1}{a}\left(\frac{\Delta^{\mu_t}}{K} + \frac{2\Delta^{\mu_t}}{\sqrt{K}} + \|\boldsymbol{\lambda}^{\mu_t}\|\sqrt{\frac{\Delta^{\mu_t}}{\beta K}}\right) + \left(\frac{mM}{a^2} + \frac{B(\boldsymbol{y}^{\mu_t}) - B^\star}{a}\right)\mu_{-1}\delta^{t+1}\right)$.

2. *If in addition* $\inf\limits_{\boldsymbol{x}\in S\setminus\{\boldsymbol{x}^\star\}} F(\boldsymbol{x}^\star)^\mathsf{T}\frac{\boldsymbol{x}-\boldsymbol{x}^\star}{\|\boldsymbol{x}-\boldsymbol{x}^\star\|} = b > 0$ *(with* $S \equiv \hat{\mathcal{C}}_r$ *or* $\tilde{\mathcal{C}}_s$*),*

   *then* $\left\|\hat{\boldsymbol{x}}_K^{(t)} - \boldsymbol{x}^\star\right\| \le \frac{1}{b}\left(\frac{\Delta^{\mu_t}}{2K} + \frac{2\sqrt{\beta\Delta^{\mu_t}}\|\boldsymbol{\lambda}^{\mu_t}\|}{\beta(K)}\right) + \frac{M_3}{b}\mu_{-1}\delta^{t+1}$, *and* $\mathcal{G}(\hat{\boldsymbol{x}}_K^{(t)}, \mathcal{C}) \le$
   $M_0\left(\frac{1}{b}\left(\frac{\Delta^{\mu_t}}{2K} + \frac{2\sqrt{\beta\Delta^{\mu_t}}\|\boldsymbol{\lambda}^{\mu_t}\|}{\beta(K)}\right) + \frac{M_3}{b}\mu_{-1}\delta^{t+1}\right)$,

*where* $M = \sup_{\boldsymbol{x}\in\mathcal{C}}\|F(\boldsymbol{x})\|$, $M_0 = DL + M$ *is a linear function of* $L$, *see the proof of Lemma 1 in App. B, and* $M_3 \triangleq m + \frac{Lm}{a}\left(\frac{m}{a}\mu_{-1} + \frac{2\sqrt{\beta\Delta^{\mu_t}}}{\beta(K+1)} + \sqrt{\frac{\Delta^{\mu_t}}{\beta}}\right) + B(\boldsymbol{y}^{\mu_t}) - B^\star$.

*Proof.* If the barrier term $B(\boldsymbol{y}_{K+1}^{(t)}) \ge B(\boldsymbol{y}^{\mu_t})$, using (43) in Lemma 10, we have

$$F(\boldsymbol{x}_K^{(t)})^\mathsf{T}(\boldsymbol{x}_{K+1}^{(t)} - \boldsymbol{x}^\star)$$
$$\le F(\boldsymbol{x}_K^{(t)})^\mathsf{T}(\boldsymbol{x}_{K+1}^{(t)} - \boldsymbol{x}^\star) + \mu_t(B(\boldsymbol{y}_{K+1}^{(t)}) - B(\boldsymbol{y}^{\mu_t}))$$
$$\le \frac{\Delta^{\mu_t}}{K+1} + \frac{2\Delta^{\mu_t}}{\sqrt{K+1}} + \|\boldsymbol{\lambda}^{\mu_t}\|\sqrt{\frac{\Delta^{\mu_t}}{\beta(K+1)}} + M\|\boldsymbol{x}^{\mu_t} - \boldsymbol{x}^\star\|.$$

If $B(\boldsymbol{y}_{K+1}^{(t)}) \le B(\boldsymbol{y}^{\mu_t})$, since $B$ is lower bounded in any compact set, and by (46) in Lemma 9 we have $\left\|\hat{\boldsymbol{x}}_{K+1}^{(t)} - \boldsymbol{x}^{\mu_t}\right\| \le \frac{2\sqrt{\beta\Delta^{\mu_t}}}{\beta(K+1)} + \sqrt{\frac{\Delta^{\mu_t}}{\beta}} \le 3\sqrt{\frac{\Delta^{\mu_t}}{\beta}} \triangleq M_2$, we let $B^\star = \inf_{\|\boldsymbol{x}-\boldsymbol{x}^\star\|\le M_2} B(\boldsymbol{x})$. Then we have

$$F(\boldsymbol{x}_K^{(t)})^\mathsf{T}(\boldsymbol{x}_{K+1}^{(t)} - \boldsymbol{x}^\star)$$
$$\le \left|F(\boldsymbol{x}_K^{(t)})^\mathsf{T}(\boldsymbol{x}_{K+1}^{(t)} - \boldsymbol{x}^\star) + \mu_t(B(\boldsymbol{y}_{K+1}^{(t)}) - B(\boldsymbol{y}^{\mu_t}))\right| + \mu_t(B(\boldsymbol{y}^{\mu_t}) - B^\star)$$
$$\le \frac{\Delta^{\mu_t}}{K+1} + \frac{2\Delta^{\mu_t}}{\sqrt{K+1}} + \|\boldsymbol{\lambda}^{\mu_t}\|\sqrt{\frac{\Delta^{\mu_t}}{\beta(K+1)}} + M\|\boldsymbol{x}^{\mu_t} - \boldsymbol{x}^\star\| + \mu_t(B(\boldsymbol{y}^{\mu_t}) - B^\star).$$

Thus in both case we have

$$a\left\|\boldsymbol{x}_{K+1}^{(t)} - \boldsymbol{x}^\star\right\|$$
$$\le F(\boldsymbol{x}_K^{(t)})^\mathsf{T}(\boldsymbol{x}_{K+1}^{(t)} - \boldsymbol{x}^\star)$$
$$\le \frac{\Delta^{\mu_t}}{K+1} + \frac{2\Delta^{\mu_t}}{\sqrt{K+1}} + \|\boldsymbol{\lambda}^{\mu_t}\|\sqrt{\frac{\Delta^{\mu_t}}{\beta(K+1)}} + \frac{mM}{a}\mu_t + \mu_t(B(\boldsymbol{y}^{\mu_t}) - B^\star).$$

where in the first inequality assumption (iii) is used and in the last inequality Lemma 9 is used. Thus we have:

$$\left\|\boldsymbol{x}_{K+1}^{(t)} - \boldsymbol{x}^\star\right\| = \frac{1}{a}\left(\frac{\Delta^{\mu_t}}{K+1} + \frac{2\Delta^{\mu_t}}{\sqrt{K+1}} + \|\boldsymbol{\lambda}^{\mu_t}\|\sqrt{\frac{\Delta^{\mu_t}}{\beta(K+1)}}\right) + \left(\frac{mM}{a^2} + \frac{B(\boldsymbol{y}^{\mu_t}) - B^\star}{a}\right)\mu_t.$$

And by Lemma 1, we obtain

$$\mathcal{G}(\boldsymbol{x}_{K+1}^{(t)}, \mathcal{C}) \le M_0\left(\frac{1}{a}\left(\frac{\Delta^{\mu_t}}{K+1} + \frac{2\Delta^{\mu_t}}{\sqrt{K+1}} + \|\boldsymbol{\lambda}^{\mu_t}\|\sqrt{\frac{\Delta^{\mu_t}}{\beta(K+1)}}\right) + \left(\frac{mM}{a^2} + \frac{B(\boldsymbol{y}^{\mu_t}) - B^\star}{a}\right)\mu_t\right).$$

If in addition $\inf\limits_{\boldsymbol{x}\in S\setminus\{\boldsymbol{x}^\star\}} F(\boldsymbol{x}^\star)^\mathsf{T}\frac{\boldsymbol{x}-\boldsymbol{x}^\star}{\|\boldsymbol{x}-\boldsymbol{x}^\star\|} = b > 0$, then similarly, from (44) in Lemma 10, we have

$$
b\left\|\hat{\boldsymbol{x}}_{K+1}^{(t)} - \boldsymbol{x}^\star\right\|
$$

$$
\leq \left| F(\boldsymbol{x}^\star)^\mathsf{T}(\boldsymbol{x}_{K+1}^{(t)} - \boldsymbol{x}^\star) + \mu_t(B(\boldsymbol{y}_{K+1}^{(t)}) - B(\boldsymbol{y}^{\mu_t})) \right| + \mu_t(B(\boldsymbol{y}^{\mu_t}) - B^\star)
$$

$$
\leq \frac{\Delta^{\mu_t}}{2\,(K+1)} + \frac{2\sqrt{\beta\Delta^{\mu_t}}\|\boldsymbol{\lambda}^{\mu_t}\|}{\beta\,(K+1)} + m\mu_t
$$

$$
+ L\frac{m}{a}\mu_t\left(\frac{m}{a}\mu_t + \frac{2\sqrt{\beta\Delta^{\mu_t}}}{\beta\,(K+1)} + \sqrt{\frac{\Delta^{\mu_t}}{\beta}}\right) + \mu_t(B(\boldsymbol{y}^{\mu_t}) - B^\star)
$$

$$
= \frac{\Delta^{\mu_t}}{2\,(K+1)} + \frac{2\sqrt{\beta\Delta^{\mu_t}}\|\boldsymbol{\lambda}^{\mu_t}\|}{\beta\,(K+1)}
$$

$$
+ \left(m + \frac{Lm}{a}\left(\frac{m}{a}\mu_{-1} + \frac{2\sqrt{\beta\Delta^{\mu_t}}}{\beta\,(K+1)} + \sqrt{\frac{\Delta^{\mu_t}}{\beta}}\right) + B(\boldsymbol{y}^{\mu_t}) - B^\star\right)\mu_t\,,
$$

Let $M_3 \triangleq m + \frac{Lm}{a}\left(\frac{m}{a}\mu_{-1} + \frac{2\sqrt{\beta\Delta^{\mu_t}}}{\beta(K+1)} + \sqrt{\frac{\Delta^{\mu_t}}{\beta}}\right) + B(\boldsymbol{y}^{\mu_t}) - B^\star$, then we have

$$
\left\|\hat{\boldsymbol{x}}_{K+1}^{(t)} - \boldsymbol{x}^\star\right\| \leq \frac{1}{b}\left(\frac{\Delta^{\mu_t}}{2\,(K+1)} + \frac{2\sqrt{\beta\Delta^{\mu_t}}\|\boldsymbol{\lambda}^{\mu_t}\|}{\beta\,(K+1)}\right) + \frac{M_3}{b}\mu_t\,.
$$

By Lemma 1, we have

$$
\mathcal{G}(\boldsymbol{x}_{K+1}^{(t)}, \mathcal{C}) \leq \frac{1}{b}\left(\frac{\Delta^{\mu_t}}{2\,(K+1)} + \frac{2\sqrt{\beta\Delta^{\mu_t}}\|\boldsymbol{\lambda}^{\mu_t}\|}{\beta\,(K+1)}\right) + \frac{M_3}{b}\mu_t\,.
$$

$\square$

**Remark 5.** *In Theorem 12, for any $t \geq \mathcal{O}(\ln K)$, we have $\left\|\boldsymbol{x}_K^{(t)} - \boldsymbol{x}^\star\right\|, \mathcal{G}(\boldsymbol{x}_K^{(t)}, \mathcal{C}) \leq \mathcal{O}(\frac{1}{K^{1/(2\xi)}})$, and $\left\|\hat{\boldsymbol{x}}_K^{(t)} - \boldsymbol{x}^\star\right\|, \mathcal{G}(\hat{\boldsymbol{x}}_K^{(t)}, \mathcal{C}) \leq \mathcal{O}(\frac{1}{K^{1/\xi}})$. Similarly, in Theorem 13, for any $t \geq \mathcal{O}(\ln K)$, we have $\left\|\boldsymbol{x}_K^{(t)} - \boldsymbol{x}^\star\right\|, \mathcal{G}(\boldsymbol{x}_K^{(t)}, \mathcal{C}) \leq \mathcal{O}(\frac{1}{\sqrt{K}})$, and $\left\|\hat{\boldsymbol{x}}_K^{(t)} - \boldsymbol{x}^\star\right\|, \mathcal{G}(\hat{\boldsymbol{x}}_K^{(t)}, \mathcal{C}) \leq \mathcal{O}(\frac{1}{K})$.*

### B.5 DISCUSSION ON ALGORITHM 1

**Advantages and disadvantages of Algorithm 1.** In Algorithm 1, the update of $\boldsymbol{x}$ (step 8) requires solving (W-EQ). Our method is especially suitable for problems where (W-EQ) is easy to solve analytically. This includes the class of affine variational inequalities, low-dimensional problems, and when optimization variables represent probabilities, for example.

For problems where (W-EQ) is hard to solve—for example, min-max optimization problems in GANs—one could use other *unconstrained* methods like GDA or EG methods (*without* projection) so as to solve $VI(\mathbb{R}^n, G)$, where $G$ is defined in (3). Algorithm 4 describes ACVI when using an unconstrained solver for the inner problems. We observe that Algorithm 4 outperforms the projection-based baseline algorithms, see for example Fig. 4.

Note that when there are constraints, the projection-based methods such as GDA, EG, OGDA etc. require solving a quadratic programming problem in each iteration (or twice per iteration for EG). This problem is often nontrivial and solving it may require using an interior point method or variants (such as the Frank-Wolfe algorithm). Because of this, Algorithm 1 can be seen as an orthogonal approach to projection-based methods, or in other words, as a complementary tool to solve (cVI) and particularly relevant when the constraints are non-trivial.

**ACVI with only equality or inequality constraints.** If there are no equality constraints, then $\mathcal{C}_=$ becomes $\mathbb{R}^n$. In this case, we have that $\boldsymbol{x}_{k+1}^{(t)}$ is the solution of:

$$
\boldsymbol{x} + \frac{1}{\beta}(F(\boldsymbol{x}) + \boldsymbol{\lambda}_k^{(t)}) - \boldsymbol{y}_k^{(t)} = \boldsymbol{0}\,.
$$

When there are no inequality constraints, we let $\boldsymbol{y}_k = \boldsymbol{x}_k$ and $\boldsymbol{\lambda}_k = \boldsymbol{0}$ for every $k$, and we can remove the outer loop. Thus, Algorithm 1 can be simplified to update only one variable $\boldsymbol{x}$ each iteration with the following updating rule:

$$\boldsymbol{x}_{k+1} \quad \textit{is the unique solution of} \quad \boldsymbol{x} + \frac{1}{\beta}\boldsymbol{P}_c F(\boldsymbol{x}) - \boldsymbol{P}_c \boldsymbol{x}_k - \boldsymbol{d}_c = \boldsymbol{0} \,.$$

## C  VARIANT OF THE ACVI ALGORITHM (V-ACVI)

The presented approach of combining interior point methods with ADMM can be used as a framework to derive additional algorithms that may be more suitable for some specific problems. More precisely, observe from Eq. 1 that we could also consider a different splitting than that in § 4. Following this approach, we present a variant of Algorithm 1 and discuss its advantages and disadvantages relative to Algorithm 1.

### C.1  INTRODUCTION OF THE VARIANT ACVI

**Deriving the v-ACVI algorithm.** By considering a different splitting in (1) we get:

$$
\begin{cases}
\min_{\boldsymbol{x},\boldsymbol{y}} F(\boldsymbol{w})^\intercal \boldsymbol{x} - \mu \sum_{i=1}^{m} \log\big(-\varphi_i(\boldsymbol{x})\big) + \mathbb{1}[\boldsymbol{C}\boldsymbol{y} = \boldsymbol{d}] \\
s.t. \qquad \boldsymbol{x} = \boldsymbol{y}
\end{cases},
$$

$$
\text{where:} \quad \mathbb{1}[\boldsymbol{C}\boldsymbol{y} = \boldsymbol{d}] = \begin{cases} 0, & \text{if } \boldsymbol{C}\boldsymbol{y} = \boldsymbol{d} \\ +\infty, & \text{if } \boldsymbol{C}\boldsymbol{y} \neq \boldsymbol{d} \end{cases}.
$$

$$\tag{50}$$

The augmented Lagrangian of (50) is thus:

$$
\mathcal{L}_\beta(\boldsymbol{x},\boldsymbol{y},\boldsymbol{\lambda}) = F(\boldsymbol{w})^\intercal \boldsymbol{x} - \mu \sum_{i=1}^{m} \log(-\varphi_i(\boldsymbol{x})) + \mathbb{1}(\boldsymbol{C}\boldsymbol{y} = \boldsymbol{d}) + \langle \boldsymbol{\lambda}, \boldsymbol{x} - \boldsymbol{y} \rangle + \frac{\beta}{2} \|\boldsymbol{x} - \boldsymbol{y}\|^2
$$

$$\tag{AL-CVI}$$

where $\beta > 0$ is the penalty parameter. We have that for $\boldsymbol{x}$ at step $k+1$:

$$
\begin{aligned}
\boldsymbol{x}_{k+1} &= \arg\min_{\boldsymbol{x}} \mathcal{L}_\beta(\boldsymbol{x}, \boldsymbol{y}_k, \boldsymbol{\lambda}_k) \\
&= \arg\min_{\boldsymbol{x}} \frac{1}{2} \left\| \boldsymbol{x} - \boldsymbol{y}_k + \frac{1}{\beta}(F(\boldsymbol{w}) + \boldsymbol{\lambda}_k) \right\|^2 - \frac{\mu}{\beta} \sum_{i=1}^{m} \log(-\varphi_i(\boldsymbol{x}))
\end{aligned}
$$

$$\tag{51}$$

The following proposition ensures the existence and uniqueness of $x_{k+1}$ in $\mathcal{C}_<$. i.e. We show that $\boldsymbol{x}_{k+1}$ is the unique solution in $\mathcal{C}_<$ of the following closed-form equation (see App. C.2 for its proof):

$$
\boldsymbol{x} - \boldsymbol{y}_k + \frac{1}{\beta}(F(\boldsymbol{w}) + \boldsymbol{\lambda}_k) - \frac{\mu}{\beta} \sum_{i=1}^{m} \frac{\nabla \varphi_i(\boldsymbol{x})}{\varphi_i(\boldsymbol{x})} = \boldsymbol{0}.
$$

$$\tag{X-CF}$$

**Proposition 3** (unique solution). *The problem (X-CF) has a solution in $\mathcal{C}_<$ and the solution is unique.*

For $\boldsymbol{y}$, the updating rule is

$$
\begin{aligned}
\boldsymbol{y}_{k+1} &= \arg\min_{\boldsymbol{y}} \mathcal{L}_\beta(\boldsymbol{x}_{k+1}, \boldsymbol{y}, \boldsymbol{\lambda}_k) \\
&= \arg\min_{\boldsymbol{y} \in \mathcal{C}_=} -\frac{1}{\beta}(\boldsymbol{\lambda}_k)^\intercal \boldsymbol{y} + \frac{1}{2}\|\boldsymbol{y} - \boldsymbol{x}_{k+1}\|_2^2 \\
&= \arg\min_{\boldsymbol{y} \in \mathcal{C}_=} \frac{1}{2}\|\boldsymbol{y} - \boldsymbol{x}_{k+1} - \frac{1}{\beta}\boldsymbol{\lambda}_k\|_2^2 \\
&= \boldsymbol{P}_c(\boldsymbol{x}_{k+1} + \frac{\boldsymbol{\lambda}_k}{\beta}) + \boldsymbol{d}_c
\end{aligned}
$$

$$\tag{y}$$

And the updating rule for dual variable $\boldsymbol{\lambda}$ is

$$
\boldsymbol{\lambda}_{k+1} = \boldsymbol{\lambda}_k + \beta(\boldsymbol{x}_{k+1} - \boldsymbol{y}_{k+1})
$$

$$\tag{$\lambda$}$$

As in § 4, we would like to choose $\boldsymbol{w}_k$ so that $\boldsymbol{w}_k = \boldsymbol{x}_{k+1}$. To this end, we need the following assumption:

**Assumption 2.** $\forall \boldsymbol{b} \in \mathbb{R}^n$ and $\mu > 0$, $\boldsymbol{x} + \frac{1}{\beta}F(\boldsymbol{x}) - \frac{\mu}{\beta}\sum_{i=1}^{m} \frac{\nabla \varphi_i(\boldsymbol{x})}{\varphi(\boldsymbol{x})} + \boldsymbol{b} = \boldsymbol{0}$ *has a solution in $\mathcal{C}_<$.*

If Assumption 2 holds true, we can let $\boldsymbol{w}_k$ be a solution of

$$\boldsymbol{w} - \boldsymbol{y}_{k+1} + \frac{1}{\beta}\big(F(\boldsymbol{w}) + \boldsymbol{\lambda}_{k+1}\big) - \frac{\mu}{\beta}\sum_{i=1}^{m}\frac{\nabla\varphi_i(\boldsymbol{w})}{\varphi_i(\boldsymbol{w})} = \boldsymbol{0} \tag{52}$$

in $\mathcal{C}_<$. And by Proposition 3, we can let $\boldsymbol{x}_{k+1}$ be the unique solution of

$$\boldsymbol{x} - \boldsymbol{y}_k + \frac{1}{\beta}\big(F(\boldsymbol{w}_k) + \boldsymbol{\lambda}_k\big) - \frac{\mu}{\beta}\sum_{i=1}^{m}\frac{\nabla\varphi_i(\boldsymbol{x})}{\varphi_i(\boldsymbol{x})} = \boldsymbol{0} \tag{$\boldsymbol{x}$}$$

in $\mathcal{C}_<$.

Note that $\boldsymbol{w}_k$ is also a solution of ($\boldsymbol{x}$). By the uniqueness of the solution of ($\boldsymbol{x}$) shown in Prop. 3, we know that $\boldsymbol{w}_k = \boldsymbol{x}_{k+1}$.

So in the $(k+1)$-th step, we can compute $\boldsymbol{x}, \boldsymbol{y}, \boldsymbol{\lambda}$ and $\boldsymbol{w}$ by ($\boldsymbol{x}$), ($\boldsymbol{y}$), ($\boldsymbol{\lambda}$) and (52), respectively. Since $\boldsymbol{w}_k = \boldsymbol{x}_{k+1}$, we can simplify our algorithm by removing variable $\boldsymbol{w}$ and only keep $\boldsymbol{x}, \boldsymbol{y}$ and $\boldsymbol{\lambda}$, see Algorithm 3.

---

**Algorithm 3** v-ACVI pseudocode.

---

1: **Input:** operator $F : \mathcal{X} \to \mathbb{R}^n$, equality $\boldsymbol{C}\boldsymbol{x} = \boldsymbol{d}$ and inequality constraints $\varphi_i(\boldsymbol{x}) \leq 0, i = [m]$, hyperparameters $\mu_{-1}, \beta > 0, \delta \in (0,1)$, number of outer and inner loop iterations $T$ and $K$, resp.

2: **Initialize:** $\boldsymbol{y}_0^{(0)} \in \mathbb{R}^n, \boldsymbol{\lambda}_0^{(0)} \in \mathbb{R}^n$

3: $\boldsymbol{P}_c \triangleq \boldsymbol{I} - \boldsymbol{C}^\intercal(\boldsymbol{C}\boldsymbol{C}^\intercal)^{-1}\boldsymbol{C}$          *where $\boldsymbol{P}_c \in \mathbb{R}^{n\times n}$*

4: $\boldsymbol{d}_c \triangleq \boldsymbol{C}^\intercal(\boldsymbol{C}\boldsymbol{C}^\intercal)^{-1}\boldsymbol{d}$          *where $\boldsymbol{d}_c \in \mathbb{R}^n$*

5: **for** $t = 0, \ldots, T-1$ **do**

6:      $\mu_t = \delta\mu_t$

7:      Denote $\hat{\varphi}(\boldsymbol{\lambda}, \boldsymbol{y})$ as the solution of $\frac{1}{\beta}\big(\mu_t \sum_{i=1}^m \frac{\nabla\varphi_i(\boldsymbol{x})}{\varphi_i(\boldsymbol{x})} - F(\boldsymbol{x}) - \boldsymbol{\lambda}\big) + \boldsymbol{y} - \boldsymbol{x} = 0$ with respect to $\boldsymbol{x}$, where $\boldsymbol{y}, \boldsymbol{\lambda}$ are variables

8:      **for** $k = 0, \ldots, K-1$ **do**

9:          $\boldsymbol{x}_{k+1}^{(t)} = \hat{\varphi}(\boldsymbol{\lambda}_k^{(t)}, \boldsymbol{y}_k^{(t)})$          *Ensures $\boldsymbol{x}_{k+1} \in \mathcal{X}_\leq$*

10:          $\boldsymbol{y}_{k+1}^{(t)} = \boldsymbol{P}_c(\boldsymbol{x}_{k+1}^{(t)} + \frac{\boldsymbol{\lambda}_k^{(t)}}{\beta}) + \boldsymbol{d}_c$

11:          $\boldsymbol{\lambda}_{k+1}^{(t)} = \boldsymbol{\lambda}_k^{(t)} + \beta(\boldsymbol{x}_{k+1}^{(t)} - \boldsymbol{y}_{k+1}^{(t)})$

12:      **end for**

13:      $(\boldsymbol{y}_0^{(t+1)}, \boldsymbol{\lambda}_0^{(t+1)}) \triangleq (\boldsymbol{y}_K^{(t)}, \boldsymbol{\lambda}_K^{(t)})$

14: **end for**

---

**Discussion: ACVI Vs. v-ACVI.** Relative to Algorithm 1, the subproblem for solving $\boldsymbol{x}$ in line 7 in Algorithm 3 becomes more complex, whereas the subproblem for $\boldsymbol{y}$ becomes simpler. Hence, in cases when the inequality constraints are simpler, or there are no inequality constraints Alg. 3 may be more suitable, as that simplifies the $\boldsymbol{x}$ subproblem. However, Algorithm 1 balances better the complexities of the subproblems, hence it may be simpler to use for general problems.

**Convergence of v-ACVI.** By analogous proofs to those in App. B, we can get similar convergence results as for Algorithm 1, that is Theorems 2 and 3. Specifically, Theorem 2 and 3 hold for Algorithm 3, provided that we replace the assumption "$F$ is monotone on $\mathcal{C}_=$" with "$F$ is monotone on $\mathcal{C}_\leq$" in Theorems 2 and 3.

### C.2 Proof of Proposition 3

To prove proposition 3, we will use the following lemma.

**Lemma 11.** $\forall \boldsymbol{b} \in \mathbb{R}^n, \forall \boldsymbol{x} \in \mathcal{C}_<, \frac{1}{2}\|\boldsymbol{x} - \boldsymbol{b}\|_2^2 - \frac{\mu}{\beta}\sum_{i=1}^m \log(-\varphi_i(\boldsymbol{x})) \to +\infty, \|\boldsymbol{x}\|_2 \to +\infty$

*Proof of Lemma 11.* We denote $\phi : \mathcal{C}_< \to \mathbb{R}$ by

$$\phi(\boldsymbol{x}) = \frac{1}{2}\|\boldsymbol{x} - \boldsymbol{b}\|_2^2 - \frac{\mu}{\beta}\sum_{i=1}^m \log(-\varphi_i(\boldsymbol{x}))$$

let $B(\boldsymbol{x}) = -\frac{\mu}{\beta}\sum_{i=1}^{m}\log(-\varphi_i(\boldsymbol{x}))$. We choose an arbitrary $\boldsymbol{x}_0 \in \mathcal{C}_<$. Then by the convexity of $B(\boldsymbol{x})$ we deduce that

$$\forall \boldsymbol{x} \in \mathcal{C}_<, \phi(\boldsymbol{x}) \geqslant \frac{1}{2}\|\boldsymbol{x} - \boldsymbol{b}\|_2^2 + B(\boldsymbol{x}_0) + \nabla B(\boldsymbol{x}_0)^\mathsf{T}(\boldsymbol{x} - \boldsymbol{x}_0) \to +\infty, \|\boldsymbol{x}\|_2 \to +\infty$$

$\square$

In the remaining, we prove Proposition 3 which guarantees that (X-CF) has a unique solution.

*Proof of Proposition 3: uniqueness of the solution of* (X-CF). Let $\phi : \mathcal{C}_< \to \mathbb{R}$ denote:

$$\phi(\boldsymbol{x}) = \frac{1}{2}\|\boldsymbol{x} - \boldsymbol{y}_k + \frac{1}{\beta}(F(\boldsymbol{w}) + \lambda_k)\|_2^2 - \frac{\mu}{\beta}\sum_{i=1}^{m}\log(-\varphi_i(\boldsymbol{x}))$$

We choose $\boldsymbol{x}_0 \in \mathcal{C}_<$. By Lemma 11, $\forall \boldsymbol{x} \in \mathcal{C}_<, \phi(\boldsymbol{x}) \to +\infty, \|\boldsymbol{x}\|_2 \to +\infty$.
So there exists $M > 0$ such that $\boldsymbol{x}_0 \in B(0, M)$ and $\forall \boldsymbol{x} \in S, \phi(\boldsymbol{x}) \leq \phi(\boldsymbol{x}_0), \boldsymbol{x}$ must belong to $B(0, M)$,where

$$B(0, M) = \{\boldsymbol{x} \in \mathbb{R}^n \,|\|\boldsymbol{x}\| \leq M\}$$

It's clear that there exists $t > 0$ such that for every $\boldsymbol{x} \in \mathcal{C}_<$ that satisfies $\phi(\boldsymbol{x}) \leq \phi(\boldsymbol{x}_0)$, $\boldsymbol{x}$ must belong to $\mathcal{C}_t$,where

$$\mathcal{C}_t = \{\boldsymbol{x} \in B(0, M) \,|\varphi_i(\boldsymbol{x}) \leq -t\} \subset \mathcal{C}_< \tag{53}$$

And we can make $t$ small enough so that $\boldsymbol{x}_0 \in \mathcal{C}_t$. $\mathcal{C}_t$ is a nonempty compact set and $\phi$ is continuous, so there exists $\boldsymbol{x}^\star \in \mathcal{C}_t$ such that $\phi(\boldsymbol{x}^\star) \leq \phi(\boldsymbol{x}), \forall \boldsymbol{x} \in \mathcal{C}_<$.

Note that $\forall \boldsymbol{x} \in \mathcal{C}_< \backslash \mathcal{C}_t$, $\phi(\boldsymbol{x}) \geq \phi(\boldsymbol{x}_0) \geq \phi(\boldsymbol{x}^\star)$. Therefore, $\boldsymbol{x}^\star$ is a global minimizer of $\phi$. $\phi$ is strongly-convex thus $\boldsymbol{x}_{k+1} = \boldsymbol{x}^\star$ is its unique minimizer. So $\boldsymbol{x} = \boldsymbol{x}_{k+1}$ if and only if $\nabla\phi(\boldsymbol{x}) = \boldsymbol{0}$.
Therefore, $\boldsymbol{x}_{k+1}$ is the unique solution of $\boldsymbol{x} - \boldsymbol{y}_k + \frac{1}{\beta}(F(\boldsymbol{w}) + \boldsymbol{\lambda}_k) - \frac{\mu}{\beta}\sum_{i=1}^{m}\frac{\nabla\varphi_i(\boldsymbol{x})}{\varphi_i(\boldsymbol{x})} = \boldsymbol{0}$. $\square$

# D    DETAILS ON THE IMPLEMENTATION

In this section, we provide the details on the implementation of the experiments shown in the main part in 2D and higher dimension bilinear game, see § D.1 and § D.2, respectively. We also provide here in § D.3 the details of the MNIST experiments presented later in App. E. In addition, we provide the source code through the following link: `https://github.com/Chavdarova/ACVI`.

## D.1    EXPERIMENTS IN 2D

We first state the considered problem fully, then describe the setup what includes the hyperparameters.

**Problems.** We consider the following constrained bilinear game (for the same experiment shown in Fig. 1 and 5):

$$\min_{x_1 \in \mathbb{R}_+} \max_{x_2 \in \mathbb{R}_+} 0.05x_1^2 + x_1 x_2 - 0.05x_2^2 \,. \tag{cBG}$$

The *Von Neumann's ratio game* (Von Neumann, 1971) (results in Fig. (a)) is as follows:

$$\min_{\boldsymbol{x} \in \Delta^2} \max_{\boldsymbol{y} \in \Delta^2} \frac{\langle \boldsymbol{x}, \boldsymbol{R}\boldsymbol{y} \rangle}{\langle \boldsymbol{x}, \boldsymbol{S}\boldsymbol{y} \rangle} \,, \tag{RG}$$

where $\Delta^2 = \left\{ \boldsymbol{z} \in \mathbb{R}^2 | \boldsymbol{z} \geq 0, \boldsymbol{e}^\mathsf{T}\boldsymbol{z} = 1 \right\}, \boldsymbol{R} = \begin{pmatrix} -0.6 & -0.3 \\ 0.6 & -0.3 \end{pmatrix}$ and $\boldsymbol{S} = \begin{pmatrix} 0.9 & 0.5 \\ 0.8 & 0.4 \end{pmatrix}$.

The so called *Forsaken* (Hsieh et al., 2021) game—used in Fig. 2(b) and in App. E—is as follows:

$$\min_{x_1 \in \mathcal{X}_1} \max_{x_2 \in \mathcal{X}_2} x_1(x_2 - 0.45) + h(x_1) - h(x_2) \,, \tag{Forsaken}$$

where $h(z) = \frac{1}{4}z^2 - \frac{1}{2}z^4 + \frac{1}{6}z^6$. The original version is unconstrained $\mathcal{X} \equiv \mathbb{R}^2$. In Fig. 2(b) we use the constraint $x_1^2 + x_2^2 \leq 4$, and in App. E we use two other constraints: $x_1 \geq 0.08$ and $x_2 \geq 0.4$.

For the toy GAN experiments, shown in Fig. 2(c), the problem is as follows:

$$\min_{\theta} \max_{\varphi} \mathop{\mathbb{E}}_{x \sim \mathcal{N}(0,2)} (\varphi x^2) - \mathop{\mathbb{E}}_{z \sim \mathcal{N}(0,1)} (\varphi \theta^2 z^2)$$
$$s.t. \quad \varphi^2 + \theta^2 \leq 4 \tag{toy-GAN}$$

In the experiment, we look at a finite sum (sample average) approximation, which we then solve deterministically in a full batch fashion.

**Setup.** For all the 2D problems, we set the step size of GDA, EG and OGDA to $0.1$, we use $k = 5$ and $\alpha = 0.5$ for LA-GDA, we set $\beta = 0.08$, $\mu_{-1} = 10^{-5}$, $\delta = 0.5$ and $\boldsymbol{\lambda}_0 = \boldsymbol{0}$ for ACVI; and run for 50 iterations. For ACVI, we set the number of outer loop iterations to $T = 20$. In the first 19 outer loop iterations, we only run one inner loop iteration, and in the last outer loop iteration, we run 30 inner loop iterations (for a total of 50 updates). In Fig. 1 and Fig. (c), we set the starting point for all algorithms. In Fig. 2(a) and (b), we set the starting point to be $(0.5, 0.5)^\mathsf{T}$ for all algorithms.

## D.2    HIGH-DIMENSION BILINEAR GAME

We set the step size of GDA, EG, and OGDA to $0.1$, using $k = 4$ and $\alpha = 0.5$ for LA-GDA. For ACVI, we set $\beta = 0.5$, $\mu_{-1} = 10^{-6}$, $\delta = 0.5$ and $\boldsymbol{\lambda}_0 = \boldsymbol{0}$ for ACVI.

The solution of (HBG) is $\boldsymbol{x}^\star = \frac{1}{500}\boldsymbol{e}$, where $\boldsymbol{e} \in \mathbb{R}^{1000}$. As a metric of the experiments on this problem, we use the relative error: $\varepsilon_r(\boldsymbol{x}_k) = \frac{\|\boldsymbol{x}_k - \boldsymbol{x}^\star\|}{\|\boldsymbol{x}^\star\|}$. In Fig.3(b), we set $\varepsilon = 0.02$ and compute the number of iterations of ACVI needed to reach the relative error given different rotation "strength" $1 - \eta$, $\eta \in (0, 1)$. Here we set the maximum number of iterations to be 50 for all algorithms. In Fig. 3(a), we set $\eta = 0.05$ and compute CPU times needed for ACVI, EG, OGDA, and LA4-GDA to reach different relative errors. Here we set the maximum run time to 1500 seconds for all algorithms. In Fig. 7 in App. E on the other hand, we fix $\eta = 0.05$, and for varying *CPU time* limits, we compute the relative error of the last iterates of ACVI, GDA, EG, OGDA, and LA4-GDA.

**More general HD-BG game (g-HBG).** Since (HBG) has perfect conditioning (that is, the interactive term $\boldsymbol{x}_1^\mathsf{T}\boldsymbol{B}\boldsymbol{x}_2$, is with $\boldsymbol{B} \equiv \boldsymbol{I}$), in App. E.2 we present results on the following more general high

dimensional bilinear game:

$$\min_{\boldsymbol{x}_1 \in \triangle} \max_{\boldsymbol{x}_2 \in \triangle} \quad \frac{\eta}{2} \cdot \mathbf{x}_1^\intercal \mathbf{A} \boldsymbol{x}_1 + (1 - \eta)\, \boldsymbol{x}_1^\intercal \mathbf{B} \boldsymbol{x}_2 - \frac{\eta}{2} \boldsymbol{x}_2^\intercal \mathbf{C} \boldsymbol{x}_2,$$
$$\triangle = \{\boldsymbol{x}_i \in \mathbb{R}^{500} | \boldsymbol{x}_i \geq -\boldsymbol{e}, \text{ and }, \boldsymbol{e}^\intercal \boldsymbol{x}_i = 0\}. \tag{g-HBG}$$

Where $\mathbf{A}$, $\mathbf{B}$ and $\mathbf{C}$ are randomly generated $500 \times 500$ matrices, and $\mathbf{A}$, $\mathbf{C}$ are positive semi-definite.

The solution of (g-HBG) is $\boldsymbol{x}^\star = \mathbf{0}$, where $\mathbf{0} \in \mathbb{R}^{1000}$. As a metric of the experiments on this problem, we use the error $\varepsilon(\boldsymbol{x}_k) = \|\boldsymbol{x}_k\|$. The remaining settings are identical to those of (HBG), explained above.

**Comparison with the Frank-Wolfe algorithm on general HD-BG** (g-HBG) **and** (gg-HBG). In App. E, we compare with the FW method (see A.4) on two problems: (i) (g-HBG), and (ii) (gg-HBG), where the objective is the same as (g-HBG) but the constraint set becomes more general, in which $\boldsymbol{C}_i$ is a randomly generated $10 \times 500$ matrix, $i = \{1, 2\}$. In both experiments, we implement FW as in Algorithm 2, where we choose $\gamma$ to be $2/(2+t)$ at the $t$-th iterate, $t = 0, \cdots, T$. For (i), the constraint set of (g-HBG) is a "shifted simplex", hence its vertices are easy to compute. This allows us to solve the linear minimization problem in Algorithm 2 of (Gidel et al., 2017a) much faster, and we refer to this implementation as *fast-FW*. In contrast, for the (gg-HBG) problem, we cannot apply this, and in that case, we use the standard linear programming routine *covopt.solvers.lp* in Python— referred herein as *FW*.

$$\min_{\boldsymbol{x}_1 \in \mathcal{U}_1} \max_{\boldsymbol{x}_2 \in \mathcal{U}_2} \quad \frac{\eta}{2} \cdot \mathbf{x}_1^\intercal \mathbf{A} \boldsymbol{x}_1 + (1 - \eta)\, \boldsymbol{x}_1^\intercal \mathbf{B} \boldsymbol{x}_2 - \frac{\eta}{2} \boldsymbol{x}_2^\intercal \mathbf{C} \boldsymbol{x}_2,$$
$$\mathcal{U}_i = \{\boldsymbol{x}_i \in \mathbb{R}^{500} | -100\boldsymbol{e} \leq \boldsymbol{x}_i \leq 100\boldsymbol{e}, \text{ and }, \boldsymbol{C}_i \boldsymbol{x}_i = \mathbf{0}\}, i = 1, 2. \tag{gg-HBG}$$

The solution of both (g-HBG) and (gg-HBG) is $\mathbf{0}$. As a metric of the experiments on this problem, we use the error $\varepsilon(\boldsymbol{x}_k) = \|\boldsymbol{x}_k\|$. The remaining settings are identical to those of (HBG), explained above.

### D.3    MNIST AND FASHION-MNIST EXPERIMENTS

For the experiments on the MNIST dataset [1], we use the source code of Chavdarova et al. (2021b) for the baselines and we build on it to implement ACVI. For completeness, we provide an overview of the implementation.

**Models.** We used the DCGAN architectures (Radford et al., 2016), listed in Table 1, and the parameters of the models are initialized using PyTorch default initialization. For experiments on this dataset, we used the *non-saturating* GAN loss as proposed in (Goodfellow et al., 2014):

$$\mathcal{L}_D = \mathbb{E}_{\tilde{\boldsymbol{x}}_d \sim p_d} \log \big( D(\tilde{\boldsymbol{x}}_d) \big) + \mathbb{E}_{\tilde{\boldsymbol{z}} \sim p_z} \log \Big( 1 - D\big(G(\tilde{\boldsymbol{z}})\big) \Big) \tag{L-D}$$

$$\mathcal{L}_G = \mathbb{E}_{\tilde{\boldsymbol{z}} \sim p_z} \log \Big( D\big(G(\tilde{\boldsymbol{z}})\big) \Big), \tag{L-G}$$

where $G(\cdot), D(\cdot)$ denote the generator and discriminator, resp., and $p_d$ and $p_z$ denote the data and the latent distributions (the latter predefined as normal distribution).

**Details on the ACVI implementation.** When implementing ACVI on MNIST, we "remove" the outer loop of Algorithm 1 (that is we set $T = 1$), and fix $\mu$ to be a small number, in particular, we select $\mu = 10^{-9}$. We randomly initialize $\boldsymbol{x}$ and $\boldsymbol{y}$ and initialize $\boldsymbol{\lambda}$ to zero. For lines 8 and 9 of Algorithm 1, we run $l$ steps of (unconstrained) EG and GD, respectively. For the update of $\boldsymbol{x}$ (using EG), we use step-size $\eta_x = 0.001$, whereas for $\boldsymbol{y}$ (using GD), we use step-size $\eta_y = 0.2$. We present results when $l \in \{1, 10\}$. At every iteration, we update $\boldsymbol{\lambda}$ using the expression in line 11 of Algorithm 1, with $\beta = 0.5$.

Because the problem in step 8 of Algorithm 1 does not change a lot over the iterations (as well as when computing $\boldsymbol{y}$), when we implement Algorithm 1 we do *not* reinitialize the variable $\boldsymbol{x}$. We

---

[1]Provided under *Creative Commons Attribution-Share Alike 3.0.*

| Generator | Discriminator |
|---|---|
| *Input:* $\boldsymbol{z} \in \mathbb{R}^{128} \sim \mathcal{N}(0, I)$ | *Input:* $\boldsymbol{x} \in \mathbb{R}^{1 \times 28 \times 28}$ |
| transposed conv. (ker: 3×3, 128 → 512; stride: 1) | conv. (ker: 4×4, 1 → 64; stride: 2; pad:1) |
| Batch Normalization | LeakyReLU (negative slope: 0.2) |
| ReLU | conv. (ker: 4×4, 64 → 128; stride: 2; pad:1) |
| transposed conv. (ker: 4×4, 512 → 256, stride: 2) | Batch Normalization |
| Batch Normalization | LeakyReLU (negative slope: 0.2) |
| ReLU | conv. (ker: 4×4, 128 → 256; stride: 2; pad:1) |
| transposed conv. (ker: 4×4, 256 → 128, stride: 2) | Batch Normalization |
| Batch Normalization | LeakyReLU (negative slope: 0.2) |
| ReLU | conv. (ker: 3×3, 256 → 1; stride: 1) |
| transposed conv. (ker: 4×4, 128 → 1, stride: 2, pad: 1) | $Sigmoid(\cdot)$ |
| $Tanh(\cdot)$ | |

Table 1: DCGAN architectures (Radford et al., 2016) used for experiments on **MNIST**. With "conv." we denote a convolutional layer and "transposed conv" a transposed convolution layer (Radford et al., 2016). We use *ker* and *pad* to denote *kernel* and *padding* for the (transposed) convolution layers, respectively. With $h \times w$ we denote the kernel size. With $c_{in} \to y_{out}$ we denote the number of channels of the input and output, for (transposed) convolution layers. The models use Batch Normalization (Ioffe & Szegedy, 2015) layers.

instead use the one from the previous iteration as initialization and update it $l$ times. The full details of the training when using an inner optimizer for step 8 of Algorithm 1 are provided in Algorithm 4, where we recall that $G(\boldsymbol{x})$ is defined as:

$$G(\boldsymbol{x}) \triangleq \boldsymbol{x} + \frac{1}{\beta}\boldsymbol{P}_c F(\boldsymbol{x}) - \boldsymbol{P}_c \boldsymbol{y}_k + \frac{1}{\beta}\boldsymbol{P}_c \boldsymbol{\lambda}_k - \boldsymbol{d}_c \qquad \text{(G-EQ)}$$

Note that in the case of MNIST, we consider only inequality constraints (and there are no equality constraints), therefore, the matrices $\boldsymbol{P}_c$ and $\boldsymbol{d}_c$ are identity and zero, respectively. Thus, there is no need for lines 3 and 4 in Algorithm 4.

---

**Algorithm 4** Pseudocode for ACVI when using an inner optimizer (MNIST experiments).

---

1: **Input:** operator $F : \mathcal{X} \to \mathbb{R}^n$, equality $\boldsymbol{Cx} = \boldsymbol{d}$ and inequality constraints $\varphi_i(\boldsymbol{x}) \leq 0, i = [m]$, hyperparameters $\mu, \beta > 0, \delta \in (0, 1)$, inner optimizer $\mathcal{A}$ (e.g. EG, GDA, OGDA), $l$ number of steps for the inner-optimizer, number of iterations $K$.
2: **Initialize:** $\boldsymbol{x}_0 \in \mathbb{R}^n, \boldsymbol{y}_0 \in \mathbb{R}^n, \boldsymbol{\lambda}_0 \in \mathbb{R}^n$
3: $\boldsymbol{P}_c \triangleq \boldsymbol{I} - \boldsymbol{C}^\mathsf{T}(\boldsymbol{CC}^\mathsf{T})^{-1}\boldsymbol{C}$                    where $\boldsymbol{P}_c \in \mathbb{R}^{n \times n}$
4: $\boldsymbol{d}_c \triangleq \boldsymbol{C}^\mathsf{T}(\boldsymbol{CC}^\mathsf{T})^{-1}\boldsymbol{d}$                    where $\boldsymbol{d}_c \in \mathbb{R}^n$
5: **for** $k = 0, \ldots, K - 1$ **do**
6:     To obtain $\boldsymbol{x}_{k+1}$: run $l$ steps of $\mathcal{A}$ solving VI$(\mathbb{R}^n, G)$, where $G$ is defined in (G-EQ)
7:     To obtain $\boldsymbol{y}_{k+1}$: run $l$ steps of GD to find $\boldsymbol{y}_{k+1}$ that minimizes:
$$-\mu \sum_{i=1}^m \log\left(-\varphi_i(\boldsymbol{y})\right) + \frac{\beta}{2} \left\| \boldsymbol{y} - \boldsymbol{x}_{k+1} - \frac{1}{\beta}\boldsymbol{\lambda}_k \right\|^2$$
8:     $\boldsymbol{\lambda}_{k+1} = \boldsymbol{\lambda}_k + \beta(\boldsymbol{x}_{k+1} - \boldsymbol{y}_{k+1})$
9: **end for**
10: **Return:** $\boldsymbol{x}_K$

---

The implementation details for the Fashion-MNIST experiment are identical to those of the MNIST experiment.

**Setup** 1**: MNIST.** The MNIST experiment in Fig. 4 in the main part (and Fig. 11, 12) has 100 randomly generated linear inequality constraints for the Generator and 100 for the Discriminator.

**Setup** 1**: projection details.** Suppose the linear inequality constraints for the Generator are $\boldsymbol{A\theta} \leq \boldsymbol{b}$, where $\boldsymbol{\theta} \in \mathbb{R}^n$ is the vector of all parameters of the Generator, $\boldsymbol{A} = (\boldsymbol{a}_1^\mathsf{T}, \cdots, \boldsymbol{a}_{100}^\mathsf{T})^\mathsf{T} \in \mathbb{R}^{100 \times n}$, $\boldsymbol{b} = (b_1, \ldots, b_{100}) \in \mathbb{R}^{100}$. We use the *greedy projection algorithm* described in (Beck, 2017). A greedy projection algorithm is essentially a projected gradient method, it is easy to implement in

high-dimension problems, and it has a convergence rate of $O(1/\sqrt{K})$. See Chapter 8.2.3 in (Beck, 2017) for more details. Since the dimension $n$ is very large, at each step of the projection, one could only project $\boldsymbol{\theta}$ to one hyperplane $\boldsymbol{a}_i^\mathsf{T}\boldsymbol{x} = b_i$ for some $i \in \mathcal{I}(\boldsymbol{\theta})$, where

$$\mathcal{I}(\boldsymbol{\theta}) \triangleq \{j | \boldsymbol{a}_j^\mathsf{T}\boldsymbol{\theta} > b_j\}.$$

For every $j \in \{1, 2, \ldots, 100\}$, let

$$\mathcal{S}_j \triangleq \{\boldsymbol{x} | \boldsymbol{a}_j^\mathsf{T}\boldsymbol{x} \leq b_j\}.$$

The greedy projection method chooses $i$ so that $i \in \arg\max\{dist(\boldsymbol{\theta}, \mathcal{S}_i)\}$. Note that as long as $\boldsymbol{\theta}$ is not in the constraint set $C_\leq = \{\boldsymbol{x} | \boldsymbol{A}\boldsymbol{x} \leq \boldsymbol{b}\}$, $i$ would be in $\mathcal{I}(\boldsymbol{\theta})$. Algorithm 5 gives the details of the greedy projection method we use for the baseline, written for the Generator only for simplicity; the same projection method is used for the Discriminator as well.

---

**Algorithm 5** Greedy projection method for the baseline.

---

1: **Input:** $\boldsymbol{\theta} \in \mathbb{R}^n$, $\boldsymbol{A} = (\boldsymbol{a}_1^\mathsf{T}, \ldots, \boldsymbol{a}_{100}^\mathsf{T})^\mathsf{T} \in \mathbb{R}^{100 \times n}$, $\boldsymbol{b} = (b_1, \ldots, b_{100}) \in \mathbb{R}^{100}$, $\varepsilon > 0$
2: **while** True **do**
3:     $\mathcal{I}(\boldsymbol{\theta}) \triangleq \{j | \boldsymbol{a}_j^\mathsf{T}\boldsymbol{\theta} > b_j\}$
4:     **if** $\mathcal{I}(\boldsymbol{\theta}) = \emptyset$ or $\max\limits_{j \in \mathcal{I}(\boldsymbol{\theta})} \frac{|\boldsymbol{a}_j^\mathsf{T}\boldsymbol{\theta} - b_j|}{\|\boldsymbol{a}_j\|} < \varepsilon$ **then**
5:         break
6:     **end if**
7:     choose $i \in \arg\max\limits_{j \in \mathcal{I}(\boldsymbol{\theta})} \frac{|\boldsymbol{a}_j^\mathsf{T}\boldsymbol{\theta} - b_j|}{\|\boldsymbol{a}_j\|}$
8:     $\boldsymbol{\theta} \leftarrow \boldsymbol{\theta} - \frac{|\boldsymbol{a}_i^\mathsf{T}\boldsymbol{\theta} - b_i|}{\|\boldsymbol{a}_i\|^2}\boldsymbol{a}_i$
9: **end while**
10: **Return:** $\boldsymbol{\theta}$

---

**Setup** 2**: MNIST & Fashion-MNIST.** We add two constraints for the MNIST experiment: we set the squared sum of all parameters of the Generator and that of the Discriminator (separately) to be less than or equal to a hyperparameter $M$. We select a large number for $M$; in particular, we set $M = 50$.

**Metrics.** We describe the metrics for the MNIST experiments shown later in App. E. We use the two standard GAN metrics, Inception Score (IS, Salimans et al., 2016) and Fréchet Inception Distance (FID, Heusel et al., 2017). Both FID and IS rely on a pre-trained classifier and take a finite set of $\tilde{m}$ samples from the generator to compute these. Since **MNIST** has greyscale images, we used a classifier trained on this dataset and used $\tilde{m} = 5000$.

**Metrics: IS.** Given a sample from the generator $\tilde{\boldsymbol{x}}_g \sim p_g$—where $p_g$ denotes the data distribution of the generator—IS uses the softmax output of the pre-trained network $p(\tilde{\boldsymbol{y}}|\tilde{\boldsymbol{x}}_g)$ which represents the probability that $\tilde{\boldsymbol{x}}_g$ is of class $c_i, i \in 1\ldots C$, i.e., $p(\tilde{\boldsymbol{y}}|\tilde{\boldsymbol{x}}_g) \in [0, 1]^C$. It then computes the marginal class distribution $p(\tilde{\boldsymbol{y}}) = \int_{\tilde{\boldsymbol{x}}} p(\tilde{\boldsymbol{y}}|\tilde{\boldsymbol{x}}_g)p_g(\tilde{\boldsymbol{x}}_g)$. IS measures the Kullback–Leibler divergence $\mathbb{D}_{KL}$ between the predicted conditional label distribution $p(\tilde{\boldsymbol{y}}|\tilde{\boldsymbol{x}}_g)$ and the marginal class distribution $p(\tilde{\boldsymbol{y}})$. More precisely, it is computed as follows:

$$IS(G) = \exp\left(\mathbb{E}_{\tilde{\boldsymbol{x}}_g \sim p_g}\left[\mathbb{D}_{KL}\big(p(\tilde{\boldsymbol{y}}|\tilde{\boldsymbol{x}}_g)||p(\tilde{\boldsymbol{y}})\big)\right]\right) = \exp\left(\frac{1}{\tilde{m}}\sum_{i=1}^{\tilde{m}}\sum_{c=1}^{C} p(y_c|\tilde{\boldsymbol{x}}_i)\log\frac{p(y_c|\tilde{\boldsymbol{x}}_i)}{p(y_c)}\right). \quad \text{(IS)}$$

It aims at estimating (i) if the samples look realistic i.e., $p(\tilde{\boldsymbol{y}}|\tilde{\boldsymbol{x}}_g)$ should have low entropy, and (ii) if the samples are diverse (from different ImageNet classes), i.e., $p(\tilde{\boldsymbol{y}})$ should have high entropy. As these are combined using the Kullback–Leibler divergence, the higher the score is, the better the performance.

**Metrics: FID.** Contrary to IS, FID compares the synthetic samples $\tilde{\boldsymbol{x}}_g \sim p_g$ with those of the training dataset $\tilde{\boldsymbol{x}}_d \sim p_d$ in a feature space. The samples are embedded using the first several layers of a pre-trained classifier. It assumes $p_g$ and $p_d$ are multivariate normal distributions and estimates the means $\boldsymbol{m}_g$ and $\boldsymbol{m}_d$ and covariances $\boldsymbol{C}_g$ and $\boldsymbol{C}_d$, respectively, for $p_g$ and $p_d$ in that feature space.

Finally, FID is computed as:

$$\mathbb{D}_{\text{FID}}(p_d, p_g) \approx \mathscr{D}_2\big((\boldsymbol{m}_d, \boldsymbol{C}_d), (\boldsymbol{m}_g, \boldsymbol{C}_g)\big) = \|\boldsymbol{m}_d - \boldsymbol{m}_g\|_2^2 + Tr\big(\boldsymbol{C}_d + \boldsymbol{C}_g - 2(\boldsymbol{C}_d\boldsymbol{C}_g)^{\frac{1}{2}}\big), \tag{FID}$$

where $\mathscr{D}_2$ denotes the Fréchet Distance. Note that as this metric is a distance, the lower it is, the better the performance.

**Hardware.** We used the Colab platform (`https://colab.research.google.com/`) and *Tesla P100* GPUs. The running times are reported in App. E.

# E  ADDITIONAL EMPIRICAL ANALYSIS

In this section, we provide some omitted plots/analyses of the results in the main paper as well as additional experiments. In particular, (i) App. E.1 lists results in 2D, (ii) App. E.2 on (HBG) and (g-HBG), whereas (iii) App. E.3 provides more detailed plots of those experiments summarized in Fig. 4 and presents additional results on other constraints on MNIST where we compare computationally-wise with *unconstained* baselines.

## E.1  ADDITIONAL EXPERIMENTS IN 2D, ON HBG AND ON G-HBG

**Depicting the omitted baselines of Fig. 1.** While Fig. 1 lists solely EG and ACVI for clarity, Fig. 5 depicts all the considered baselines in this paper on the cBG problem for completeness.

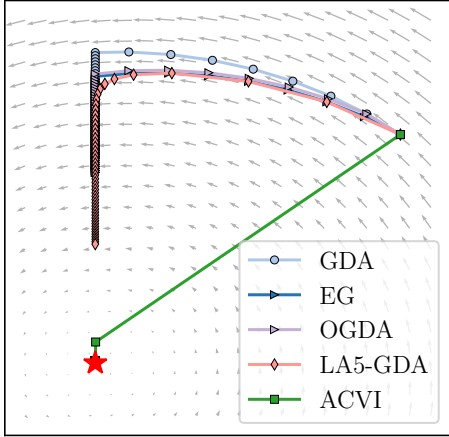

Figure 5: In addition to Fig. 1, here we depict all the considered baselines on the cBG problem. The solution at $(0,0)$ is denoted with a red star ($\star$), and the vector field of this problem with gray arrows. See App. D.1 for details on the implementation.

**Additional experiments: varying constraints on the *Forsaken* problem.** The *Forsaken* game was first pointed out in (Hsieh et al., 2021) and is particularly relevant because it has *limit cycles*, despite that it is in 2D. Since we are missing a tool to detect if we are in a limit cycle when in higher dimensions, this example is a popular benchmark in many recent works. Interestingly, in Fig. 2(b) we observe that ACVI is the only method that escapes the limit cycle. However, since in those simulations, given the initial point the constraints are not active throughout the training, in this section, we run experiments with additional constraints. Fig. 6 depicts the baseline methods and ACVI on the Forsaken problem with two different constraints than that considered in Fig. 2 (that $x_1^2 + x_2^2 \leq 4$). Since this game is non-monotone, we observe that for some constraints the baseline methods—GDA, EG, OGDA, LA4-GDA—stay near the constraint (and do not converge). This may indicate that ACVI may have better chances of converging for *broader* problem classes than monotone VIs, relative to baseline methods whose convergence may depend on the constraints, and when hitting a constraint may be significantly slower (as Fig. 1 illustrates).

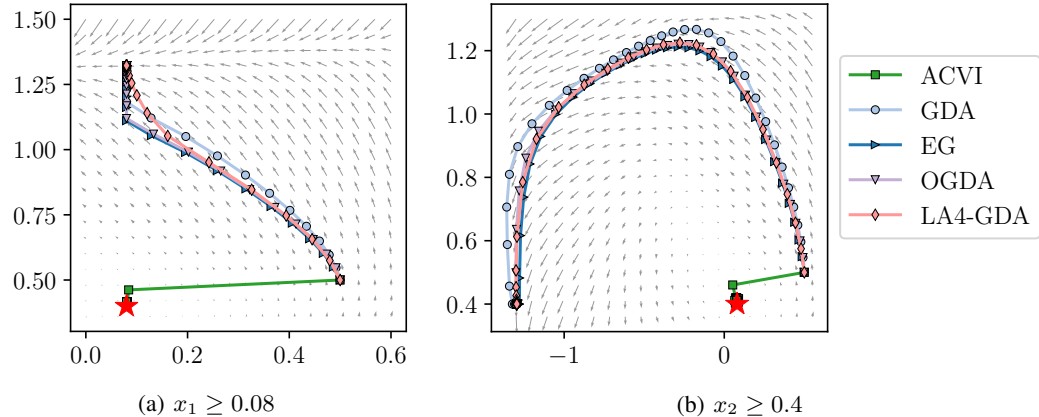

(a) $x_1 \geq 0.08$    (b) $x_2 \geq 0.4$

Figure 6: *Forsaken game with different constraints*: we consider two additional (to that in Fig. 2) constraints: (a) that $x_1 \geq 0.08$, and (b) that $x_2 \geq 0.4$. See App. D.1 for details on the implementation, and App. E.1 for a discussion.

## E.2   ADDITIONAL EXPERIMENTS HBG AND ON G-HBG

**Complementary analysis to those in Fig. 3.** Similar to Fig. 3, in Fig. 7 we run experiments on the HBG problem. However, here for a given fixed CPU time, we depict the relative error of the considered baselines and ACVI.

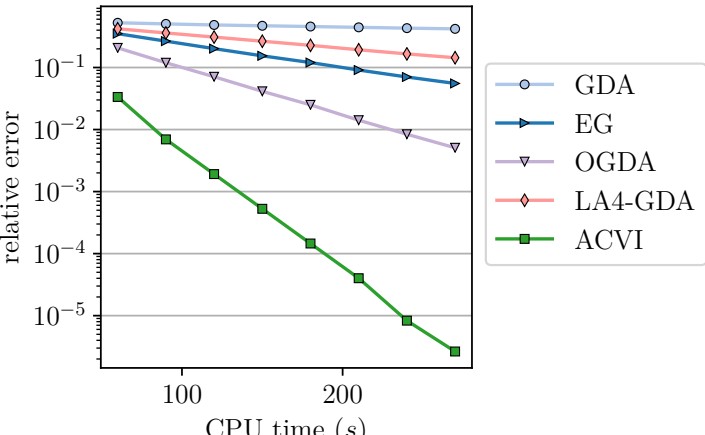

Figure 7: Given varying CPU time (in seconds), depicting the relative error (see App. D.2) of GDA, EG, OGDA, LA4-GDA, and ACVI (Algorithm 1) on the HBG problem where $\eta$ is fixed to $\eta = .05$ (hence the vector field is highly rotational). This experiment complements those in Fig. 3 in the main paper. For the details on the implementation, see App. D.2.

**Additional experiments on** (g-HBG). In Fig. 8 we run experiments on the generalized HBG problem (g-HBG). In figure 8(a), we compute the number of iterations needed to reach $\varepsilon$-distance to solution for varying intensity of the rotational component $(1 - \eta)$; in figure 8(b), we compute the error of the last iterate given fixed CPU time. We observe that despite the highly rotational monotone vector field, ACVI converges significantly faster in terms of wall clock time in higher dimensions as well.

**Comparison with Frank-Wolf algorithm on** (g-HBG) **and** (gg-HBG). Similar to Fig. 8, in Fig. 9 we also run experiments on (g-HBG), but here we compare ACVI with FW. We observe that ACVI outperforms FW even when we make use of the special structure of the simple constraint set when solving the linear minimization problem in FW (the *fast FW* method). Similar to Fig. 9(b), in Fig. 10

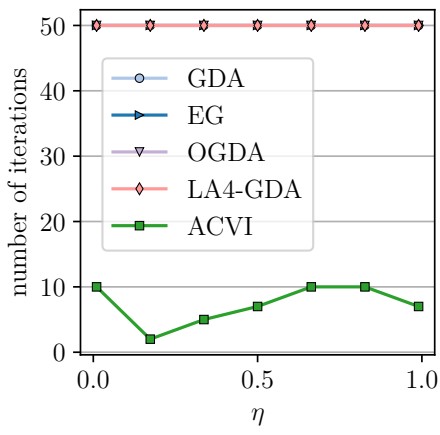
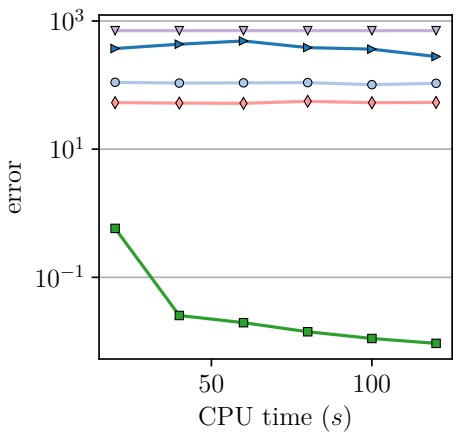

(a) Varying rotational intensity $(1 - \eta)$   (b) Achieved error given varying fixed CPU time

Figure 8: *General high-dimensional bilinear game* (g-HBG): comparison of ACVI with the GDA, EG, OGDA, and LA4-GDA baselines (described in App. A.4). **Left:** number of iterations ($y$-axis) needed to reach an $\epsilon$-distance to the solution, for varying intensity of the rotational component $1 - \eta$ ($\eta$ is the $x$-axis) of the vector field (the smaller the $\eta$ the higher the rotational component). We fix a threshold of the maximum number of iterations, and we stop the experiment. **Right:** distance to the solution (see App. D.2) of the last iterate ($y$-axis) for a varying wall-clock CPU time allowed to run each experiment ($x$-axis); in this experiment $\eta$ is fixed to $\eta = 0.05$. See App. E.2 and D.2 for discussion and details on the implementation, respectively.

we run experiments on the (gg-HBG) problem, where we fix CPU time and depict the relative error of ACVI and FW.

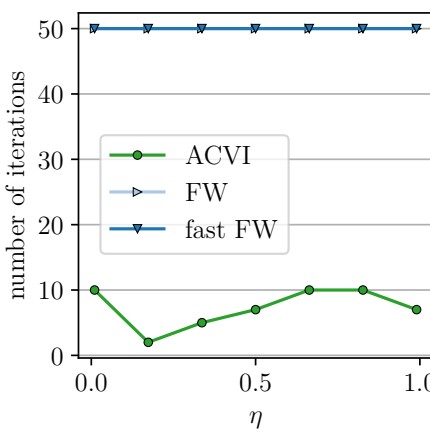
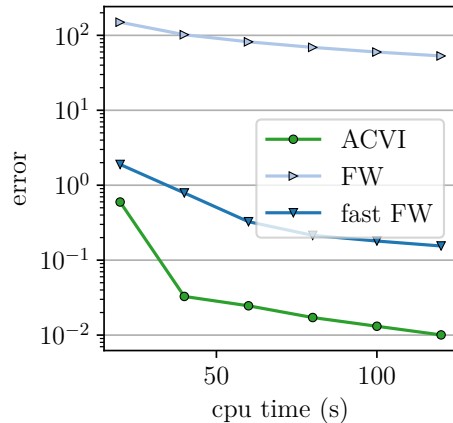

(a) Varying rotational intensity $(1 - \eta)$   (b) Achieved error given varying fixed CPU time

Figure 9: *General high-dimensional bilinear game* (g-HBG): comparison of ACVI with FW baseline (Algorithm 2). **Left:** number of iterations ($y$-axis) needed to reach an $\epsilon$-distance to the solution, for varying intensity of the rotational component $1 - \eta$ ($\eta$ is the $x$-axis) of the vector field (the smaller the $\eta$ the higher the rotational component). We fix a threshold of the maximum number of iterations, and we stop the experiment. **Right:** distance to the solution (see App. D.2) of the last iterate ($y$-axis) for a varying wall-clock CPU time allowed to run each experiment ($x$-axis); in this experiment $\eta$ is fixed to $\eta = 0.05$. See App. E.2 and D.2 for discussion and details on the implementation, respectively.

Since FW (and variants, such as approximate and accelerated) rely on a specific structure of the constraints, FW can be extremely slow when those assumptions are not met–see discussion by Jaggi

(2013) in §3 as well as examples in §4 therein. In contrast, the herein-presented ACVI Algorithm focuses on constraints of a general form, and further variants can be derived out of it to also exploit the structure of the constraints. We leave exploiting such constraint structure—including extending FW to VIs and deriving variants of ACVI—for future work.

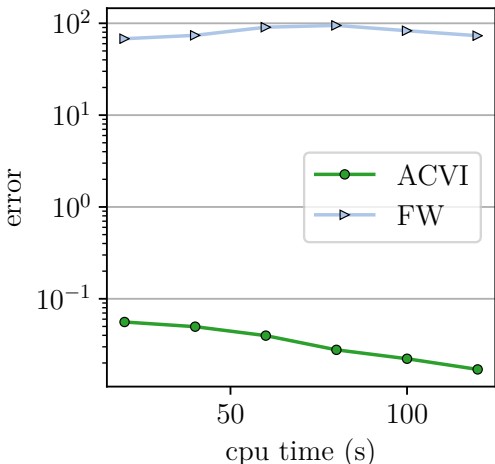

Figure 10: Given varying CPU time (in seconds), depicting the relative error (see App. D.2) of FW and ACVI (Algorithm 1) on the HBG problem where $\eta$ is fixed to $\eta = .05$ (hence the vector field is highly rotational). For the details on the implementation, see App. D.2.

### E.3 EXPERIMENTS ON MNIST AND FASHION-MNIST

#### E.3.1 SETUP 1: EXPERIMENTS ON MNIST WITH LINEAR INEQUALITIES

In this section, we present more detailed results of the summarizing plot in Fig. 4 of the main paper. For this experiment, we used linear inequalities as described in § D.3. Unlike in subsection E.3.2, here all the baselines are projected methods (that is, the same problem setting applies to ACVI and the baselines).

Fig. 11 and 12 depict the comparisons with projected GDA and projected EG, respectively. We observe that ACVI converges fast relative to the corresponding baseline. When choosing a larger number of steps for the inner problem $l = 10$ (see Algorithm 4) the wall-clock time per iteration increases, and interestingly the ACVI steps compensate for that and overall converge as fast as when $l = 1$.

#### E.3.2 SETUP 2: EXPERIMENTS ON (FASHION-)MNIST WITH QUADRATIC INEQUALITIES

In this section, we consider the MNIST and Fashion-MNIST datasets, which are unconstrained problems so as to make use of the well-established performance metrics (which are otherwise unclear in the non-monotone settings, where we do not know the optimal solution apriori). We augment the problem with a mild constraint which requires that the norm of the per-player parameters does not exceed a certain value (see App. D.3). We compare ACVI with *unconstrained* baselines, which sets ACVI at a disadvantage as the projection requires additional computation. However, the primary purpose of these experiments is to observe if Algorithm 1 is competitive computationally-wise when lines 8 and 9 are non-trivial and require an (unconstrained) solver. However note that since MNIST is a relatively easy problem, it may not answer the natural question if ACVI has advantages on problems augmented with constraints over standard unconstrained methods. We leave such analyses for future work. The implementation and the used metrics are described in App. D.3.

Fig. 13 summarizes the experiments in terms of the obtained FID score over time. We observe that ACVI (although it uses two solvers at each iteration) is yet performing *competitively* to *unconstrained* GDA and EG. Figures 14–19 provide in addition samples of the Generator and IS scores, separately for each method. Figures 20 and 21 depict samples generated by the different methods,

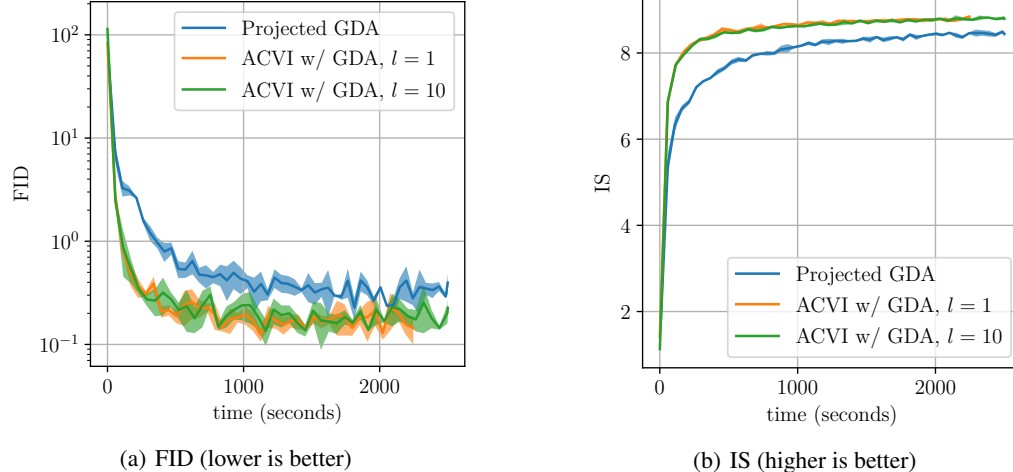

(a) FID (lower is better)   (b) IS (higher is better)

Figure 11: *Setup* 1*: Comparison between ACVI and GDA, and the **projected GDA** on MNIST with linear inequalities (described in § D.3). $l$ denotes the number of steps for the inner problems, see Algorithm 4. The depicted results are over multiple seeds. The FID and IS metrics as well as the implementation details are described in App. D.3. See App. E.3.1 for discussion.

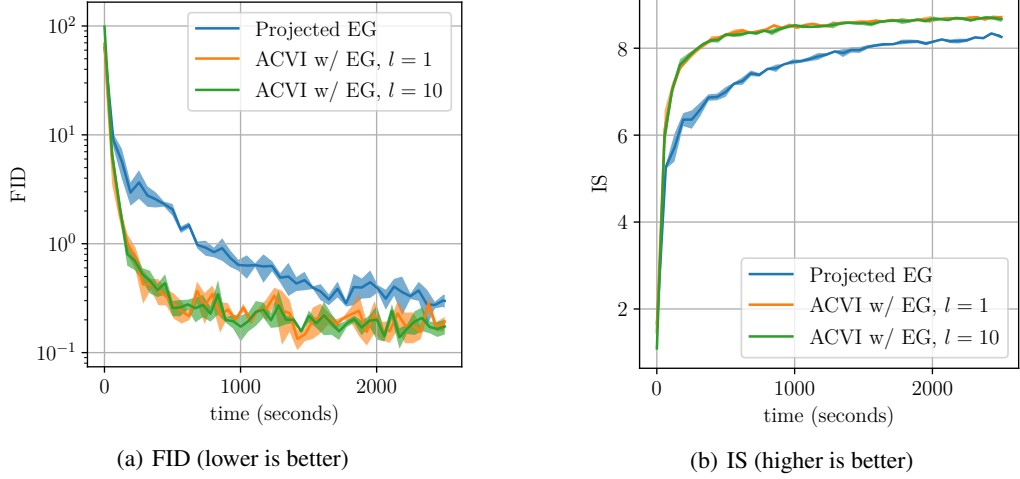

(a) FID (lower is better)   (b) IS (higher is better)

Figure 12: *Setup* 1*: Comparison between ACVI and EG, and the **projected EG** on MNIST with linear inequalities (described in § D.3). $l$ denotes the number of steps for the inner problems, see Algorithm 4. The depicted results are over multiple seeds. The FID and IS metrics as well as the implementation details are described in App. D.3. See App. E.3.1 for discussion.

when trained on the Fashion-MNIST dataset. We believe that further exploring the type of constraints to be added, or the implementation options (e.g., $l$, step-size) may be proven fruitful even for problems that are originally unconstrained–as such an approach may reduce the rotational component of the original vector field, what in turn causes faster convergence or may help in escaping limit cycles for problems beyond monotone ones.

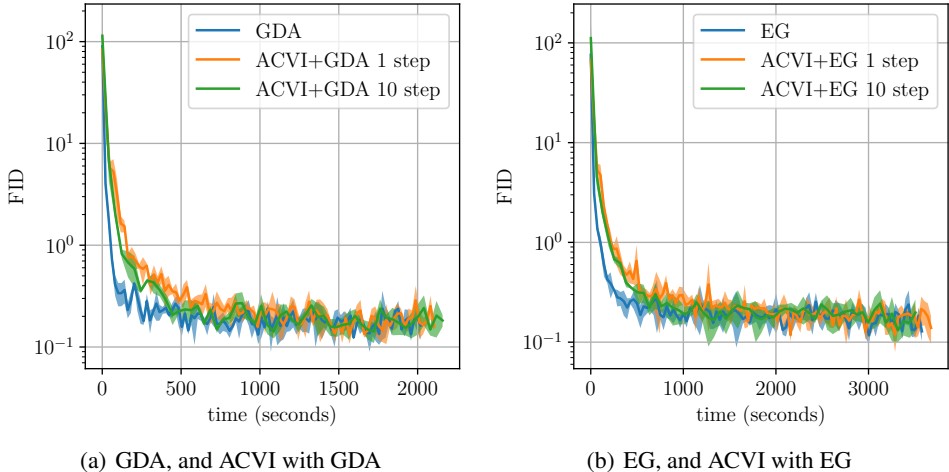

(a) GDA, and ACVI with GDA

(b) EG, and ACVI with EG

Figure 13: Summary of the experiments on MNIST, using FID (lower is better). 13(a): GDA and ACVI with GDA, and 13(b): EG and ACVI with EG, using $l = \{1, 10\}$ for ACVI. Using step size of 0.001. The depicted results are over multiple seeds. See App. D.3 and E.3 for details on the implementation and discussion, resp.

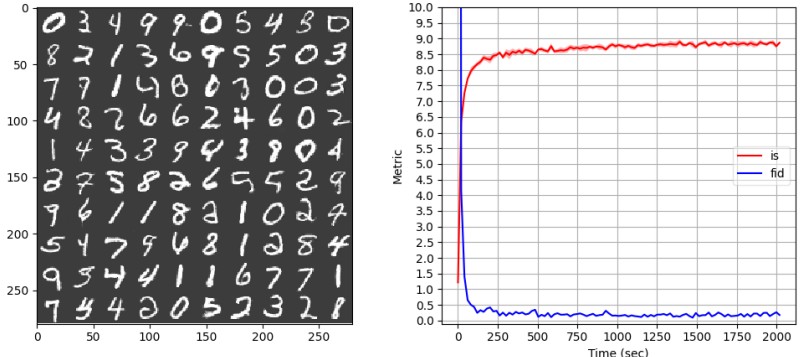

Figure 14: GDA on MNIST. Left: samples $\tilde{x}_g \sim p_g$ of the last iterate of the Generator. Right: FID and IS of GDA, depicted in blue and red, respectively. Using step size of 0.001.

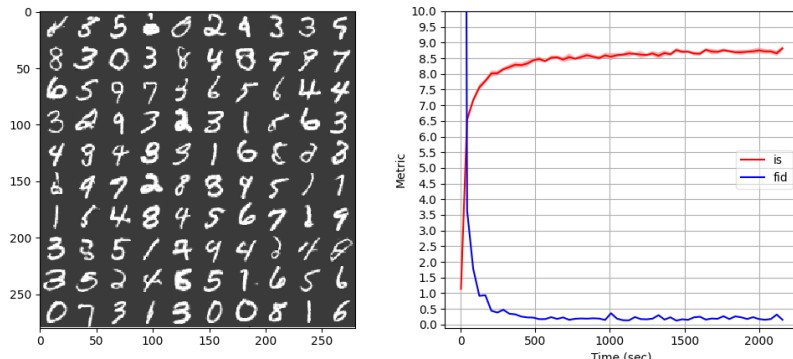

Figure 15: ACVI (Algorithm 1) with 10 GDA steps on MNIST. Left: samples $\tilde{x}_g \sim p_g$ of the last iterate of the Generator. Right: FID and IS of GDA, depicted in blue and red, respectively. Using step size of $0.001$ for $x$ and $0.2$ for $y$, and $l = 10$ both for $x$ and $y$.

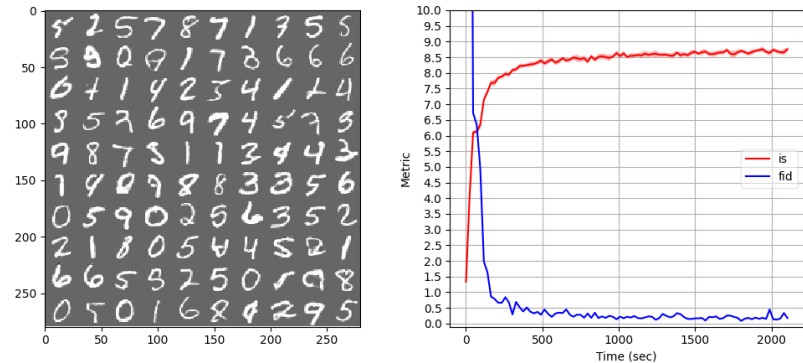

Figure 16: ACVI (Algorithm 1) with 1 GDA step on MNIST. Left: samples $\tilde{x}_g \sim p_g$ of the last iterate of the Generator. Right: FID and IS of GDA, depicted in blue and red, respectively. Using step size of $0.001$ for $x$ and $0.2$ for $y$, and $l = 1$ both for $x$ and $y$.

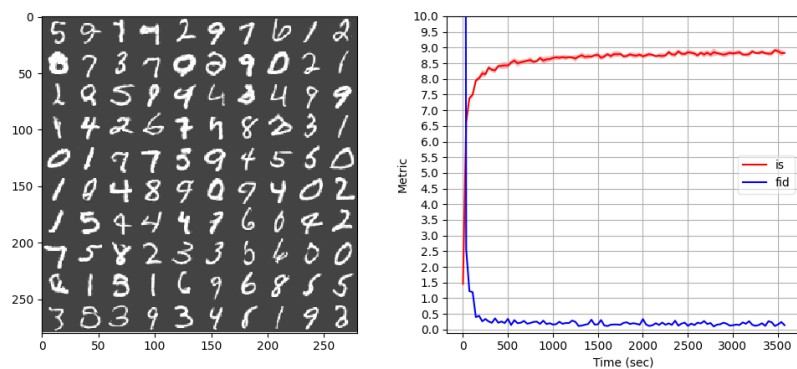

Figure 17: EG on MNIST. Left: samples $\tilde{x}_g \sim p_g$ of the last iterate of the Generator. Right: FID and IS of EG, depicted in blue and red, respectively. Using step size of $0.001$.

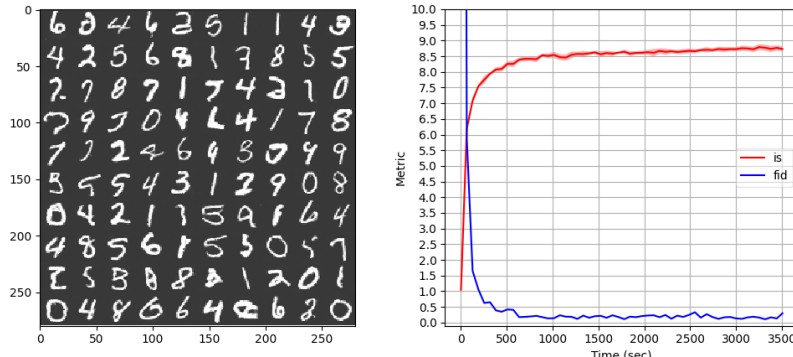

Figure 18: ACVI (Algorithm 1) with 10 EG steps on MNIST. Left: samples $\tilde{x}_g \sim p_g$ of the last iterate of the Generator. Right: FID and IS of GDA, depicted in blue and red, respectively. Using step size of $0.001$ for $x$ and $0.2$ for $y$, and $l = 10$ both for $x$ and $y$.

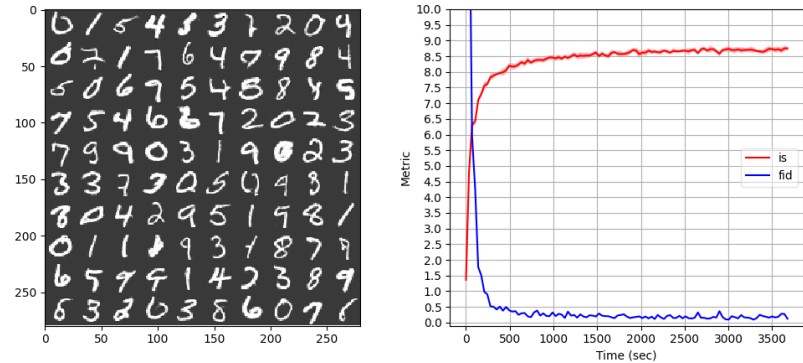

Figure 19: ACVI (Algorithm 1) with 1 EG step on MNIST. Left: samples $\tilde{x}_g \sim p_g$ of the last iterate of the Generator. Right: FID and IS of GDA, depicted in blue and red, respectively. Using step size of $0.001$ for $x$ and $0.2$ for $y$, and $l = 1$ both for $x$ and $y$.

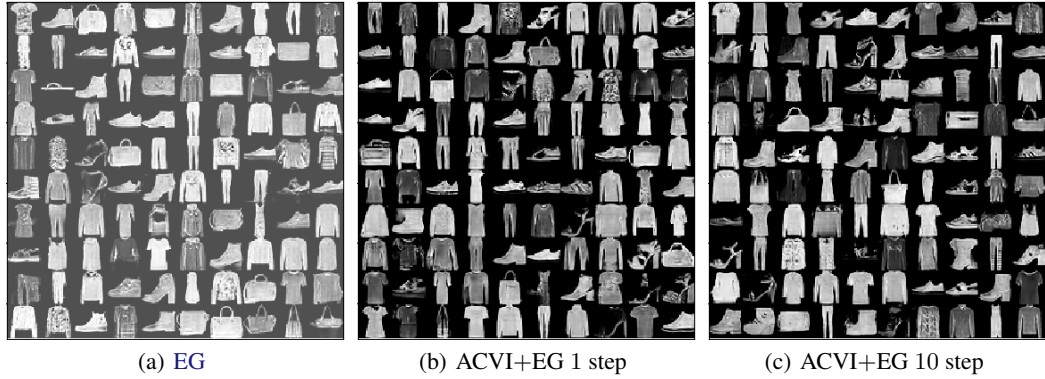

(a) EG        (b) ACVI+EG 1 step        (c) ACVI+EG 10 step

Figure 20: Generated images at fixed wall-clock computation time (3000s) by: the baseline EG, and by ACVI with $l \in \{1, 10\}$ on the Fashion-MNIST (Xiao et al., 2017) dataset. See App. D.3 and E.3 for details on the implementation and discussion, resp.

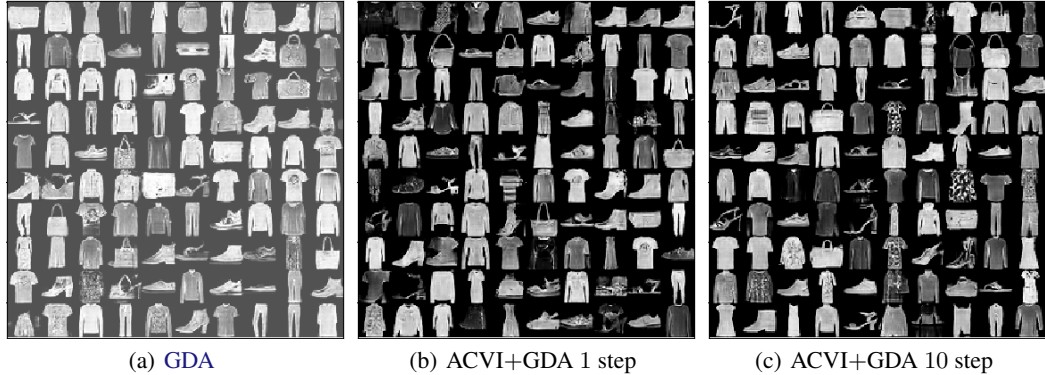

(a) GDA       (b) ACVI+GDA 1 step       (c) ACVI+GDA 10 step

Figure 21: Generated images at fixed wall-clock computation time (3000s) by: the baseline GDA, and by ACVI with $l \in \{1, 10\}$ on the Fashion-MNIST (Xiao et al., 2017) dataset. See App. D.3 and E.3 for details on the implementation and discussion, resp.

