# OpenReview forum: "Solving Constrained Variational Inequalities via a First-order Interior Point-based Method"
_ICLR.cc/2023/Conference — ICLR 2023 notable top 25%_

### Official Review · Reviewer_WuKs · 2022-10-26

**Confidence:** 3
**Correctness:** 3
**Technical Novelty And Significance:** 2
**Empirical Novelty And Significance:** 3
**Recommendation:** 6

**Clarity, Quality, Novelty And Reproducibility:**

## Clarity
I found paper clear in overall. There are a few typos:
1. Typos in equation (AL-CVX)
2. Typos in the last line of (KKT)

**Strength And Weaknesses:**

### Strength
1. The method helps to alleviate the problem in minimax optimization when the projection is not cheap.
2. The idea seems novel to me.
3. In the experiments, in some problems, the method has obvious gains.


### Weaknesses
1. The method brings a couple of subproblems to solve for the primal variables in Lagrangian, e.g., (Y-EQ) and (W-EQ). But I also think it might not be a big problem. (Y-EQ) seems to be strongly convex, so it can be well-solved. (W-EQ) is associated with a strongly-monotone variational inequality, so it can also be solved fast.
2. Even though the paper claims the algorithm is parameter-free, we may need access to problem parameters when solving the subproblems mentioned above.

Questions:
1. In the design of the algorithm, why in the equation (1) we want to introduce a new variable y = x in order to use ADMM? Can this problem be solved directly before we replace x = y?
2. In Section 4.2, it mentioned that in the monotone case, (some) constraints need to be active. Where does this assumption appear in Theorem 3?
3. Why not compare the algorithm with Frank-Wolfe type of algorithm in the experiment?

**Summary Of The Paper:**

The paper aims to solve constrained monotone variational inequality when the projection step on the constraint is not simple. In particular, it uses the interior point method to handle the constraint, and then reformulate it to a form that can be solved by ADMM. For the convergence result, it provides guarantees for two settings: (a) star-$\xi$-monotone operator; (b) monotone and not purely rotational.

**Summary Of The Review:**

Please see above.



--------------------------------------
Thanks for the authors' response! I will keep my score.

---

> ### Author Response · Authors · 2022-11-12
> **Response to Reviewer WuKs**
>
> Thank you very much for your time reading our work and for your feedback.
> We corrected the typos in equation (AL-CVX); see revised pdf. We believe that equation (KKT) is correct, and it is consistent with that in Sec. 1.3.2 in [Facchinei and Pang, 2003]. We may have caused confusion because we forgot to define the notation $\perp$ in the last line of (KKT)--which we fixed now. Nonetheless, please let us know if that is not what you referred to.
>
>
> **1. On the new variable $\mathbf{y}$ in the derivation of ACVI.** The standard approach to  directly solve that sub-problem
> is by Newton's method, which is second order (hence we avoid it).
> Projecting the vectors on the equality constraint set $\{ \mathbf{x}|\mathbf{C}\mathbf{x}=\mathbf{d} \}$ using the projection matrix can cause the vectors to fall out of the domain of the logarithm term. But by introducing a new variable $\mathbf{y} = \mathbf{x}$ and using ADMM, we avoid this issue.
> We added a comment to point this out, thanks!
>
> **2.  Assumption in Theorem 3.** Assumption (iii) in Theorem 3 requires that $F(\mathbf{x}^{\star})\ne \boldsymbol{0}$, which indicates that some constraints are active at $\mathbf{x}^\star$. This is because if no constraints are active at $\mathbf{x}^\star$, then $\mathbf{x}^\star$ is an interior point of the constraint set. In this case, (cVI) in Sec. 1 holds if and only if $F(\mathbf{x}^\star)=\boldsymbol{0}$.  We discuss this right after Theorem 3.
>
> **3. Empirical comparison with Frank-Wolfe.**   Frank-Wolfe (FW) methods exploit a given structure of the constraints. For general form constraints, its approximate variants may be extensively slow; hence ACVI has the advantage that can solve VIs with constraints of a more general form.
>
> In addition, FW has not been extended to VIs, and hence it lacks a convergence guarantee. It has been extended only to zero-sum min-max -- we added experiments for such min-max problems in App. E.2.
> Nonetheless, since there are different FW variants we leave extending these to VIs for future work as well as a more extensive comparison to ACVI (see our discussion in App. E.2.). Note that different variants can be derived from ACVI as well to incorporate some structure of the constraints; herein we focus only on the general-form constraints, and exploiting some structure of the constraints is a promising direction.
>
> > Even though the paper claims the algorithm is parameter-free, we may need access to problem parameters when solving the subproblems mentioned above.
>
> Thanks for the question. There are two cases:
>
> - For problems where these sub-problems *can* be solved analytically---which is the focus of this paper, we do not need access to the problem parameters to guarantee convergence.
>
> - For the case when the subproblems need to be solved through some gradient-based method, one can use parameter-free methods, e.g.,
>  the one in  Diakonikolas (2020) mentioned in the related works section; and the overall method will be parameter-free.
>
>
> Hence, our method is parameter-free in either case. Note that it is standard (and practically relevant, see discussions in  Diakonikolas (2020)) to make a method parameter-free through techniques such as line search (in our case through the outer loop), this comment is just saying that our analyses are complete in that sense.

---

> > ### Author Response · Authors · 2022-11-17
> > **Following up on the discussion with Reviewer WuKs**
> >
> > Dear Reviewer WuKs,
> >
> > The questions asked were mainly about the parameter-free guarantee, and while the main part lists the non-parameter-free guarantee for simplicity, the former, as we pointed out, is carried out in the Appendix in detail.  In the revised version, we also added a comparison with Frank-Wolfe.
> > We believe our response fully answers the concerns raised, and if no additional clarifications are needed, please consider re-evaluating your score.
> >
> > Thank you very much for your valuable time to review our work

---

### Official Review · Reviewer_2xGb · 2022-10-26

**Confidence:** 4
**Clarity, Quality, Novelty And Reproducibility:** See above.
**Correctness:** 3
**Technical Novelty And Significance:** 3
**Empirical Novelty And Significance:** 2
**Recommendation:** 6

**Strength And Weaknesses:**

The paper is considering a challenging problem to solve. VI is an important topic of study and including complex constraints seems to be a completely unexplored area. However, there are some problems with the existing results which I would like to point out:

1. My main concern is in the update step of x. It seems that to solve every single step, one requires to solve a system of nonlinear equations, which is a VI problem itself. This problem is challenging and for many applications, one may need a separate solver to get such soltions. However, in the analysis, it seems to be solved exactly. Current complexity results take this as an oracle. Effectively, the theoretical results don't capture the actucal computational complexity of the problem and hence, they are deceptive in nature. Note that this oracle needs to be called in every inner iteration. This makes the technical implementation to be a three-loop algorithm.

2. Update of y may not be easy for general \phi. Hence, even though the starting framework is general, the method is restrictive in nature.

3. The paper claims that Lipschitz smoothness (Assumption 1) is not required in general. However, if I am not ignoring anything, the reason is precisely that x updated solve F as VI exactly for each inner look. If subproblem solvers are taken into consideration then solving such VI for arbitrary F may already be difficult.

4. The paper in the main part does not discuss how to large T (number of outer loops) needs to be? The convergence rate results are devoid of T. I believe we must have a small mu to get to a good soltuion. However, the error introduced by mu is not characterized in the discussion.




**Summary Of The Paper:**

This paper considers variational inequality (VI) problems where complex constraints exist. They use the framework of the interior point method with log-barriers to handle the constraints. They merge this framework with the ADMM algorithm to provide the final design of the algorithm which is a two-loop algorithm. The outer loop decreases the penalty parameter progressively which is similar to traditional IP methods for optimization. The inner loop is for implementing the ADMM algorithm.They provide convergence rate for each outer loop (t) in various settings including monotonicity, strong/\xi monotonicity, their "star" (or generalized) versions.

**Summary Of The Review:**

Overall, I believe the paper is studying an important and quite general framework. But it falls short on many fronts as explained in the weakness part.

---

> ### Author Response · Authors · 2022-11-12
> **Response to Reviewer 2xGb**
>
> Thank you for your time reviewing our paper and for your feedback.
>
> **1. On the update step of x.**
>
> - In this paper, for our theoretical results, we consider only the case when the sub-problems can be solved analytically (in this case, there is no added complexity). Among others, this includes the class of affine variational inequalities (AVIs), low-dimensional problems, and when optimization variables represent probabilities, for example.
> This is a smaller problem class, but it is yet very practically relevant because it arises often in applications. We point this out in Sec. 4 (above Alg. 1), and discuss it further in App. B.5.
>
>
> - Note, however, that if we cannot solve the problems exactly, there could be extensions of ACVI which may not necessarily be three-loop. To see this, notice from Algorithm 1, that the $\mathbf{x}$ sub-problem changes relatively little over the iterations (only $\mathbf{y}$ and $\mathbf{\lambda}$ change). Hence, it may be worth *(i)* solving this sub-problem approximately by running several iterations of GDA or EG (say 100) *only the first time* it is ever solved; and then *(ii)* at each consecutive iteration initializing $\mathbf{x}$ to be the approximate solution found at the previous step and only run one step of GDA (or EG). This approach---known as warm-starting---adds only a constant overhead (it remains 2-loop, and only at the first iteration we have  3 loops).
> We allude to this technique empirically (there may be other approaches/techniques as well), but such ACVI variants are out of the scope of this paper.
>
>
> - In addition, we provide empirical insights that show the approach in this paper is better than the three-loop method to apply the herein approach for the projection operator (so as to remain having a first-order method)---due to the extensive zig-zagging of the latter when hitting a constraint (see Fig. 1). The MNIST experiments also show that our algorithm is practical even when line 8 is difficult to solve.
>
> **2. On the update of $\mathbf{y}$.** For any $\varphi$ that satisfies $\varphi_i$ is convex, $i=1,2,\cdots,m$, updating $\mathbf{y}$ only requires solving an unconstrained strongly convex problem. Even when the dimension is very high, like in deep neural networks, this problem could be easily solved simply by the gradient descent method.
> In contrast, the projection step in the projection-based methods usually requires solving a much more complicated constrained convex programming problem by another solver; hence, to our knowledge, it is the least restrictive method to solve CVIs.
>
> Note that in our experiments on GANs, for lines 8 and 9 of Alg. 1, we run $l$ ($l=1,10$ in our experiments) steps of (unconstrained) GDA/EG and GD, respectively, see App. D (in particular Algorithm 4) for more details. This approach is easy to implement and outperforms the baselines.
>
> **3. Regarding the Lipschitz smoothness.**
> We respond for the two cases:
>
> - For problems where these sub-problems *can* be solved analytically, having such a guarantee (without L-Lip. asm.) is an advantage of ACVI; other methods do not have such a guarantee for this problem class.
> Non $L$-Lipschitz operators are a different yet wide class of problems (just like in minimization).  As an example, consider $F(\mathbf{x})=(x_1^2,x_2^2, \cdots, x_n^2)^\intercal$, in which case $F$ is not L-smooth, hence existing proofs do not hold.
> Note that $\forall \mathbf{x},\mathbf{y}$, $\langle F(\mathbf{x})-F(\mathbf{y}), \mathbf{x}-\mathbf{y}\rangle=\sum_{i=1}^{n}(x_i^2-y_i^2)(x_i-y_i)=\sum_{i=1}^{n}(x_i+y_i)(x_i-y_i)^2$, from which we can see that $F$ is monotone in $\mathbb{R}^n_+$. We let $\varphi(\mathbf{y})=-\mathbf{y}$. For this example, both line 8 and line 9 can be solved analytically (we only need to solve $n$ single-variable quadratic equations for both lines).
>
> - Moreover, (in our opinion) one can further use the above-described approach when solving the subproblems approximately (of warm-starting) and exploit certain structures to guarantee convergence for problems where L-Lipschitzness is a non-realistic assumption.
> Hence, it is unclear that your comment is true even for the approximate case (since in the minimization case we do have convergence guarantees without L-Lipschitzness, but while assuming other structures; so a similar approach can be used), and this is a nice future direction.
>
> **4. How large $T$ should be.**
> As we remark in the main paper, we state the theorem in a way that requires us to know that parameter (as is standard in the literature), but we add the "complete" (parameter-free) convergence rate which takes the outer loop into consideration in App. B. 4.

---

> > ### Author Response · Authors · 2022-11-17
> > **Following up on the discussion with Reviewer 2xGb**
> >
> > Dear Reviewer 2xGb,
> >
> > We believe we fully addressed your raised concerns, and we included more explicit analyses for question 4 in App. B.4.
> > Regarding your questions 1 and 2, there is indeed room for future work, but our theoretical guarantees already address a relevant and wide class of problems, and the empirical results allude to implementations for general cVIs.
> > We would like to emphasize that the VI setting is significantly more challenging to analyze, and in this paper, we first point out a problem of projection-based methods which arises in the VI context, and then we both propose a novel principled method and provide convergence rates (contrasting this with works in the unconstrained setting, where novelty in the method and convergence rates for such general settings are typically derived over multiple papers). We highlight that extensions--where the subproblems are replaced with one gradient update are possible--but the convergence guarantees are left for future work.
> > Please let us know if further clarifications are needed, or alternatively please consider re-evaluating your score.
> >
> > Thanks a lot for your valuable feedback to improve our paper

---

### Official Review · Reviewer_9sD3 · 2022-10-28

**Confidence:** 3
**Correctness:** 4
**Technical Novelty And Significance:** 4
**Empirical Novelty And Significance:** Not applicable
**Recommendation:** 8

**Clarity, Quality, Novelty And Reproducibility:**

*Clarity:* I found the paper quite clear, except for the very technical parts in the appendix (the author(s) may be forgiven for that).

*Novelty:* to the best of my knowledge, this is the first method for VI's providing feasible solutions without the need for projections or Jacobian computations.

*Reproducibility:* I did not check the experiments carefully, but many details are given in the appendix about the experiments. As usual, there is a bit of an issue when CPU time is considered.

*Quality:* in spite of the limiting assumptions, I find this to be a worthwhile conceptual contribution to the VI literature, with potential for practical impact. Do note that I did not check the calculations to the last line.

**Strength And Weaknesses:**

*Strengths*

- VIs form a very important family of problems.
- The method proposed seems genuinely new.
- Empirical performance in the (small) experiments is promising.
- The theoretical analysis is quite interesting.

*Weaknesses*

- The assumptions required on the VI are quite strong. However, it is conceivable that they can be weakened significantly.
- The main theorems "hide" the dependence of constants; this is standard in the literature, but can in principle hide important factors depending on the dimension, on the conditioning of $C$ or any other such important information.

**Summary Of The Paper:**

The paper considers variational inequalities whose feasible sets include linear equality and convex inequality constraints.

When the VI operator is monotone, there are known globally convergent methods (such as Korpelevich's projected extragradeint) for solving these problems, with known averaged and last-iterate convergence rates. However, these methods require metric projections onto the feasible set, or Jacobian computations for the VI ooperation, which can be quite costly.

The goal of the paper is to develop an interior-point method that bypasses the need for projections, at the cost of requiring

(i) a single matrix inversion at the beginning;
(ii) that the (W-EQ) problem in the text be efficiently solvable (this is the case eg. for linear VI's as those occurring in classical zero-sum games);
(iii) also, properties stronger than strict monotonicity, plus that the VI operation $F$ be "non-rotational."

Under these strong assumptions, averaged and last-iterate convergence rates are proven. The method itself is a variant (in some sense) alternating direction method of multipliers. In particular, it is purely first-order. Additionally, its interior nature means that it requires no projections.

Some experiments suggest that the new method compares favorably with alternatives in the literature. Intriguingly, this seems to hold even when the VI operation is very "rotational."



**Summary Of The Review:**

This paper offers a novel method for solving VIs that circumvents some of the limitations of projection-based approaches and second order methods. The method is globally convergent under strong assumptions, and shoing this requires a nontrivial analysis. Moreover, it seems to perform well in experiments.

---

> ### Author Response · Authors · 2022-11-12
> **Response to Reviewer 9sD3**
>
> Thank you very much for the time you took to review our paper, and for your appreciation of this work.
>
> **Regarding the assumptions.** Considering that existing proofs for MVI use computer-aided proofs (unlike ours), and that our $\xi$-monotonicity assumption is much wider class than strong monotonicity, the assumptions are somewhat on par with relevant results. Nonetheless, we agree that these are somewhat stronger than just monotonicity (we leave this for next step).
>
> **The constants in the rates.** We agree that having the constants is clearer. But since the expressions are quite long in our case, due to space constraints we list these fully in the appendix (see Theorems 10 and 11--- restatements of Theorems 2 and 3, respectively).
> In the main part, we show the most important $L$-Lipschitz constant (which only appears for the third statements of the theorems).

---

### Official Review · Reviewer_P1Mi · 2022-10-31

**Confidence:** 3
**Correctness:** 4
**Technical Novelty And Significance:** 3
**Empirical Novelty And Significance:** 3
**Recommendation:** 6

**Clarity, Quality, Novelty And Reproducibility:**

The paper is well-written and easy to follow. This is the first algorithm under the considered setting with both solid theoretical analysis and sufficient empirical support.

**Strength And Weaknesses:**

**Strength:**

The paper provides the first first-order interior point method for constrained variational inequalities with a global convergence guarantee. A clear roadmap toward how the algorithm is built as well as its theoretical analysis is provided. Extensive numerical experiments are presented to demonstrate the clear advantage of the algorithm. It is surprising to see that the proposed algorithm can efficiently approach the optimal point while others cycle around the optimal or zigzag near the constraints.

**Weaknesses:**

1. The outer loop $T$ of Algorithm 1 is not discussed precisely, which might cause some confusion. I guess the role of $T$ is to ensure that $\mu_t$ (line 9) is sufficiently small to handle the constraints $\varphi$. The complexity should also take into account $T$ and the subproblems, which will contribute additional terms, and thus the lower bounds are only matched up to logarithmic factors. The algorithm can be difficult to implement, also because of this unavoidable triple-loop structure.

2. I am a little bit confused about how to compare the results in Theorem 2 ($\xi$-monotone) and 3 (monotone). The rates in Theorem 2 are strictly worse than in Theorem 3 when $\xi>1$, but $\xi$-monotonicity should be a stronger assumption.

3. Could the author comment why assumption $(iii)$ is needed in Theorem 3? Why does it require $F(x^*)\neq 0$, i.e., some constraints are active at $x^*$? I think this is implied by the condition $\inf F(x^*)^\top \cdots>0$ that is used for the average guarantee.

4. Minor: In Section 2, the term "MVI" is not introduced; In Section 3, typos in eq. (AL-CVX); In Section 4, the notation $\perp$ is not defined in eq.(KKT); The paper states that the algorithm approaches the solution from the analytic center, which is the reason why cyclical behavior is reduced. Could the author explain why?



**Summary Of The Paper:**

The paper derives a first-order algorithm for constrained variational inequalities based on the interior point method and ADMM. Convergence rates of the algorithm for both $\xi$-monotone and monotone cases are provided. Numerical experiments show its excellent behavior in practice compared to existing works.

**Summary Of The Review:**

The contributions are enough for acceptance.

---

> ### Author Response · Authors · 2022-11-12
> **Response to Reviewer P1Mi**
>
> We thank you for your time to review the paper and for your feedback. We addressed the minor comments (please see the revised version). Below we answer the remaining questions.
>
> **1.1 Outer loop $T$.**
>
> Thanks for the question.
>
> - Yes, the role of $T$ is to ensure that $\mu_t$ (line $9$) is sufficiently small to handle the constraints $\varphi$. In the main part, we give parameter-depend guarantees and rates; and in this case, $T$ is irrelevant.
>
> - We introduce $T$ to make our method parameter-free, that is, to be independent of the choice of $\mu$. We discussed this below the Theorem $11$ (in the Appendix) in the original submission, and now we extended the discussion in App. B.4, and point to it from Remark 2, Sec. 4.2. Thanks for the suggestion, and please let us know in case the added discussions are still unclear.
>
> **1.2 Difficulty to implement the algorithm due to the loops.**
>
> - The only other alternative when one has general-form constraints is a three-loop, and ours is a two-loop in the exact solving case, and can be made two-loop even if the subproblems can't be solved analytically, as we explain below.
>
> - Also, as the discussion above suggests, we can remove the outer loop or run very few outer loop iterations when $t<T$ to reduce the computation. We use this approach for our MNIST experiments, see App. D.3 for discussion (and Alg. 4), which outperforms all the baselines.
>
> - Further, notice that the sub-problems change only a little bit over the iterations, thus when solving the sub-problems approximately, we can initialize the optimization variable to the one from the previous solution and run only one step of some gradient method. Hence, extensions of Alg. 1 are possible to "convert" the sub-problems to only one update.  However, this is beyond the scope of this paper, and here we focus on the cases when the sub-problems can be computed analytically. This class of problems is smaller relative to the entire MVI class, but such problems arise relatively often (we discuss this in App. B.5).
>
> **2. On the assumptions in Theorem 2 and 3.**
>
> -The assumptions in Theorem 2 and Theorem 3 are very different, hence they are not directly comparable. Theorem 3 requires $\text{inf}_{\boldsymbol{x} \in S \backslash \{ \boldsymbol{x}^\star \} } \ \ F \left( \boldsymbol{x} \right) ^\intercal\frac{\boldsymbol{x}-\boldsymbol{x}^\star}{\lVert \boldsymbol{x}-\boldsymbol{x}^\star \rVert}>0$,
> where $S \equiv \hat{\mathcal{C}}_r$ or $\tilde{\mathcal{C}}_s$, while Theorem 2 does not. So we could not conclude that $\xi$-monotonicity is stronger--and the reverse is not true as well, see our discussion below Theorem 3.
>
> **3. On the assumption *(iii)*  in Theorem 3.**
>
> -Yes, the requirement of
> $F(\mathbf{x}^\star)\ne 0$ is implied by the condition
> $\underset{ \boldsymbol{x}\in S \setminus \{ \boldsymbol{x}^\star\}}{inf}F (\mathbf{x}^\star) ^\intercal
> \frac{ \boldsymbol{x}-\boldsymbol{x}^\star}{
> \lVert{ \boldsymbol{x}-\boldsymbol{x}^\star }\rVert
> } >0$
> (with $S \equiv \hat{\mathcal{C}}_r$ or $\tilde{\mathcal{C}}_s$),  used for the average guarantee.
>
> Assumption *(iii)* is equivalent to the cosine of the angle between vector $F(\mathbf{x}^\star)$ and vector $\boldsymbol{x}-\boldsymbol{x}^\star$ (i.e. $\cos\theta_\mathbf{x}\triangleq \frac{F(\mathbf{x}^\star)^\intercal(\boldsymbol{x}-\boldsymbol{x}^\star)}{\lVert F(\mathbf{x}^\star)\rVert \lVert{ \boldsymbol{x}-\boldsymbol{x}^\star }\rVert}$) has a positive lower bound on $S \setminus \{ \boldsymbol{x}^\star\}$.
> We require assumption *(iii)* in Theorem 3 because in Theorem 9 in App. B, we have shown that
> $|| F(\mathbf{x}^\star)^\intercal(\hat{\mathbf{x}}_{K+1}^{(t)}-\mathbf{x}^\star) || \leq \mathcal{O}(1/\sqrt{K}).$
>
> With it, we can upper bound $ || \hat{\mathbf{x}}_{K+1}^{(t)}-\mathbf{x}^\star ||$ by
>
> $ F( \mathbf{x}^\star)^\intercal (\hat{\mathbf{x}}_{K+1}^{(t)} - \mathbf{x}^\star )$.
>
> From this, we deduce that
> $ || \hat{\mathbf{x}}_{K+1}^{(t)} - \mathbf{x}^\star || \leq \mathcal{O}(1/\sqrt{K}) $.
>
> We note that a  similar assumption to (iii) is made in the convergence proof of the exact line search method, see for example [Chapter 3.2 of Nocedal, Jorge, and Stephen J. Wright. "Numerical optimization 2nd edition." (2006)]. In particular, the global convergence of the line search method requires the assumption that the angle between the search direction and the gradient in each iteration is bounded away from $90\degree$.

---

> > ### Author Response · Authors · 2022-11-17
> > **Following up on the discussion with Reviewer P1Mi**
> >
> > Dear Reviewer P1Mi,
> >
> > The questions asked were mainly about the outer loop, which we addressed in App. B.4 in the revised version.
> > We believe our responses address your concerns, and provided no additional clarifications are needed, please consider re-evaluating your score.
> >
> > Thank you very much for your valuable time and feedback

---

### Decision · Program_Chairs · 2023-01-20

**Decision:**

Accept: notable-top-25%

**Justification For Why Not Higher Score:**

Strong assumptions.

**Justification For Why Not Lower Score:**

Novel algorithm for a challenging problem.

**Metareview: Summary, Strengths And Weaknesses:**

This well-written paper studies the challenging problem of solving constrained variational inequality where constraint projection operation is costly. Authors propose, possibly the first method for such problems, which combines ADMM and Interior Point methods and provide a convergence rate for their parameter-free method under some strong assumptions. Authors also provide experimental results comparing their method with other known baselines to showcase their computational efficiency. The main drawback for the procedure is that it assumes exact solvability of two different sub-problems which may be computationally challenging. This means that the objective and constraints need to be simple enough. Further it is not clear how the convergence rate change when we only have approximate solutions for the inner problems and whether the algorithm will still remain parameter-free. Reviewers identified a few unclear parts, which. the authors seems to have partially addressed. I hope the authors paraphrase these discussion in the next revision.

Nit: The term "rotational" is not defined.

**Note From Pc:**

if the above contains the word "oral" or "spotlight" please see: "oral" presentation means -> notable-top-5% and "spotlight" means -> notable-top-25%. As stated in our emails, we are disassociating presentation type from AC recommendations